# Inference-time Alignment with Rewards in Besov Spaces: Provable Advantages of Feature Learning and Multi-Step Policy Updates

**Naoki Nishikawa** [1 2]  **Taiji Suzuki** [1 2]

## Abstract

Inference-time alignment, the approach of adapting pre-trained models to reward feedback during inference, has proven empirically effective at improving language-model performance. Despite its success, theoretical foundations remain underdeveloped, especially in practical settings where neural networks are employed as reward models. In this paper, we explore the advantages of neural networks and how to effectively train them for inference-time alignment. Assuming that the true reward function lies in Besov spaces to capture the non-uniform smoothness, we compare neural networks to linear estimators and show that feature learning capability of neural networks is crucial for improving performance. We further analyze algorithms for training neural-network reward estimators. Specifically, we consider a multi-step algorithm that alternates between sampling from the current policy and refitting the reward estimator, and prove that it improves the regret, especially when the true reward exhibits local structure.

## 1. Introduction

Inference-time compute (Brown et al., 2024; Snell et al., 2024; Wu et al., 2024; OpenAI, 2024; Guo et al., 2025) has been attracting attention as a new paradigm for further enhancing the performance of pre-trained language models (LMs). By effectively leveraging the computational budget available at inference time, one can enhance the quality of model outputs without being restricted to pre-constructed datasets. A variety of techniques are included in this paradigm, e.g., long chains of thought (Wei et al., 2022; Li et al., 2024), self-evaluation and revision

of their own outputs (Zheng et al., 2023; Wu et al., 2025), and exploration of improved responses (Yao et al., 2023; Zhang et al., 2024). Among these approaches, *inference-time alignment*, a framework that samples LM responses to maximize the reward, has been shown to offer a simple yet highly effective means of improving performance.

The methods for inference-time alignment have been widely studied from theoretical perspectives. For example, Yang et al. (2024); Beirami et al. (2025); Mroueh & Nitsure (2025) analyzed the performance of Best-of-$N$ alignment, which is the most basic method for inference-time alignment. Moreover, Huang et al. (2025b) pointed out the limitations of Best-of-$N$ alignment and proposed a new method based on $\chi^2$-divergence regularization. While these studies give insights into how each method is effective, their analyses are mainly under the fixed reward model and do not incorporate the process of training the reward model. Foster et al. (2025) analyzed the training of reward models and showed the advantage of a multi-turn exploration method. However, their analyses focus on the setting where the true reward and its estimator are linear with fixed features, which is far from practical settings where neural networks (NNs) are used. This raises the following question:

> *What advantages do NNs offer for inference-time alignment, and how can we unlock their full potential?*

One of our concrete questions is how the *feature learning* capability of NNs can improve inference-time alignment. While feature learning is known to be crucial in supervised learning, it remains unclear how it helps in inference-time alignment, where the goal is to produce responses with high true reward. To investigate this, we model the true reward function using an extension of Besov spaces (DeVore & Popov, 1988; DeVore et al., 1993) and study the performance of NN-based reward estimators under this function class. In particular, we theoretically compare NNs with *linear estimators*, a family of methods that lack feature learning, and characterize how NNs yield provable advantages.

Another question we address is how to train and leverage NN-based reward estimators to improve the quality of responses. Most existing theoretical studies treat the reward estimator as fixed and focus on designing inference-time algorithms. We first show a regret bound when simply lever-

---

[1]The University of Tokyo, Tokyo, Japan [2]RIKEN AIP, Tokyo, Japan. Correspondence to: Naoki Nishikawa <nishikawa-naoki259@g.ecc.u-tokyo.ac.jp>.

aging a reward estimator trained via standard regression in a prior inference-time alignment algorithm. Moreover, to go beyond this plug-in approach, we further consider a *multi-round* inference-time alignment scheme, where we iteratively update both the reward estimator and the policy by querying a reward oracle at each round. This iterative procedure allows the policy to progressively concentrate around the optimal response, leading to better aligned outputs. We provide a theoretical analysis that quantifies how the regret can be further reduced by this multiple-update approach.

Our contributions are summarized as follows:

1. **Superiority of NNs against linear estimators.** We demonstrate the advantage of NNs in inference-time alignment. Specifically, we consider the setting where the true reward function lies in Besov spaces. The Besov space is a general class of functions that has non-uniform smoothness over the input domain, which requires our estimator to perform feature learning for the optimal estimation error rate (Suzuki, 2019). We derive the upper bound on regularized objective for inference-time alignment when we employ NN-based reward estimator. Moreover, we show sub-optimality of alignment methods based on a reward model based on linear estimators by leveraging the fact that linear estimators cannot achieve optimal rate to estimate the true reward. This highlights the advantage of feature learning ability by NNs in inference-time alignment.

2. **Regret bound for NN-based reward estimator.** We derive an upper bound of the regret for inference-time alignment when the true reward lies in Besov spaces. We utilize the result of Huang et al. (2025b) that characterizes the regret bound by the squared loss error and the *coverage*, which measures how large the pretrained generative model has mass around the maximum reward point (Jin et al., 2021; Xie et al., 2021; Zhu et al., 2023; Zhan et al., 2024; Li et al., 2023; Xiong et al., 2024).

3. **Improved analysis of regret by multiple-step update.** We also analyze an algorithm that iteratively and actively learns NN-based reward models from the responses of the trained model, and show that it achieves a smaller regret. Since our theoretical analysis requires boundedness of the coverage throughout the algorithm, we utilize a novel Gaussian perturbation technique. With the help of this method, we show that the regret is improved by multi-step updates.

### 1.1. Other Related Works

**Capabilities of NNs in Regression.** Theoretical analyses of NNs and their advantages over other models have been extensively studied in the context of regression problems. For example, Schmidt-Hieber (2020) and Suzuki (2019)

showed that NNs can achieve minimax optimal rates for estimating functions in Hölder spaces and Besov spaces, respectively. Suzuki & Nitanda (2021) extended the analysis to the case of anisotropic Besov spaces (Nikol'skii, 1975; Vybiral, 2006; Triebel, 2011). They also showed the lower bounds on the estimation error for linear estimators, demonstrating the superiority of NNs over linear estimators. Hayakawa & Suzuki (2020) also analyzed the upper bounds for NNs and lower bounds for linear estimators, and showed that NNs are superior to linear estimators for function classes with sparsity. Furthermore, Petersen & Voigtlaender (2018) and Imaizumi & Fukumizu (2019) analyzed the estimation error of NNs for complicated functions with piecewise smoothness. Unlike these studies, our analysis focuses on the setting of inference-time alignment, which aims to find the response that maximizes the reward function, rather than minimizing the estimation error.

**Theory on maximizing black-box functions.** Our study is highly related to the literature on black-box optimization. In particular, previous studies such as Minsker (2012), Minsker (2013), Grill et al. (2015), Wang et al. (2018) and Singh (2021) consider the setting where the objective function lies in RKHS, Hölder or Besov spaces, sometimes with additional assumptions on the structure of the function. While some of the techniques from these studies can be applied to our analysis, this paper differs in two aspects: (i) our analysis considers the setting of inference-time alignment, where the function to maximize is conditioned on a prompt; (ii) we assume some additional structure on the reward function, and demonstrate how the advantage of NNs and multi-step training emerge.

### 1.2. Notations

Let $d_X, d_Y \in \mathbb{Z}_{>0}$ be the dimensions of prompts and responses, respectively, and let $d = d_X + d_Y$. Let $\Omega_X = [0,1]^{d_X}, \Omega_Y = [0,1]^{d_Y}, \Omega = [0,1]^d$. Moreover, for $\epsilon > 0$, we define $\Omega_X^{[\epsilon]} := [\epsilon, 1 - \epsilon]^{d_X}$ and $\Omega_Y^{[\epsilon]} := [\epsilon, 1 - \epsilon]^{d_Y}$. Let $\lambda$ be the Lebesgue measure on $\Omega$.

For a probability density $p$ on $\Omega$ and a measurable set $A \subset \Omega$ with $\int_A p(x) \, dx > 0$, we write $p \mid A$ for the conditional density of $x \sim p$ given the event $\{x \in A\}$, i.e., $(p \mid A)(x) = \frac{p(x) \mathbb{1}_A(x)}{\int_A p(x) dx}$.

For a function $f : \Omega \to \mathbb{R}$, let $\|f\|_p := \|f\|_{L^p(\Omega)} := \left(\int_\Omega |f|^p dx\right)^{1/p}$ for $p \in (0, \infty]$, and $\|f\|_\infty := \|f\|_{L^\infty(\Omega)} := \sup_{x \in \Omega} |f(x)|$. We also define $L^{(p,q)} := \{f : \Omega_X \times \Omega_Y \to \mathbb{R} \mid \|f\|_{(p,q)} < \infty\}$ for $1 \le q \le p \le \infty$, where the norm $\|\cdot\|_{(p,q)}$ is defined as $\|f\|_{(p,q)} := \left(\int_{\Omega_Y} \|f(\cdot, y)\|_p^q dy\right)^{1/q}$.

For $\iota > 0$, a set $S$, a metric $\rho$, let $B(x, \iota; \rho)$ be the $\rho$-ball

with center $x$ and radius $\iota$, and $\mathcal{M}(\iota; S, \rho)$ be the $\iota$-covering number of $S \subset \mathbb{R}^d$ with respect to $\rho$. In particular, for the Euclidean norm $\|\cdot\|_2$, we denote $B_2(x, \iota) := B(x, \iota; \|\cdot\|_2)$ and $\mathcal{M}_2(\iota, S) := \mathcal{M}(\iota; S, \|\cdot\|_2)$.

## 2. Problem Setting

In this section, we formulate the problem setting of inference-time alignment with rewards in Besov spaces. We first introduce the formal setting of inference-time alignment and describe algorithms for it. Then, we define Besov spaces.

### 2.1. Inference-time Alignment

Inference-time alignment is a problem of generating a response $y \in \Omega_Y$ with high reward for a given prompt $x \in \Omega_X$. More formally, let $P_X$ be a distribution over $\Omega_X$ and $\pi_{\text{ref}}(y \mid x)$ be a base policy, which is typically a pre-trained language model. Let $r^\circ : \Omega \to [-R, R]$ $(R > 0)$ be a reward function that evaluates the quality of the response $y$ for the prompt $x$. We can only access the reward function through an oracle that returns

$$r^\dagger = r^\circ(x, y) + \xi, \quad \xi \sim \mathcal{N}(0, \sigma^2), \tag{1}$$

i.e., a noisy observation of the reward for a given prompt-response pair. Since observing rewards is expensive (e.g., it requires human evaluation), we can only access a limited number of samples from the oracle.

We note that, for simplicity, prompts and responses are represented as vectors rather than token sequences in our setting. This can be interpreted as embedding the texts into vectors using a pre-trained encoder.

As a technical assumption, we assume that $P_X$ has a bounded density $\rho$ satisfying $\rho(x) \in [\underline{\rho}, \overline{\rho}]$ for all $x \in \Omega_X$, where $\underline{p}, \overline{p} > 0$ are universal constants. Moreover, we assume that $\pi_{\min} \leq \pi_{\text{ref}}(y \mid x) \leq \pi_{\max}$ for all $x \in \Omega_X$ and $y \in \Omega_Y$, where $\pi_{\min}, \pi_{\max} > 0$ are universal constants.

**Algorithms for inference-time alignment.** The simplest method for inference-time alignment is *Best-of-N sampling* (BoN), where we generate $N$ candidate outputs $y_1, \ldots, y_N$ from the reference policy $\pi_{\text{ref}}(\cdot \mid x)$ for a given prompt $x$, and return the one with the highest estimated reward, i.e., $y_{i^*}$ with $i^* := \arg\max_{i \in [N]} \widehat{r}(x, y_i)$. The reward estimator $\widehat{r}$ (called *reward model*) is trained beforehand using oracles in (1). While BoN performs well empirically and theoretically (Lightman et al., 2024; Brown et al., 2024; Huang et al., 2025a), it is known to sometimes suffer from *over-optimization* or *reward hacking*: BoN may over-trust the reward model and select responses that score highly under $\widehat{r}$ but have low true reward $r^\circ$.

A common way to mitigate over-optimization in LM alignment is to regularize the policy so that it stays close to the reference policy (e.g., Ziegler et al. (2019), Stiennon et al. (2020) and Nakano et al. (2021)). More specifically, we use the policy $\pi(\cdot \mid x)$ that optimizes the following regularized objective:

$$\mathcal{L}(\pi)(x) := \mathbb{E}_{y \sim \pi(\cdot \mid x)}[\widehat{r}(x, y)] - \mu \cdot D(\pi(\cdot \mid x) \| \pi_{\text{ref}}(\cdot \mid x)),$$

where $D(\cdot \| \cdot)$ is a divergence between two distributions, e.g. KL divergence. In particular, Huang et al. (2025b) propose to use the $\chi^2$-regularized objective and show that it effectively avoids over-optimization and achieves better regret than BoN. They also propose an algorithm called *Inference-time Pessimism* to approximately sample from the optimal policy under the $\chi^2$-regularized objective.

### 2.2. Besov Spaces with Anisotropic Integrability

In this paper, we assume that the reward function $r^\circ$ lies in a Besov space with anisotropic integrability. This space is a function class that extends the ordinary Besov space. Roughly speaking, Besov space with anisotropic integrability allows the regularity (or uniformity of smoothness) to differ across directions (prompts $x$ or responses $y$, in this paper). We provide its formal definition here.

For a function $f : \Omega \to \mathbb{R}$, we define the *r-th difference of $f$ in the direction $h \in \mathbb{R}^d$* as

$$\Delta_h^r(f)(x) := \Delta_h^{r-1}(f)(x + h) - \Delta_h^{r-1}(f)(x),$$
$$\Delta_h^0(f)(x) := f(x),$$

for $x \in \Omega$ with $x + rh \in \Omega$, otherwise, let $\Delta_h^r(f)(x) = 0$.

**Definition 1** (Modulus of Smoothness). For a function $f \in L^{(p_X, p_Y)}(\Omega)$ where $p_X, p_Y \in [1, \infty]$, the *r-th modulus of smoothness* of $f$ is defined by $w_{r,(p_X,p_Y)}(f, t) = \sup_{h \in \mathbb{R}^d : \|h\|_2 \leq t} \|\Delta_h^r(f)\|_{(p_X, p_Y)}$ for $t > 0$.

In short, the modulus of smoothness is the $L^p$-norm of the $r$-th order finite derivative. With this modulus of smoothness, we define the Besov space $B_{(p_X,p_Y),q}^s(\Omega)$ as follows.

**Definition 2** (Besov Space with Anisotropic Integrability). For $1 \leq p_Y \leq p_X \leq \infty, 0 < q \leq \infty, s > 0, r := \lfloor s \rfloor + 1$, let the semi-norm $|\cdot|_{B_{(p_X,p_Y),q}^s}$ be

$$|f|_{B_{(p_X,p_Y),q}^s} := \begin{cases} \left(\int_0^\infty (t^{-s} w_{r,(p_X,p_Y)}(f, t))^q \frac{dt}{t}\right)^{\frac{1}{q}} & (q < \infty), \\ \sup_{t > 0} t^{-s} w_{r,(p_X,p_Y)}(f, t) & (q = \infty). \end{cases}$$

The Besov space with anisotropic integrability $B_{(p_X,p_Y),q}^s(\Omega)$ is defined as

$$B_{(p_X,p_Y),q}^s(\Omega) := \left\{ f \in L^{(p_X,p_Y)}(\Omega) \,\Big|\, \|f\|_{B_{(p_X,p_Y),q}^s} < \infty \right\}.$$

where $\|f\|_{B_{(p_X,p_Y),q}^s} := \|f\|_{(p_X,p_Y)} + |f|_{B_{(p_X,p_Y),q}^s}$.

Intuitively, the parameter $s$ represents the smoothness of each coordinate of the function. If $s$ is large, then the function is smooth. The parameter $p_X$ and $p_Y$ represent *uniformity* of the smoothness over the input space $\Omega$. We see that, when they are small, the smoothness of functions in the

class is guaranteed only in an average sense over the input domain, hence the function can have local bumps around some input point $x$. *The feature learning ability plays a crucial role in detecting such a bumpy point* to achieve the optimal rate (Suzuki, 2019).

When $p_X = p_Y = p$, the space $B^s_{(p_X, p_Y), q}(\Omega)$ reduces to the ordinary Besov space (DeVore & Popov, 1988; De-Vore et al., 1993). Moreover, if $p_X = p_Y = q = \infty$, then $B^s_{(p_X, p_Y), q}(\Omega)$ coincides with the Hölder space $C^s(\Omega)$ (Triebel, 2011).

Throughout this paper, we assume that the true reward $r^\circ$ belongs to the space $B^s_{(\infty, p), q}(\Omega)$ for some $p \in [1, \infty]$. This setting corresponds to the situation where the true reward has uniform smoothness for the direction of the prompts, while having non-uniform smoothness for the direction of the responses. For simplicity, we denote this space by $B^s_{p,q}(\Omega)$. This assumption captures the scenario in which the reward is uniformly smooth with respect to prompts but only non-uniformly smooth with respect to responses. Additionally, in some part of the paper, we assume that $r^*(x) := \max_{y \in \Omega_Y} r^\circ(x, y)$ belongs to the space $B^s_{(p,p),q}(\Omega_X)$. We denote this space by $\bar{B}^s_{p,q}(\Omega_X)$. Moreover, for any function space $\mathcal{F}$ equipped with a norm $\|\cdot\|_{\mathcal{F}}$, we define $U(\mathcal{F}) := \{f \in \mathcal{F} \mid \|f\|_{\mathcal{F}} \leq 1\}$.

# 3. Superiority of Neural Networks over Linear Estimators

In this section, we discuss the gap in capability between NNs and linear estimators in inference-time alignment. Specifically, we compare the two estimators in the sense of maximizing $\chi^2$-regularized objective defined as

$$\mathcal{L}_{\chi^2}(\pi; r)(x)$$
$$:= \mathbb{E}_{y \sim \pi(\cdot|x)}[r(x, y)] - \mu \cdot \chi^2(\pi(\cdot \mid x) \| \pi_{\text{ref}}(\cdot \mid x)), \quad (2)$$

where $\chi^2(q_1 \| q_0) := \frac{1}{2} \int_{\Omega_Y} \{(q_1(y)/q_0(y))^2 - 1\} q_0(y) \, dy$. As we mentioned in Section 2.1, the policy maximizing $\chi^2$-regularized objective achieves the optimal regret bound. Therefore, the value of the objective should be a good metric to compare the reward estimators.

For the true reward $r^\circ$ and its estimator $\hat{r}$, we consider the following value to measure how effective the estimator is for inference-time alignment

$$\overline{\mathcal{L}}(\hat{r}) := \mathbb{E}_{x \sim P_X}[\mathcal{L}(\hat{\pi}; r^\circ)(x)],$$
$$\text{where } \hat{\pi}(\cdot \mid x) := \arg\max_{\pi(\cdot|x)} \mathcal{L}(\pi; \hat{r})(x).$$

In other words, we consider the value of the regularized objective *corresponding to the true reward* for the policy optimized for the objective *corresponding to the estimator*. We show that the NN-based reward estimator $\hat{r}_{\text{NN}}$ is superior to linear reward estimator $\hat{r}_{\text{lin}}$ with respect to $\overline{\mathcal{L}}$.

## 3.1. Upper Bound for Neural Networks

To formally state the upper bound of $\overline{\mathcal{L}}(\hat{r}_{\text{NN}})$, we first introduce the class $\Phi(L, W, S, B)$ of neural networks.

**Class of Neural Networks.** To obtain the regret bound for neural network estimators of the reward, we formally define the class of neural networks used in this paper. Let $\eta := \max\{0, \cdot\}$ be the ReLU activation function. Then, a neural network with depth $L$ and width $W$ is defined as

$$f(x) = (A_L \eta(\cdot) + b_L) \circ \cdots \circ (A_2 \eta(\cdot) + b_2) \circ (A_1 x + b_1),$$

where $A_i \in \mathbb{R}^{d_{i+1} \times d_i}, b_i \in \mathbb{R}^{d_{i+1}}$ for $i \in [L]$ with $d_1 = d, d_{L+1} = 1$, and $\max_i d_i \leq W$. Then, we define the class $\Phi'(L, W, S, B)$ of neural networks with depth $L$, width $W$, sparsity $S$ and norm bound $B$ as

$$\Phi'(L, W, S, B) := \left\{ f \, \middle| \, \begin{array}{l} \max_i\{\|A_i\|_\infty, \|b_i\|_\infty\} \leq B, \\ \sum_{i=1}^L (\|A_i\|_0 + \|b_i\|_0) \leq S \end{array} \right\}.$$

where $\|\cdot\|_\infty$ is the maximum absolute value of the entries ($\ell^\infty$-norm as a vector) and $\|\cdot\|_0$ is the number of non-zero elements ($\ell^0$-norm as a vector). The $\ell^0$-norm constraint imposes sparsity of the model that controls the complexity of the model appropriately. Due to the technical convenience to analyze the estimation error, we consider the class of clipped neural networks defined as $\Phi(L, W, S, B) := \{\min\{\max\{f, -R\}, R\} \mid f \in \Phi'(L, W, S, B)\}$. Since the clipping function can be realized by ReLU units, this setting is not far from practical scenarios.

**Upper Bound of $\overline{\mathcal{L}}(\hat{r}_{\text{NN}})$.** We consider the NN-based reward estimator $\hat{r}_{\text{NN}}$ defined as a least-squares estimator in the class $\Phi(L, W, S, B)$, i.e.,

$$\hat{r}_{\text{NN}} := \arg\min_{\hat{r} \in \Phi(L, W, S, B)} \sum_{i=1}^n (r_i^\dagger - \hat{r}(x_i, y_i))^2, \quad (3)$$

where $x_1, \ldots, x_n \sim P_X$ are i.i.d. samples, $y_i \sim \pi_{\text{ref}}(\cdot \mid x_i)$ ($i \in [n]$), and $r_1^\dagger, \ldots, r_n^\dagger$ are the returns of oracles (1) for the pairs $(x_1, y_1), \ldots, (x_n, y_n)$. Then, we get the following upper bound.

**Theorem 3.** *Suppose that $L, W, S, B$ are set to $L \simeq L_0$, $W \simeq N$, $S \simeq NL_0$, $\log B \simeq 1$, where $N \simeq n^{\frac{d}{2s+d}}$, $L_0 \simeq \log(N) \log\log(N)$. Then, for any $r^\circ \in U(B^s_{p,q}(\Omega))$, it holds*

$$\overline{\mathcal{L}}(r^\circ) - \overline{\mathcal{L}}(\hat{r}_{\text{NN}}) \lesssim n^{-\frac{2s}{2s+d}} \log^2 n.$$

The proof can be found in Appendix B.2. This theorem states that the difference of the objective value between the NN-based estimator $\hat{r}_{\text{NN}}$ and the true reward $r^\circ$ vanishes as $n \to \infty$ with a convergence rate that matches the minimax optimal estimation error in nonparametric regression.

## 3.2. Lower Bound for Linear Estimators

Next, we introduce linear estimators and establish a lower bound on $\overline{\mathcal{L}}(\widehat{r}_{\text{lin}})$. A *linear estimator* is a class of estimators that can be written as

$$\widehat{r}_{\text{lin}}(x, y) = \sum_{i=1}^{n} r_i^{\dagger} \varphi_i(x, y; X^n, Y^n), \qquad (4)$$

where $X^n := (x_1, \ldots, x_n), Y^n := (y_1, \ldots, y_n)$, and $\varphi_i(\cdot; X^n, Y^n)$ are measurable functions that depend on $(x, y)$ and $X^n, Y^n$, but not on $r_1^{\dagger}, \ldots, r_n^{\dagger}$. This class includes a wide range of estimators such as $k$-NN regression, kernel ridge regression with a fixed kernel function, and sieve estimators. Linear estimators cannot capture nonlinear effects of the output and thus cannot perform nonlinear output-dependent feature learning (although they may learn features that depend only on the input, as in PCA). This difference induces the following sub-optimal rate.

**Theorem 4.** *Suppose that, for any true reward $r^{\circ} \in U(B_{p,q}^s(\Omega))$, the linear estimator defined in* (4) *satisfies*

*(a)* $\|\widehat{r}_{\text{lin}} - r^{\circ}\|_{\infty} \xrightarrow{\mathbb{P}} 0 \quad (n \to \infty)$,
*(b)* $\mathbb{P}_{D^n}\left[\|\widehat{r} - r^{\circ}\|_2^2 \gtrsim \varepsilon^2\right] \geq \delta$,
*(c)* $\mathbb{E}_{D^n}\left[\left\|\mathbb{E}_{y \sim \pi_{\text{ref}}(\cdot|x)}[\widehat{r}_{\text{lin}}(\cdot, y) - r^{\circ}(\cdot, y)]\right\|_{L^2}^2\right] = o_n(\varepsilon^2)$,

*for some constant $\delta \in (0, 1]$, where $D^n := \{(x_i, y_i, r_i^{\dagger})\}_{i=1}^n$ and $\varepsilon^2 := \mathbb{E}_{D^n}\left[\|\widehat{r}_{\text{lin}} - r^{\circ}\|_2^2\right]$. Then, there exists $r^{\circ} \in U(B_{p,q}^s(\Omega))$ such that*

$$\overline{\mathcal{L}}(r^{\circ}) - \overline{\mathcal{L}}(\widehat{r}_{\text{lin}}) \gtrsim n^{-\frac{2s-v}{2s+d-v}},$$

*for sufficiently large $n \in \mathbb{Z}_{>0}$, where $v := 2(d_Y/p - d/2)_+$.*

The proof can be found in Appendix B.3. This theorem shows that the gap of $\overline{\mathcal{L}}$ between $\widehat{r}_{\text{lin}}$ and $r^{\circ}$ does not vanish as quickly as that between $\widehat{r}_{\text{NN}}$ and $r^{\circ}$ shown in Theorem 3, in particular when $d_X < d_Y$ and $p \in [1, 2d_Y/d)$. This demonstrates the sub-optimality of linear estimators for inference-time alignment.

**Remark.** Condition (a) requires uniform consistency of the estimator; (b) assumes a form of stability with respect to $D^n$; and (c) assumes that the error is small when the reward is averaged over prompts. These assumptions are natural in standard settings for linear estimators. In particular, (a) is known to hold for many types of linear estimators including $k$-NN, Nadaraya–Watson, and kernel ridge estimators. Moreover, we can check that (b) holds in a common situation for $k$-NN, for example. See Appendix B.4 for details.

## 4. Single-step Algorithm and its Analysis

The ultimate goal of inference-time alignment is to find a policy $\pi$ that achieves small *regret* $\mathcal{R}$ defined as

$$\mathcal{R} := \mathbb{E}_{x \sim P_X}[r^*(x) - \mathbb{E}_{y \sim \pi(\cdot|x)} r^{\circ}(x, y)],$$

where $r^*(x) := \max_{y \in \Omega_Y}[r^{\circ}(x, y)]$. This quantifies the expected gap between the optimal reward and the reward obtained by sampling from $\pi$.

In this section, as a first step of regret analysis, we present a simple method for learning an NN-based reward model and derive a regret bound for the resulting policy.

### 4.1. Algorithm

Here, we present how to generate the responses with higher reward through the reward estimation. First, we generate $n$ input prompts $x_1, \ldots, x_n$ i.i.d. from $P_X$, and for each $i \in [n]$, we generate the responses $y_i \sim \pi_{\text{ref}}(\cdot \mid x_i)$ from our pretrained reference model. We then observe noisy reward oracle outputs as $r_i^{\dagger} := r^{\circ}(x_i, y_i) + \xi_i$ as in (1), where $\xi_i \sim \mathcal{N}(0, \sigma^2)$ is the observation noise. We fit the neural network $\widehat{r}_{\text{NN}}$ via empirical risk minimization in (3), where $L, W, S, B$ will be set appropriately depending on the smoothness of the true reward and the data size. Using the estimated reward $\widehat{r}$, we update the generative model in accordance to the reward. For $\mu > 0$, we define $\pi_{\mu}^{\chi}$ by

$$\pi_{\mu}^{\chi}(\cdot \mid x) := \underset{\pi(\cdot|x):\text{density on } \Omega_Y}{\arg\max} \mathcal{L}_{\chi^2}(\pi; \widehat{r})(x),$$

where $\mathcal{L}_{\chi^2}$ is the $\chi^2$-regularized objective defined in (2)[1]. Inference-time Pessimism (Huang et al., 2025b) is a practical algorithm for obtaining samples from $\pi_{\mu,N}^{\text{Pes}}$, which approximates $\pi_{\mu}^{\chi}$, where $N \in \mathbb{Z}_{>0}$ determines the number of samples drawn from $\pi_{\text{ref}}(\cdot \mid x)$.

We then generate a response $\widehat{y}_{\text{NN}}(x)$ for a prompt $x$ by sampling $\widehat{y}_{\text{NN}}(x) \sim \pi_{\mu,N}^{\text{Pes}}(\cdot \mid x)$.

### 4.2. Theoretical Analysis

For the analysis, we put the following assumption.

**Assumption 5.** *For a true reward function $r^{\circ} \in B_{p,q}^s(\Omega)$ $(p, q \in [1, \infty], s > d_Y/p)$, we define $r^*(x) := \max_{y \in \Omega_Y} r^{\circ}(x, y)$ and $S_{\epsilon}(x) := \{y \mid r^*(x) - r^{\circ}(x, y) \leq \epsilon\}$. Let $\gamma \in [0, \frac{d}{s-d/p})$, $c > 0$ and $\epsilon_0 > 0$ be constants. We assume that $\lambda(S_{\epsilon}(x)) \geq c\epsilon^{\gamma}$ for all $\epsilon \in (0, \epsilon_0]$ and $x \in \Omega_X$. Moreover, we assume that $\|r^{\circ}\|_{\infty} \leq R$ for some constant $R > 0$.*

**Remark.** The condition $s > d/p$ ensures that $r^{\circ}$ is continuous, which is necessary to obtain a meaningful regret bound. The condition $\lambda(S_{\epsilon}(x)) \gtrsim \epsilon^{\gamma}$ is a regularity assumption on the near-optimal set around an optimal point $y^*(x)$. It guarantees that sampling from the reference policy $\pi_{\text{ref}}$ has a non-negligible chance to produce responses close to optimal ones.

---

[1]Huang et al. (2025c) showed that $\chi^2$-divergence provides a more robust policy against over-optimization and yields lower regret than the usual KL-divergence regularization. Hence, we also employ $\chi^2$-divergence in this paper.

We obtain the following regret bound.

**Theorem 6.** *Suppose that we set the parameters of the network as $L = O(\log N_0), W = O(N_0 \log N_0), S = O(N_0 \log^2 N_0), \log B = O(\log N_0)$ for $N_0 = n^{\frac{1}{2s+d}}$ sufficiently large. Under Assumption 5, the policy $\pi_{\mu,N}^{\mathrm{Pes}}$ satisfies*

$$\mathbb{E}_{D^n}[\mathbb{E}_{x \sim P_X}[\mathbb{E}_{\widehat{y}_{\mathrm{NN}}(x) \sim \pi_{\mu,N}^{\mathrm{Pes}}}[r^*(x) - r^\circ(x, \widehat{y}_{\mathrm{NN}}(x))]]]$$
$$\lesssim n^{-\frac{s}{2s+d} \cdot \frac{2}{2+\gamma}} \log^{\frac{4}{2+\gamma}} n.$$

It is known that $n^{-\frac{s}{2s+d}}$ is the minimax optimal rate (Donoho & Johnstone, 1998; Giné & Nickl, 2015) in terms of $L^2$-norm to estimate a function in the Besov space. The regret bound is slower than this rate up to the factor $\frac{2}{2+\gamma}$. This difference is a cost to convey the $L^2$-norm error into an $L^\infty$-type bound locally around a maximizer using the volume condition on the upper-level set (Assumption 5).

**Proof sketch**   The key of the proof of this theorem is the regret bound given by Huang et al. (2025b) that characterizes the balance between the reward estimation error and the *coverage* of the reference policy. Let

$$\mathcal{E}^2(x) := \mathbb{E}_{y \sim \pi_{\mathrm{ref}}(\cdot|x)}[(\widehat{r}(x, y) - r^\circ(x, y))^2],$$

be the $L^2$-estimation error of our reward estimator $\widehat{r}$. For two policies $\pi_1, \pi_2$, we define the coverage as

$$\mathcal{C}(x; \pi_1, \pi_2) := \mathbb{E}_{y \sim \pi_1(\cdot|x)}\left[\frac{\pi_1(y|x)}{\pi_2(y|x)}\right].$$

**Proposition 7** (Huang et al. (2025b)). *For $\mu > 0$ and $N \in \mathbb{Z}_{>0}$, let $\pi_{\mu,N}^{\mathrm{Pes}}(\cdot \mid x)$ be the output of Inference-Time Pessimism. Then, for any comparator policy $\pi^*$, there exists a constant $c > 0$ such that*

$$\mathbb{E}_{y \sim \pi^*(\cdot|x)}[r^\circ(x, y)] - \mathbb{E}_{y \sim \pi_{\mu,N}^{\mathrm{Pes}}(\cdot|x)}[r^\circ(x, y)]$$
$$\lesssim \mu \cdot \mathcal{C}(x; \pi^*, \pi_{\mathrm{ref}}) + \mu^{-1} \cdot \mathcal{E}^2(x) + \mu^{-1} \mathcal{E}(x) e^{-\frac{\mu N}{c(R+\mu)}}.$$

From this relationship, we see a trade-off between the estimation error $\mathcal{E}^2$ and the coverage. To achieve a smaller regret, the reference policy $\pi_{\mathrm{ref}}$ should "cover" a region around the maximum reward point and the reward function should be estimated accurately.

We obtain an upper bound on the estimation error $\overline{\mathcal{E}}^2 := \mathbb{E}_{x \sim P_X}[\mathcal{E}(x)^2]$, namely, $\mathbb{E}_{D^n}[\overline{\mathcal{E}}^2] \lesssim n^{-\frac{2s}{2s+d}} \log^2 n$. This convergence rate matches that in Suzuki (2019) for non-parametric regression of functions in Besov spaces. Moreover, we can prove the following lemma, which demonstrates that we can construct a comparator policy $\pi^*$ that attains both small regret and coverage $\mathcal{C}(x; \pi^*, \pi_{\mathrm{ref}})$.

**Lemma 8.** *Suppose that $r^\circ \in B_{p,q}^s(\Omega)$ satisfies Assumption 5. Then, for any $\epsilon \in (0, \epsilon_0]$, there exists a comparator policy $\pi_\epsilon^*$ satisfying (i) $r^*(x) - \mathbb{E}_{y \sim \pi_\epsilon^*(\cdot|x)}[r^\circ(x, y)] \le \epsilon$, and (ii) $\mathcal{C}(x; \pi_\epsilon^*, \pi_{\mathrm{ref}}) \le \epsilon^{-\gamma}$ for all $x \in \Omega_X$.*

---

**Algorithm 1** Multi-step Training for the Reward Model

---

    **Input:** Base policy $\pi_{\mathrm{ref}}$, size of oracle queries $n$.
1: Set $\pi^{(0)} := \pi_{\mathrm{ref}}, T := \lceil \log n \rceil, n_0 := \lfloor n/T \rfloor$.
2: **for** $\tau = 1, \ldots, T$ **do**
3:     Draw $n_0$ samples $\{x_t\}_{t=(\tau-1)n_0+1}^{\tau n_0} \sim P_X$.
4:     Draw the responses $\{y_t\}_{t=(\tau-1)n_0+1}^{\tau n_0}$ with
       $y_t \sim \pi^{(\tau-1)}(\cdot \mid x_t)$.
5:     Get the set of indices with valid responses:
       $\mathcal{T}_\tau := \{t \mid (\tau-1)n_0 + 1 \le t \le \tau n_0, y_t \in \Omega_Y\}$.
6:     Observe the true rewards $\{r_t^\dagger\}_{t \in \mathcal{T}_\tau}$:
       $r_t^\dagger := r^\circ(x_t, y_t) + \xi_t \quad (t \in \mathcal{T}_\tau)$.
7:     Train the reward model $\widehat{r}^{(\tau)} \in \Phi(L, W, S, B)$:
       $\widehat{r}^{(\tau)} := \arg\min_r \sum_{t \in \mathcal{T}_\tau}(r(x_t, y_t) - r_t^\dagger)^2$.
8:     Get the smoothed reward model $\widetilde{r}^{(\tau)} := K * \widehat{r}^{(\tau)}$.
9:     Set $\pi_{\mathrm{trnc}}^{(\tau-1)} := \pi^{(\tau-1)} \mid \Omega_Y^{[\eta/2]}$.
10:    Define the policy $\pi_{\mathrm{Pes}}^{(\tau)}$ by Inference-time Pessimism
      with base policy $\pi_{\mathrm{trnc}}^{(\tau-1)}$ and reward model $\widetilde{r}^{(\tau)}$.
11:    Set $\pi_G^{(\tau)}(\cdot \mid x) := \pi_{\mathrm{Pes}}^{(\tau)}(\cdot \mid x) * \mathcal{N}(0, (\sigma^{(\tau)})^2 I_{d_Y})$.
12:    Define the new policy $\pi^{(\tau)}$ as
      $\pi^{(\tau)}(\cdot \mid x) := \kappa^{(\tau)} \mathrm{Unif}(\Omega_Y) + (1 - \kappa^{(\tau)})\pi_G^{(\tau)}(\cdot \mid x)$.
13: **end for**
14: **return:** policy $\pi_{\mathrm{Pes}}^{(T)}$.

---

Using the comparator policy $\pi_\epsilon^*$, the regret can be upper-bounded as follows:

$$\mathbb{E}_{D^n}[\mathbb{E}_{x \sim P_X}[\mathbb{E}_{\widehat{y}_{\mathrm{NN}}(x) \sim \pi_{\mu,N}^{\mathrm{Pes}}}[r^*(x) - r^\circ(x, \widehat{y}_{\mathrm{NN}}(x))]]]$$
$$\lesssim \mathbb{E}_{x \sim P_X}\left[r^*(x) - \mathbb{E}_{y \sim \pi_\epsilon^*(\cdot|x)}[r^\circ(x, y)]\right]$$
$$+ \mathbb{E}_{D^n, x}\left[\mathbb{E}_{y \sim \pi_\epsilon^*(\cdot|x)}[r^\circ(x, y)] - \mathbb{E}_{y \sim \pi_{\mu,N}^{\mathrm{Pes}}(\cdot|x)}[r^\circ(x, y)]\right].$$

The first and second terms can be bounded using Lemma 8 and Proposition 7, respectively. Choosing $\mu, \epsilon > 0$ to balance these two terms completes the proof sketch. The full proof can be found in Appendix C.

## 5. Multi-step Algorithm for Improved Regret

In this section, we propose a multi-step algorithm for training a neural-network-based reward estimator and show that it attains a better regret bound than a single-step approach.

We begin by introducing the following assumptions, which underpin our algorithm.

**Assumption 9.** *For a true reward function $r^\circ \in B_{p,q}^s(\Omega)$ $(p, q \in [1, \infty], s > d_Y/p)$, we assume that $\|r^\circ\|_\infty \le R$ for some constant $R > 0$. Moreover, we define $y^*(x) := \arg\max_{y \in \Omega_Y} r^\circ(x, y)$, $S_\epsilon(x)$ as in Assumption 5, and $\overline{S}_\epsilon := \{(x, y) \mid r^*(x) - r^\circ(x, y) \le \epsilon\}$. We assume the following conditions:*

*(A1) There exist constants $\overline{c}, \underline{c}, \epsilon_0 > 0$ and $\beta, \gamma \in \left[1 + \frac{d}{s \min\{2, p\}}, \frac{d}{s - d/p}\right]$ with $\beta \le \gamma$ such that*

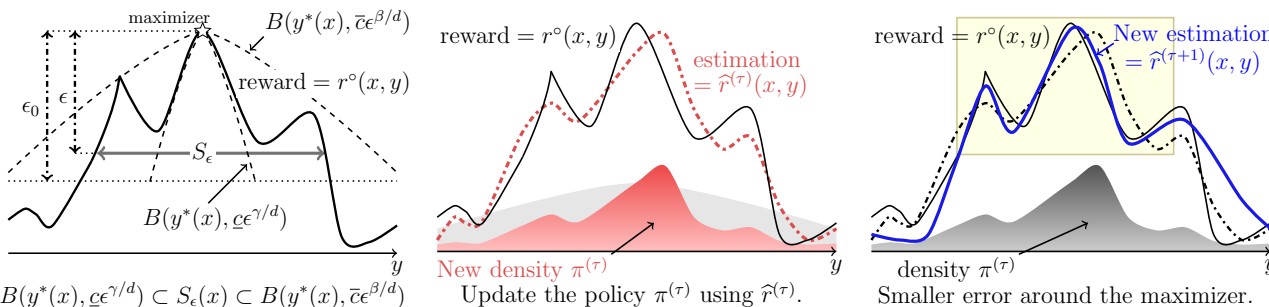

$B(y^*(x), \underline{c}\epsilon^{\gamma/d}) \subset S_\epsilon(x) \subset B(y^*(x), \overline{c}\epsilon^{\beta/d})$. Update the policy $\pi^{(\tau)}$ using $\widehat{r}^{(\tau)}$. Smaller error around the maximizer.

*Figure 1.* Conceptual illustrations of our assumptions and algorithm. (Left) In Assumption 9, we impose assumptions on the local landscape of the reward around the maximum point. Specifically, we assume that for all $\epsilon \in (0, \epsilon_0]$, the super-level set $S_\epsilon(x)$ satisfies $B(y^*(x), \underline{c}\epsilon^{\gamma/d}) \subseteq S_\epsilon(x) \subseteq B(y^*(x), \overline{c}\epsilon^{\beta/d})$. We remark that, when $p$ is small, our assumptions allow locally bumpy shapes of the reward function (as in the figure), since the Besov space $B_{p,q}^s$ includes such functions. (Middle, Right) Our multi-step algorithm (Algorithm 1) mainly consists of two procedures: first, we estimate the true reward $r^\circ$ by $\widehat{r}^{(\tau)}$ trained using samples from $\pi^{(\tau-1)}$. Then, we obtain the updated policy $\pi^{(\tau)}$, which prioritizes responses $y$ with high estimated rewards. Hence, in the next step $\tau + 1$, reward estimation becomes more accurate near the maximizer, which results in a higher expected reward for responses generated from $\pi^{(\tau+1)}$.

$$B_2(y^*(x), \underline{c}\epsilon^{\gamma/d}) \subseteq S_\epsilon(x) \subseteq B_2(y^*(x), \overline{c}\epsilon^{\beta/d}).$$

**(A2)** *There exist constants* $c_0, C_0 > 0$ *such that* $\mathcal{M}_2(\delta, \overline{S}_\epsilon) \leq C_0\big(1 + \delta^{-d}\lambda(\overline{S}_\epsilon)\big)$ *for all* $\epsilon, \delta \in (0, c_0]$.

**(A3)** *It holds* $r^* \in \overline{B}_{p,q}^s(\Omega_X)$.

**(A4)** *It holds* $y^*(x) \in \Omega_Y^{[\eta]}$ *for some constant* $\eta > 0$.

**Remark.** **(A1)** requires the super-level set to be concentrated around the maximizer, with both lower and upper bounds on the distance from the maximizer. The lower bound on this distance is necessary for the algorithm to capture the rough location of the super-level set, whereas the upper bound is needed to narrow down the location of $y^*(x)$ within it. A simple example is the case where $r^\circ(x, \cdot)$ is locally strongly convex around $y^*(x)$; in this case, the assumption holds with $\beta = \gamma = d/2$. See also the middle and left part of Figure 1. **(A2)** is adopted from Wang et al. (2018), which studies the oracle complexity of optimizing Hölder-smooth functions. This imposes a regularity condition on the set $\overline{S}_\epsilon$. This assumption is satisfied, for example, when $\overline{S}_\epsilon$ is a finite union of $\|\cdot\|_2$-balls. **(A3)** is a technical assumption to guarantee that the NNs can learn to roughly determine whether a response $y$ achieves a high reward or not. **(A4)** is also a technical one that eliminates the situation where the optimal response is too close to the boundary, which would make it difficult to collect information around it.

Assumption 9 imposes a stronger assumption than Assumption 5, which was used in the analysis of the single-step algorithm. However, we do not believe that imposing Assumption 9 on the single-step algorithm would improve its regret. See Appendix F for more details.

### 5.1. Algorithm

Our multi-step algorithm is presented in Algorithm 1. We also provide an illustrative explanation in the right part of Figure 1. Basically, it repeats the alignment method in the previous section multiple times; in other words, we alternately update the NN-based reward estimator $\widehat{r}^{(\tau)}$ (line 7) and the policy $\pi^{(\tau)}$ using Inference-time Pessimism (line 10). As we have seen in Theorem 6, the responses $y$ generated from $\pi^{(\tau)}$ are expected to achieve smaller regret, i.e., "concentrate" around the maximum reward point $y^*(x)$ compared to those from the base policy $\pi^{(\tau-1)}$ (the policy in the previous step). Then, the new reward estimator $\widehat{r}^{(\tau)}$ is trained to minimize the error for the samples $(x, y)$ generated by the new policy $\pi^{(\tau)}$. Since the neural network $\widehat{r}^{(\tau)}$ can "neglect" estimating the true rewards at the points far from $y^*(x)$, it is expected to achieve a smaller estimation error near the maximizer. Moreover, since $\pi^{(\tau)}$ concentrates around the optimal response $y^*(x)$, the policy is expected to have smaller coverage.

Based on the observation above, a natural first idea is to define the updated policy $\pi^{(\tau)}$ as follows:

$$\pi^{(\tau)}(\cdot \mid x) := \pi_{\mathrm{Pes}}^{(\tau)}(\cdot \mid x) \approx \arg\max_{\pi(\cdot \mid x)} \mathcal{L}_{\chi^2}(\pi^{(\tau-1)}; \widehat{r}^{(\tau)})(x),$$

where $\pi_{\mathrm{Pes}}^{(\tau)}$ is the policy defined by Inference-time Pessimism with appropriate $\mu > 0, \mathbb{Z}_{>0}$. However, updating the policy in this way causes the multi-step algorithm to fail. This is because we only have $L^2$-norm guarantee for our reward estimate $\widehat{r}^{(\tau)}$, which is insufficient to bound the coverage $\mathcal{C}(x; \pi_\epsilon^*, \pi^{(\tau)})$.

To avoid this, we employ two types of "mollification". First, in line 8, we mollify the reward estimator $\widehat{r}^{(\tau)}$ using

the kernel defined as

$$K(x) := \sum_{j=1}^{r} \binom{r}{j}(-1)^{1-j} \frac{1}{j^d} \left(\frac{2}{\gamma^2\pi}\right)^{d/2} K_{\frac{j\gamma}{\sqrt{2}}}(x),$$

where $K_\gamma(x) := \exp\left(-\gamma^{-2}\|x\|_2^2\right)$, and obtained the smoothed reward model $\widetilde{r}^{(\tau)} := K * \widehat{r}^{(\tau)}$, where

$$K * f := \int_{\mathbb{R}^d} \overline{f}(z) K(x-z) \, \mathrm{d}z, \quad \overline{f}(x) := \begin{cases} f(x) & (x \in \Omega), \\ 0 & (x \notin \Omega). \end{cases}$$

This enables us to obtain a prompt-uniform estimation error from $L^2$-norm guarantee on the reward model. Moreover, in line 11, we mollify the density of the policy $\pi_{\mathrm{Pes}}^{(\tau)}$ obtained via Inference-time Pessimism by adding Gaussian noise $\mathcal{N}(0, (\sigma^{(\tau)})^2 I_{d_Y})$ to each generated point. This guarantees that the distribution of generated points covers the maximum reward point with non-vanishing probability. The noise scale $\sigma^{(\tau)}$ is set to trade off exploitation of the current reward model against sufficient coverage: if $\sigma^{(\tau)}$ is too small, sampling becomes overly concentrated around potentially biased maximizers and misses regions of high true reward; if it is too large, samples become nearly independent of the learned model and fail to leverage the information gathered so far. For details on the hyperparameter configuration, including $\sigma^{(\tau)}$, please refer to Appendix E.

### 5.2. Theoretical Guarantee

We prove the following theorem, which provides a regret bound for Algorithm 1.

**Theorem 10.** *Suppose that $r^\circ \in U(B_{p,q}^s)$ satisfies Assumption 9. For any $\upsilon \in (0,1)$, the output $\pi_{\mathrm{Pes}}^{(T)}$ of Algorithm 1 satisfies*

$$\mathbb{E}_{x \sim P_X|\Omega_X^{[\eta]}}[r^*(x) - \mathbb{E}_{y \sim \pi_{\mathrm{Pes}}^{(T)}}[r^\circ(x,y)]]$$

$$\lesssim n^{-\frac{1}{2+\gamma}\frac{2s}{2s+d}\frac{1+u'}{1-u}} \operatorname{poly}\log(n),$$

*with probability $1-\upsilon$, where $u := \frac{2\beta\varsigma}{2s+d}\frac{\beta}{\gamma}\frac{2s'}{2s'+d}\frac{2}{2+\gamma}$, $u' := \frac{2s'}{2s'+d}\frac{\beta}{2+\gamma}$, $\varsigma \in (0, s-d/p)$ and $s' := s\min\{1, p/2\}$.*

The proof is given in Appendix E. The regret bound for the multi-step algorithm in this theorem is better than that of the single-step algorithm in Theorem 6 by a factor of $\frac{1+u'}{1-u}(> 1)$. This highlights *the efficacy of the multi-step algorithm compared with the single-step algorithm*. Moreover, the factor $\frac{1+u'}{1-u}$ depends on $\beta$, which quantifies how small the super-level set of the reward $r^\circ(x,\cdot)$ is: larger $\beta$ yields a better regret rate. This suggests that *multi-step training enables the NN-based reward estimator to capture local structure of true reward around the maximizer*.

**Proof sketch** As mentioned earlier, the advantage of the multi-step algorithm is that iteratively updating the policy and the reward estimator makes the policy concentrate around the optimal response. This leads to (I) *smaller*

*reward-estimation error* and (II) *smaller coverage*. The first point is supported by the following lemma.

**Lemma 11** (Estimation Error under Small Regret). *Assume that $r^\circ \in U(B_{p,q}^s)$ satisfies Assumption 9. Moreover, suppose that $\pi$ is a policy satisfying $\mathbb{E}_{x \sim P_X}[r^*(x) - \mathbb{E}_{y \sim \pi(\cdot|x)}[r^\circ(x,y)]] \leq \mathcal{R}$. Let $D^n = \{(x_i, y_i, r_i^\dagger)\}_{i=1}^n$ be a dataset where $x_i \sim P_X$, $y_i \sim \pi(\cdot \mid x_i)$, and $r_i^\dagger = r^\circ(x_i, y_i) + \xi_i$ with $\xi_i \sim \mathcal{N}(0, \sigma^2)$. Then, the estimator $\widehat{r}$ defined as $\widehat{r} := \arg\min_{r \in \Phi(L,W,S,B)} \sum_{i=1}^n (r(x_i, y_i) - r_i^\dagger)^2$ for appropriate $L, W, S, B$ satisfies*

$$\mathbb{E}_{D^n}\left[\|\widehat{r} - r^\circ\|_{L^2(P_X \otimes \pi)}^2\right] \lesssim \mathcal{R}^{\frac{2\beta\varsigma}{2s+d}} \cdot n^{-\frac{2s}{2s+d}} \log^4(n),$$

*where $(P_X \otimes \pi)(\mathrm{d}x, \mathrm{d}y) := P_X(\mathrm{d}x)\pi(\mathrm{d}y \mid x)$.*

When $\beta = 0$, the convergence rate matches the existing results on regression by NNs for Besov space (Suzuki, 2019) (up to log-factors). As $\beta$ increases and the super-level set of $r^\circ$ becomes smaller, the estimation-error rate improves.

Moreover, the following lemma addresses the second point, i.e., improved coverage.

**Lemma 12.** *Suppose that $r^\circ \in U(B_{p,q}^s)$ satisfies Assumption 9. Moreover, let $\widehat{\pi}$ be a policy satisfying $\sup_{x \in \Omega_X} \mathbb{E}_{y \sim \widehat{\pi}(\cdot|x)}[r^*(x) - r^\circ(x,y)] \leq \mathcal{R}$ for some $\mathcal{R} \in (0, 1/2)$, and define the policy $\pi_G$ as $\pi_G(\cdot \mid x) = \widehat{\pi}(\cdot \mid x) * \mathcal{N}(0, \sigma^2)$ for an appropriate $\sigma$. Then, for any $\epsilon \in (0, \mathcal{R})$, there exists a comparator policy $\pi_\epsilon^*$ satisfying (i) $r^*(x) - \mathbb{E}_{y \sim \pi_\epsilon^*(\cdot|x)}[r^\circ(x,y)] \leq \epsilon$; and (ii) $\mathcal{C}(x; \pi_\epsilon^*, \pi_G) \leq \epsilon^{-\gamma}\mathcal{R}^\beta$ for all $x \in \Omega_X$.*

Both lemmas show that *each step in the algorithm benefits from the small regret $\mathcal{R}$ of the previous step*, through small estimation error and coverage. Using these two lemmas, we prove that, during multi-step training, each step improves both the regret $\mathcal{R}$ and the reward-estimation rate. Consequently, after $O(\log n)$ steps, we achieve the rate stated in Theorem 10.

## 6. Conclusion

This paper analyzes the training of reward estimators for inference-time alignment. We consider a setting in which the true reward lies in a Besov space, whose functions can exhibit non-uniform smoothness over the input space. Due to the feature learning ability of NNs, we show that, when the uniformity of the smoothness $p$ is small, NNs can outperform approaches based on linear estimators in terms of the regularized objective value. In addition, we propose a multi-step update method for inference-time alignment and analyze its regret bound. Under the assumption that the super-level set of the reward is concentrated around the maximizer, we show that the multi-step method improves regret compared to a single-step algorithm by progressively refining the estimated location of the reward maximizer.

**Limitation and future work.** In this paper, we analyzed a simplified setting of inference-time alignment where prompts and responses are represented as vectors, which enables us to isolate the role of NNs' feature learning. A key limitation is the gap to realistic settings; extending the analysis to token-sequence function classes and developing Transformer-based inference-time alignment algorithms are natural future directions.

We also assumed that the true reward belongs to a Besov space, a broad function class allowing spatially non-uniform smoothness. While this generality is one of the main contributions of the paper, it requires additional technical tools such as smoothing and Gaussian mollification, and introduces practical limitations including algorithmic complexity and additional hyperparameters. Addressing these issues is an important direction for future work.

Furthermore, we analyzed oracle complexity for training the reward model without optimizing computational costs. Reducing both training and inference costs while preserving the regret gains of our multi-step algorithms is an additional direction for future work.

## Acknowledgements

NN was partially supported by JSPS KAKENHI (25H01107), JST ACT-X (JPMJAX24CK) and JST BOOST (JPMJBS2418). TS was partially supported by JSPS KAKENHI (24K02905) and JST CREST (PMJCR2015). This research is supported by the National Research Foundation, Singapore and the Ministry of Digital Development and Information under the AI Visiting Professorship Programme (award number AIVP-2024-004). Any opinions, findings and conclusions or recommendations expressed in this material are those of the author(s) and do not reflect the views of National Research Foundation, Singapore and the Ministry of Digital Development and Information.

## Impact Statement

This paper presents work whose goal is to advance the field of Machine Learning. There are many potential societal consequences of our work, none of which we feel must be specifically highlighted here.

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

# A. Estimating Functions in Besov Spaces with Anisotropic Integrability

In this section, we analyze the estimation error for Besov Spaces with Anisotropic Integrability. Specifically, we show that (i) we can recover the approximation and estimation error bound established in Suzuki (2019) for the Besov spaces with anisotropic integrability; and (ii) we have an improved estimation error rate when we can access the policy with small regret.

We introduce some additional notations. By the definition of $r$-th difference of $f$, we have

$$\Delta_h^r(f)(x) := \begin{cases} \sum_{j=0}^{r} \binom{r}{j}(-1)^{r-j} f(x+jh) & (x \in \Omega, x+rh \in \Omega) \\ 0 & \text{(otherwise)}. \end{cases}$$

The modulus of smoothness corresponding to $B_{p,q}^s = B_{(\infty,p),q}^s$ is defined as

$$w_{r,p}(f,t) := \sup_{\|h\|_2 \le t} \left\{ \int_{\Omega_Y} \left( \sup_{x \in \Omega_X} |\Delta_h^r(f)(x,y)| \right)^p \mathrm{d}y \right\}^{1/p}.$$

Let $\mathcal{K} : \mathbb{R} \to \mathbb{R}$ and $\mathcal{K}_m : \mathbb{R} \to \mathbb{R}$ be functions defined as

$$\mathcal{K}(x) = \begin{cases} 1 & \text{if } x \in [0,1], \\ 0 & \text{otherwise}, \end{cases}$$

$$\mathcal{K}_m(x) = (\underbrace{\mathcal{K} * \cdots * \mathcal{K}}_{m+1 \text{ times}})(x),$$

where $f * g(x) := \int f(x-t)g(t)\mathrm{d}t$ is the convolution of functions $f$ and $g$. The function $\mathcal{K}_m$ is called the *cardinal B-spline of order* $m$. Then, for $k \in \mathbb{N}_{>0}$ and $\boldsymbol{j} = (j_1, \ldots, j_d) \in \mathbb{Z}^d$, let $M_{k,\boldsymbol{j}}^d : \mathbb{R}^d \to \mathbb{R}$ be the function defined as

$$M_{k,\boldsymbol{j}}^d(x) := \prod_{i=1}^{d} \mathcal{K}_m(2^k x_i - j_i),$$

Intuitively, the integer $k$ controls the spatial resolution, and $\boldsymbol{j}$ controls the location of the function. We remark that the support of $M_{k,\boldsymbol{j}}^d$ is the hyper-rectangle written by

$$\text{supp}(M_{k,\boldsymbol{j}}^d) = \prod_{i=1}^{d} \left[ 2^{-k} j_i, 2^{-k}(j_i + m + 1) \right]. \tag{5}$$

## A.1. Approximating the Function by Splines

In this section, we approximate a function $f \in B_{p,q}^s$ using a sum of $M_{k,\boldsymbol{j}}^d$ ($\boldsymbol{j} = (j_1, \ldots, j_d) \in \mathbb{Z}^d$) for some $k$.

We fix the order $m \in \mathbb{N}$ of the cardinal B-spline bases and $r \in \mathbb{Z}$ with $r > s$. We define the set $\mathcal{J}_k \subset \mathbb{Z}^d$ of tuples and the set $\mathcal{D}_k \subset \mathbb{R}^d$ of cubes as

$$\mathcal{J}_k := \{-m, -m+1, \ldots, 2^k - 1\}^d,$$

$$\mathcal{D}_k := \{D_{\boldsymbol{j}} \mid (j_1, \ldots, j_d) \in \mathcal{J}_k\}, \quad \text{where} \quad D_{\boldsymbol{j}} := \left[ \frac{j_1}{2^k}, \frac{j_1+1}{2^k} \right) \times \cdots \times \left[ \frac{j_d}{2^k}, \frac{j_d+1}{2^k} \right).$$

Importantly, for any $\boldsymbol{j} \in \mathcal{J}_k$, it holds $\text{supp}(M_{k,\boldsymbol{j}}) \subset \Omega$, there exists $x \in \text{supp}(M_{k,\boldsymbol{j}}) \cap \Omega$ such that $M_{k,\boldsymbol{j}}(x) \ne 0$. Conversely, for any $\boldsymbol{j} \notin \mathcal{J}_k$, we have $M_{k,\boldsymbol{j}}(x) = 0$ for all $x \in \text{supp}(M_{k,\boldsymbol{j}}) \cap \Omega$.

For a cube $D \in \mathcal{D}_k$, we choose a polynomial $P_D$ that is a near-best $L^{(\infty,p)}(D)$-approximation to $f$ among polynomials of coordinate degree less than $r$, i.e., let $\mathcal{P}_r$ is a set of polynomials defined as

$$\mathcal{P}_r := \left\{ \sum_{i=1}^{I} a_i x_1^{l_{i,1}} \cdots x_d^{l_{i,d}} \;\middle|\; I \in \mathbb{Z}_{\ge 0}, a_i \in \mathbb{R}^d, l_{i,j} \in \mathbb{Z}_{\ge 0}, \max_{i,j} l_{i,j} < r \right\},$$

and $P_D(f) \in \mathcal{P}_r$ is a polynomial satisfying

$$\|P_D(f) - f\|_{L^{(\infty,p)}(D)} \le A \cdot \inf_{P \in \mathcal{P}_r} \|P - f\|_{L^{(\infty,p)}(D)},$$

for some constant $A \ge 1$. Below, we define the local error in $D$ of approximation by polynomials in $\mathcal{P}$ by

$$E_r(f, D) := \inf_{P \in \mathcal{P}_r} \|P - f\|_{L^{(\infty,p)}(D)}.$$

Then, we present the following lemma on $P_D$, which shows that $P_D$ is also a near-best approximation on larger cubes.

**Lemma 13.** *Let $D \in \mathcal{D}_k$, and $D'$ be a cube satisfying $D' \supset D$ and $\lambda(D') \le a\lambda(D)$ with $a \ge 1$. Then, it holds*

$$\|f - P_D\|_{L^{(\infty,p)}(D')} \le c_{r,d,a,A} E_r(f, D'),$$

*where $c_{r,d,a,A}$ is a constant depending at most on $r, d, a$ and $A$.*

*Proof.* See the proof of Lemma 3.3 in DeVore & Popov (1988). □

Additionally, we present the following lemma, which states that $E_r(f, D)$ can be bounded by the modulus of smoothness of $f$. The proof can be done in the same way as Dung & Ullrich (2011).

**Lemma 14.** *Let $f \in L^{(p,\infty)}(\Omega)$ and $D \subseteq \Omega$ be a cube with edge length $l$. Then, it holds*

$$E_r(f, D) \le c_{r,d} w_{r,p}(f, l),$$

*where $c_{r,d}$ is a constant that depends only on $r$ and $p$.*

Next, we define the space $\Pi_{k,r}$ ($=: \Pi_k$ simply, if $r$ is clear from the context) of piecewise polynomials as

$$\Pi_{k,r} := \left\{ \sum_{D \in \mathcal{D}_k} \mathbb{1}_{x \in D}(x) Q_D(x) \,\middle|\, Q_D \in \mathcal{P}_r \right\}.$$

Moreover, let $\Sigma_k$ denote the linear span of the splines $M_{k,\boldsymbol{j}}$ ($\boldsymbol{j} \in \mathcal{J}_k$):

$$\Sigma_k := \left\{ \sum_{\boldsymbol{j} \in \mathcal{J}_k} c_{\boldsymbol{j}} M_{k,\boldsymbol{j}} \,\middle|\, c_{\boldsymbol{j}} \in \mathbb{R} \right\}.$$

Then, we have

$$\Sigma_k = \begin{cases} C^{r-2}(\Omega) \cap \Pi_{k,r} & (r > 1), \\ \Pi_{k,r} & (r = 1). \end{cases}$$

Let $S_k : L^{(\infty,p)}(\Omega) \to \Pi_{k,r}$ be an operator to give a near-best approximation to a piecewise polynomial function defined as

$$S_k(f)(x) = \sum_{D \in \mathcal{D}_k} \mathbb{1}_{x \in D}(x) P_D(f)(x).$$

Moreover, let $Q_k : \Pi_{k,r} \to \Sigma_k$ be an operator that maps piecewise polynomial to the set of splines defined as

$$Q_k(P) = \sum_{\boldsymbol{j} \in \mathcal{J}_k} \alpha_{k,\boldsymbol{j}}(P) M_{k,\boldsymbol{j}},$$

where the functional $\alpha_{k,\boldsymbol{j}}$ (or simply $\alpha_{\boldsymbol{j}}$ if $k$ is clear from the context) can be defined by extending the one defined on $\Sigma_k$ by the de Boor-Fix formula. We present the following lemma, which bounds the difference between $S \in \Pi_k$ and $Q_k(S)$.

**Lemma 15.** *Let $S \in \Pi_k$ and $D \in \mathcal{D}_k$. Additionally, let $\widetilde{D}$ denote the smallest cube satisfying $\widetilde{D} \supseteq D_{\boldsymbol{j}}$ for all $\boldsymbol{j} \in \mathcal{J}_k$ with* $\mathrm{supp}\, M_{k,\boldsymbol{j}}^d \cap D \neq \emptyset$. *Then, it holds*

$$\|Q_k(S)\|_{L^{(\infty,p)}(D)} \lesssim \|S\|_{L^{(\infty,p)}(\widetilde{D})},$$

*and*

$$\|S - Q_k(S)\|_{L^{(\infty,p)}(D)} \lesssim E_r(S, \widetilde{D}),$$

*Proof.* This can be proved in the same way as Lemma 4.3 of DeVore & Popov (1988). □

Finally, we define the operator $T_k : L^{(\infty,p)}(\Omega) \to \Sigma_k$ as the composition of $S_k$ and $Q_k$, i.e.,

$$T_k(f) := Q_k(S_k(f)).$$

We prove the following theorem, corresponding to Theorem 4.5 in DeVore & Popov (1988), which presents the error bound between $f$ and $T_k(f)$.

**Theorem 16.** *For any $f \in L^{(\infty,p)}(\Omega)$, it holds*

$$\|f - T_k(f)\|_{L^{(\infty,p)}(\Omega)} \leq C_{r,d,p,A} w_{r,p}(f, 2^{-k}),$$

*where $C_{r,d,p,A}$ is a constant that depends only on $r, d, p$ and $A$.*

*Proof.* For each $D \in \mathcal{D}_k$, we have

$$\|f - T_k(f)\|_{L^{(\infty,p)}(D)} \lesssim \|f - S_k(f)\|_{L^{(\infty,p)}(D)} + \|S_k(f) - Q_k(S_k(f)\|_{L^{(\infty,p)}(D)}. \tag{6}$$

As for the first term, we have

$$\|f - S_k(f)\|_{L^{(\infty,p)}(D)} = \|f - P_D(f)\|_{L^{(\infty,p)}(D)} \lesssim E_r(f, D). \tag{7}$$

The second term can be bounded using Lemma 15 and the definition of $E_r$ as

$$\begin{aligned}
\|S_k(f) - Q_k(S_k(f))\|_{L^{(\infty,p)}(D)} &\lesssim E_r(S_k(f), \widetilde{D}) \\
&\leq \sum_{D' \in \mathcal{D}_k, D' \subset \widetilde{D}} E_r(S_k(f), D') \\
&\lesssim \sum_{D' \in \mathcal{D}_k, D' \subset \widetilde{D}} \|P_D(S_k(f)) - S_k(f)\|_{L^{(\infty,p)}(D')}.
\end{aligned}$$

By the definition of $S_k$, we have $P_D(S_k(f)) = P_D(f)$. Moreover, for any $x \in D'$, we have $S_k(f)(x) = P_{D'}(f)(x)$. Therefore, it holds

$$\|S_k(f) - Q_k(S_k(f))\|_{L^{(\infty,p)}(D)} \lesssim \sum_{D' \in \mathcal{D}_k, D' \subset \widetilde{D}} \|P_D(f) - P_{D'}(f)\|_{L^{(\infty,p)}(D')}.$$

Each term in the right-hand side can be bounded as

$$\begin{aligned}
\|P_D(f) - P_{D'}(f)\|_{L^{(\infty,p)}(D')} &\lesssim \|f - P_D(f)\|_{L^{(\infty,p)}(D')} + \|f - P_{D'}(f)\|_{L^{(\infty,p)}(D')} \\
&\lesssim E_r(f, D') \leq E_r(f, \widetilde{D}),
\end{aligned}$$

where we used Lemma 13 in the second inequality. Since the number of cubes $D' \in \mathcal{D}_k$ satisfying $D' \subset \widetilde{D}$ depends only on $d$ and $r$. Thus, we have

$$\|S_k(f) - Q_k(S_k(f))\|_{L^{(\infty,p)}(D)} \lesssim E_r(f, \widetilde{D}). \tag{8}$$

Combining (6), (7) and (8), we have

$$\|f - T_k(f)\|_{L^{(\infty,p)}(D)} \lesssim E_r(f, D) + E_r(f, \widetilde{D}) \lesssim E_r(f, \widetilde{D}). \tag{9}$$

Now, each point $x \in \Omega$ appears in only a constant depending only on $r$ and $d$ number of cubes $\widetilde{D}$. Therefore, using Lemma 14 and the similar argument of Lemma 3.1 in DeVore & Popov (1988), we have

$$\|f - T_k(f)\|_{L^{(\infty,p)}(\Omega)}^p \lesssim \sum_{D \in \mathcal{D}} E_r(f, \widetilde{D})^p$$

$$\lesssim t^{-d} \int_{\|h\|_\infty \leq t} \|\Delta_h^r(f, x)\|_{L^{(\infty,p)}(\Omega(rs))}^p \, \mathrm{d}h$$

with $t := \max l_{\widetilde{D}} \lesssim 2^{-k}$. The assertion is proved by using $w_{r,p}(f, t)^p \lesssim w_{r,p}(f, 2^{-k})$. $\qquad\square$

## A.2. Equivalent Norm for Besov spaces with Anisotropic Integrability

Theorem 16 implies that we can approximate $f \in B_{p,q}^s$ with $T_k(f) = \sum_{\boldsymbol{j} \in \mathcal{J}_k} \alpha_{\boldsymbol{j}}(S_k(f)) M_{k,\boldsymbol{j}} \in \Sigma_k$. In particular, we have $T_k(f) \to f$ as $k \to \infty$ in the sense of $L^{(\infty,p)}$. We define $t_k(f)$ as

$$t_k(f) := T_k(f) - T_{k-1}(f),$$

with $T_{-1} := 0$. Using the spline decomposition $t_k(f) = \sum_{\nu \in \mathcal{J}_k} \alpha_{k,\nu}(t_k(f)) M_{k,\nu}$, we have Then, we have

$$f = \sum_{k=0}^\infty t_k(f) = \sum_{k=0}^\infty \sum_{\nu \in \mathcal{J}_k} \alpha_{k,\nu}(t_k(f)) M_{k,\nu}, \tag{10}$$

in the sense of $L^{(\infty,p)}$. In this subsection, we aim to derive an equivalent form of the norm $\|\cdot\|_{B_{p,q}^s}$ using $(\alpha_{k,\nu}(t_k(f)))_{k,\nu}$.

We first aim to equivalently express $\|S\|_{L^{(\infty,p)}}$ for $S \in \Sigma_k$ using $(\alpha_{\boldsymbol{j}}(S))_{\boldsymbol{j}}$. To do this, we first prove the following lemma.

**Lemma 17.** *For any $p \in [1, \infty]$ and $P \in \Pi_k$, we have*

$$|\alpha_{\boldsymbol{j}}(P)| \leq C 2^{k d_Y/p} \|P\|_{L^{(\infty,p)}(D_{\boldsymbol{j}})},$$

*for some constant $C > 0$.*

*Proof.* Using the same argument as Lemma 4.1 in DeVore & Popov (1988), we have

$$|\alpha_{\boldsymbol{j}}(P)| \leq C' \|P\|_{L^\infty(D_{\boldsymbol{j}})},$$

for some constant $C' > 0$.

We upper bound the right-hand side using $\|P\|_{L^{(\infty,p)}(D_{\boldsymbol{j}})}$. Let $D_{\boldsymbol{j}}^X$ and $D_{\boldsymbol{j}}^Y$ be the projections of $D_{\boldsymbol{j}}$ onto the $X$- and $Y$-coordinates, respectively, so that $D_{\boldsymbol{j}} = D_{\boldsymbol{j}}^X \times D_{\boldsymbol{j}}^Y$. For each fixed $x \in D_{\boldsymbol{j}}^X$, the inverse inequality for polynomials on $D_{\boldsymbol{j}}^Y$ gives

$$\sup_{y \in D_{\boldsymbol{j}}^Y} |P(x,y)| \leq C \lambda(D_{\boldsymbol{j}}^Y)^{-1/p} \|P(x,\cdot)\|_{L^p(D_{\boldsymbol{j}}^Y)}.$$

Taking the supremum over $x \in D_{\boldsymbol{j}}^X$, we obtain

$$\|P\|_{L^\infty(D_{\boldsymbol{j}})} = \sup_{x \in D_{\boldsymbol{j}}^X} \sup_{y \in D_{\boldsymbol{j}}^Y} |P(x,y)|$$

$$\leq C \lambda(D_{\boldsymbol{j}}^Y)^{-1/p} \sup_{x \in D_{\boldsymbol{j}}^X} \|P(x,\cdot)\|_{L^p(D_{\boldsymbol{j}}^Y)}$$

$$\leq C \lambda(D_{\boldsymbol{j}}^Y)^{-1/p} \left\| \sup_{x \in D_{\boldsymbol{j}}^X} |P(x,\cdot)| \right\|_{L^p(D_{\boldsymbol{j}}^Y)}$$

$$= C 2^{k d_Y/p} \|P\|_{L^{(\infty,p)}(D_{\boldsymbol{j}})},$$

where we used $\lambda(D_{\boldsymbol{j}}^Y) = 2^{-k d_Y}$. This completes the proof. $\qquad\square$

Using the lemma above, we equivalently express $\|S\|_{L^{(\infty,p)}}$ using $(\alpha_{\boldsymbol{j}}(S))_{\boldsymbol{j}}$.

**Lemma 18.** *Let $S \in \Sigma_k$ and $S = \sum_{\boldsymbol{j} \in \mathcal{J}_k} \alpha_{\boldsymbol{j}}(S) M_{k,\boldsymbol{j}}$. Then, for any $p \in [1,\infty]$, we have*

$$c_1 \|S\|_{L^{(\infty,p)}} \leq \left( \sum_{\boldsymbol{j} \in \mathcal{J}_k} |\alpha_{\boldsymbol{j}}(S)|^p 2^{-kd_Y} \right)^{1/p} \leq c_2 \|S\|_{L^{(\infty,p)}},$$

*for some constant $c_1, c_2 > 0$.*

*Proof.* The proof is almost the same as Lemma 4.2 in DeVore & Popov (1988).

We first prove the right-most inequality. Since it holds $S \in \Sigma_k \subseteq \Pi_k$, Lemma 17 implies that

$$\alpha_{\boldsymbol{j}}(S) \leq C 2^{kd_Y/p} \|S\|_{L^{(\infty,p)}(D_{\boldsymbol{j}})}.$$

Therefore, we have

$$\sum_{\boldsymbol{j} \in \mathcal{J}_k} |\alpha_{\boldsymbol{j}}(S)|^p 2^{-kd_Y} \leq C \sum_{\boldsymbol{j} \in \mathcal{J}_k} \|S\|^p_{L^{(\infty,p)}(D_{\boldsymbol{j}})}.$$

Since each point in $\Omega$ belongs to at most constant number of cubes $D_{\boldsymbol{j}}$, we have $\sum_{\boldsymbol{j} \in \mathcal{J}_k} \|S\|^p_{L^{(\infty,p)}(D_{\boldsymbol{j}})} \lesssim \|S\|_{L^{(\infty,p)}}$. This concludes the first part of the proof.

Next, we prove the left-most side. For each point $(x,y) \in \Omega$, the number of B-splines $M_{k,\boldsymbol{j}}$ satisfying $M_{k,\boldsymbol{j}}(x,y) \neq 0$ is at most constant. Therefore, we have

$$|S(x,y)|^p = \left| \sum_{\boldsymbol{j} \in \mathcal{J}_k} \alpha_{\boldsymbol{j}}(S) M_{k,\boldsymbol{j}}(x,y) \right|^p \lesssim \sum_{\boldsymbol{j} \in \mathcal{J}_k} |\alpha_{\boldsymbol{j}}(S)|^p |M_{k,\boldsymbol{j}}(x,y)|^p.$$

Taking the supremum over $x$ and integrating over $y$, we have

$$\|S\|^p_{L^{(\infty,p)}} \lesssim \sum_{\boldsymbol{j} \in \mathcal{J}_k} |\alpha_{\boldsymbol{j}}(S)|^p \int_{\Omega_Y} \left( \sup_{x \in \Omega_X} |M_{k,\boldsymbol{j}}(x,y)| \right)^p \mathrm{d}y \lesssim C \sum_{\boldsymbol{j} \in \mathcal{J}_k} |\alpha_{\boldsymbol{j}}(S)|^p 2^{-kd_Y},$$

which completes the proof. $\qquad\square$

We also provide inverse estimates to Theorem 16. We define

$$\sigma_k(f) := \begin{cases} \inf_{S \in \Sigma_k} \|f - S\|_{L^{(\infty,p)}(\Omega)}, & (k \geq 0) \\ \|f\|_{L^{(\infty,p)}(\Omega)}. & (k = -1) \end{cases}$$

and obtain the following bound.

**Lemma 19.** *For each $k > 0$ and $f \in L^{(\infty,p)}(\Omega)$, we have*

$$w_r(f, 2^{-k}) \leq c \sum_{l=-1}^{k} 2^{(l-k)(r-1+1/p)} \sigma_l(f),$$

*for some constant $c > 0$.*

This lemma can be proved in the same way as Theorem 4.8 in DeVore & Popov (1988) using our Lemma 18 instead of their eq. (4.12).

Finally, we provide the norm expressed by $\alpha_{\boldsymbol{j}}(S_k(f))$ equivalent to $\|f\|_{B^s_{p,q}}$. To provide the statement, for a sequence $(a_k)_k$ whose component functions belong to $L^{(\infty,p)}(\Omega)$, we define the norm $\|\cdot\|_{l^\alpha_{p,q}}$ as follows:

$$\|(a_k)_k\|_{l^\alpha_{p,q}} = \left( \sum_{k=0}^{\infty} \left[ 2^{k\alpha} \|a_k\|_{L^{(\infty,p)}(\Omega)} \right]^q \right)^{1/q}.$$

Then, we obtain the following theorem.

**Theorem 20.** *Let $p, q \in [1, \infty]$ and $s \in (0, r - 1 + 1/p)$. Then, the following norms are equivalent to $\|f\|_{B^s_{p,q}}$ up to constants:*

$$N_1(f) := \|(\sigma_k(f))_k\|_{l^s_{p,q}} + \|f\|_{L^{(\infty,p)}(\Omega)},$$

$$N_2(f) := \|(f - T_k(f))_k\|_{l^s_{p,q}} + \|f\|_{L^{(\infty,p)}(\Omega)},$$

$$N_3(f) := \|(t_k(f))_k\|_{l^s_{p,q}},$$

$$N_4(f) := \left\{ \sum_{k=0}^{\infty} \left[ 2^{k(s - d_Y/p)} \left( \sum_{j \in \mathcal{J}_k} |\alpha_{k,\nu}(t_k)|^p \right)^{1/p} \right]^q \right\}^{1/q}.$$

*Proof.* We first prove that $N_1(f) \leq N_2(f) \lesssim \|f\|_{B^s_{p,q}}$. Theorem 16 implies that $\|f - T_k(f)\|_{L^{(\infty,p)}} \lesssim w_{r,p}(f, 2^{-k})$. Since we can discretize the definition of Besov norm as

$$\|f\|_{B^s_{p,q}} \sim \|f\|_{L^{(\infty,p)}} + \left( \sum_{k=0}^{\infty} (2^{ks} w_{r,p}(f, 2^{-k}))^q \right)^{1/q},$$

we have $N_2(f) \lesssim \|f\|_{B^s_{p,q}}$. Moreover, by definition of $\sigma_k$, we have $\sigma_k(f) \leq \|f - T_k(f)\|_{L^{(\infty,p)}}$, which yields $N_1(f) \leq N_2(f)$.

Next, we prove $\|f\|_{B^s_{p,q}} \lesssim N_1(f)$. Lemma 19 implies that

$$w_r(f, 2^{-k}) \leq c2^{-k} \sum_{l=-1}^{k} 2^{l(r-1+1/p)} \sigma_l(f).$$

Using discrete Hardy inequality (see (5.2) and (5.3) in DeVore & Popov (1988)), we have

$$\left\| w_r(f, 2^{-k}) \right\|_{l^s_{p,q}} \lesssim \|\sigma_k(f)\|_{l^s_{p,q}}.$$

Using the discretized form of Besov norm again, we have $\|f\|_{B^s_{p,q}} \lesssim N_1(f)$. Therefore, we have

$$\|f\|_{B^s_{p,q}} \sim N_1(f) \sim N_2(f).$$

We prove the equivalence of $N_2$ and $N_3$. By the definition of $t_k$, we have

$$t_k(f) = T_k(f) - T_{k-1}(f) = (T_k(f) - f) + (f - T_{k-1}(f)).$$

Therefore, we have

$$\|t_k(f)\|_{L^{(\infty,p)}} \lesssim \|T_k(f) - f\|_{L^{(\infty,p)}} + \|f - T_{k-1}(f)\|_{L^{(\infty,p)}},$$

which implies $N_3(f) \lesssim N_2(f)$. Additionally, since $T_k = t_0 + \cdots + t_k$ and $T_k \to f$ in $L^{(p,\infty)}$, we have $f - T_k(f) = \sum_{l=k+1}^{\infty} t_l(f)$. Therefore, we have

$$\|f - T_k(f)\|_{L^{(\infty,p)}} \lesssim \sum_{l=k+1}^{\infty} \|t_l(f)\|_{L^{(\infty,p)}}.$$

Using discrete Hardy inequality, we have

$$\|(f - T_k(f))_k\|_{l^s_{p,q}} \lesssim \|(t_k(f))_k\|_{l^s_{p,q}} = N_3(f).$$

Moreover, we have

$$\|f\|_{L^{(\infty,p)}} = \|f - T_{-1}(f)\|_{L^{(\infty,p)}} \lesssim \sum_{l=0}^{\infty} \|t_l(f)\|_{L^{(\infty,p)}}.$$

Hölder's inequality implies

$$
\begin{aligned}
\|f\|_{L^{(\infty,p)}} &= \sum_{l=0}^{\infty} \left( 2^{-ls} \cdot 2^{ls} \|t_l(f)\|_{L^{(\infty,p)}} \right) \\
&\lesssim \left( \sum_{l=0}^{\infty} 2^{-\frac{qls}{q-1}} \right)^{\frac{q-1}{q}} \left( \sum_{l=0}^{\infty} \left( 2^{ls} \|t_l(f)\|_{L^{(\infty,p)}} \right)^q \right)^{\frac{1}{q}} \\
&\lesssim \left( \sum_{l=0}^{\infty} \left( 2^{ls} \|t_l(f)\|_{L^{(\infty,p)}} \right)^q \right)^{\frac{1}{q}} = N_3(f).
\end{aligned}
$$

Therefore, we have

$$
N_2(f) = \|(f - T_k(f))_k\|_{l_{p,q}^s} + \|f\|_{L^{(\infty,p)}} \lesssim N_3(f).
$$

Hence, we have $N_2(f) \sim N_3(f)$.

Finally, we prove the equivalence of $N_3$ and $N_4$. Applying Lemma 18 with $S = t_k = \sum_{\nu \in \mathcal{J}_k} \alpha_{k,\nu}(t_k) M_{k,\nu}$, we have

$$
\|t_k\|_{L^{(\infty,p)}} \sim \left( \sum_{\boldsymbol{j} \in \mathcal{J}_k} |\alpha_{k,\nu}(t_k)|^p 2^{-kd_Y} \right)^{1/p}.
$$

Therefore, it holds

$$
\begin{aligned}
N_3(f) &= \left( \sum_{k=0}^{\infty} \left[ 2^{ks} \|t_k\|_{L^{(\infty,p)}(\Omega)} \right]^q \right)^{1/q} \\
&\sim \left( \sum_{k=0}^{\infty} \left[ 2^{ks} \left( \sum_{\boldsymbol{j} \in \mathcal{J}_k} |\alpha_{k,\nu}(t_k)|^p 2^{-kd_Y} \right)^{1/p} \right]^q \right)^{1/q} \\
&\sim \left( \sum_{k=0}^{\infty} \left[ 2^{k(s-d_Y/p)} \left( \sum_{\boldsymbol{j} \in \mathcal{J}_k} |\alpha_{k,\nu}(t_k)|^p \right)^{1/p} \right]^q \right)^{1/q} = N_4(f),
\end{aligned}
$$

which completes the proof. $\qquad\square$

As a byproduct, we obtain the following corollary.

**Corollary 21.** *Let $p \in [1,\infty], q \in [1,\infty]$, and assume $s > d_Y/p$. Then every $f \in B_{p,q}^s([0,1]^d)$ has a continuous representative. In particular, $B_{p,q}^s([0,1]^d) \hookrightarrow C(Q)$ in the sense of continuous representatives.*

*Proof.* Choose an integer $r \geq 2$ such that $s < r - 1 + 1/p$. Theorem 20 implies that, for $f \in B_q^s(L_p(Q))$ we have a decomposition $f = \sum_{k=0}^{\infty} t_k(f)$ in $L^{(\infty,p)}$, and it holds

$$
\|(t_k)\|_{\ell_q^s(L_p)} = \left( \sum_{k=0}^{\infty} \left( 2^{ks} \|t_k(f)\|_{L^{(\infty,p)}} \right)^q \right)^{1/q} < \infty.
$$

Since it holds $S \in \Sigma_k \subseteq \Pi_k$, the proof of Lemma 17 implies that

$$
\|S\|_{L^\infty} \lesssim 2^{kd_Y/p} \|S\|_{L^{(\infty,p)}}
$$

Applying this to $S = t_k$, we obtain

$$\|t_k\|_{L^\infty([0,1]^d)} \lesssim 2^{k d_Y/p} \|t_k\|_{L^{(\infty,p)}} = 2^{-k(s-d_Y/p)} \big(2^{ks}\|t_k\|_{L^{(\infty,p)}}\big).$$

Therefore, we have

$$\sum_{k=0}^{\infty} \|t_k\|_{L^\infty} \lesssim \sum_{k=0}^{\infty} 2^{-k(s-d_Y/p)} \big(2^{ks}\|t_k\|_{L^{(\infty,p)}}\big).$$

If $q > 1$, Holder's inequality implies that

$$\sum_{k=0}^{\infty} 2^{-k(s-d_Y/p)} \big(2^{ks}\|t_k\|_{L_p(Q)}\big) \le \left(\sum_{k=0}^{\infty} \Big[2^{-k(s-d_Y/p)}\Big]^{\frac{q}{q-1}}\right)^{\frac{q-1}{q}} \left(\sum_{k=0}^{\infty} \big(2^{ks}\|t_k\|_{L^{(\infty,p)}}\big)^q\right)^{1/q}.$$

The first factor is finite because $s - d_Y/p > 0$. Hence, we have

$$\sum_{k=0}^{\infty} \|t_k\|_{L^\infty([0,1]^d)} < \infty.$$

The same argument follows for the case $q = 1$ from $2^{-k(s-d_Y/p)} \le 1$. Thus, $\sum_{k=0}^{\infty} t_k$ converges absolutely in $L^\infty([0,1]^d)$, hence uniformly on $[0,1]^d$.

Since $r \ge 2$, the B-splines $M_{k,\nu}$ are continuous. Each $t_k \in \Sigma_k$ is a finite linear combination of such B-splines, so each $t_k$ is continuous on $[0,1]^d$. Therefore the uniform limit $F := \sum_{k=0}^{\infty} t_k$ is continuous on $[0,1]^d$. On the other hand, by Theorem 5.1 the same series converges to $f$ in $L^{(\infty,p)}([0,1]^d)$. Hence it holds $F = f$ almost everywhere on $[0,1]^d$. Therefore $f$ has the continuous representative $F$. $\qquad\square$

### A.3. Approximation Error via Neural Networks

We first prove the following theorem, which gives the approximation error bound for a fixed small set $\Omega' \subseteq \Omega$.

**Theorem 22** (Approximation Error for Besov Spaces). *Let $\Omega' \subseteq \Omega$ be a measurable set satisfying $\mathcal{M}(\iota; \Omega', \|\cdot\|_2) \le C_0\big(1 + \lambda(\Omega')\iota^{-d}\big)$ for all $\iota \in (0, c_0]$ with some constants $C_0, c_0 > 0$. Assume that $f \in B_{p,q}^s(\Omega)$ with $p, q \in (0, \infty]$, $s > \delta_0$, where $\delta_0 := (1/p - 1/r)_+$. Moreover, suppose that $m \in \mathbb{N}$ satisfies $0 < s < \min\{m, m-1+1/p\}$. Let $\nu \in (0, \frac{s-\delta_0}{\delta_0})$, and $N \in \mathbb{N}_{>0}$ be a sufficiently large integer. We define $N' := \lambda(\Omega')^{\frac{\nu}{1+\nu}} N$ and $\epsilon := N^{-s/d-(1+\nu^{-1})(d/p-s)_+} \log^{-1} N$. Then, there exists an FNN $f \in \Psi(L, W, S, B)$ with*

$$L = L_0, \quad W \lesssim N'W_0, \quad S \lesssim (L-1)W_0^2 N' + N', \quad B = O(N^{d(1+\nu^{-1})(d/p-s)_+}),$$

*such that $\|f - f^\circ\|_{L^r(\Omega)} \lesssim N^{-s/d}$, where*

$$L_0 := 3 + 2\left\lceil \log_2\left(\frac{3^{d\vee m}}{\epsilon c_{d,m}}\right) + 5\right\rceil \lceil \log_2(d \vee m)\rceil, \quad W_0 := 6dm(m+2) + 2d.$$

*and $c_{d,m}$ is a constant depending only on $d$ and $m$.*

We use the following lemma for the proof of Theorem 22. The proof can be done in the same way as Theorem 3.1 of Dũng (2011) (or Lemma 2 of Suzuki & Nitanda (2021)) using (10) and Theorem 16.

**Lemma 23.** *Suppose the function $f \in B_{p,q}^s(\Omega)$ and the constants $m \in \mathbb{N}$, $\delta_0$, $\nu$ satisfy the same conditions as in Theorem 22. Fix an integer $K \in \mathbb{N}$, and let $N = 2^{dK}$. We define $\epsilon$ in the same way as in Theorem 22. Then, there exists $f_N$ such that $\|f - f_N\|_{L^r(\Omega)} \lesssim N^{-s/d}\|f\|_{B_{p,q}^s}$, and $f_N$ can be written as*

$$f_N(x) = \sum_{(k,j) \in E_N} \alpha_{k,j} M_{k,j}^d(x) := \sum_{k=0}^{K} \sum_{j \in J(k)} \alpha_{k,j} M_{k,j}^d(x) + \sum_{k=K+1}^{K^*} \sum_{i=1}^{n_k} \alpha_{k,j_i} M_{k,j_i}^d(x),$$

*where $K^* = \left\lceil K\big(1 + \frac{1}{\nu}\big)\right\rceil$, $n_k = \lceil 2^{d(K-\nu(k-K))}\rceil$ $(k = K+1, \dots, K^*)$, $\{j_i\}_{i=1}^{n_k} \subset \mathcal{J}_k$, and the coefficients $(\alpha_{k,j})_{k,j}$ satisfies $\max_{(k,j) \in E_N} |\alpha_{k,j}| \lesssim 2^{K^* \cdot (d/p-s)_+}$.*

Now, we prove Theorem 22.

*Proof of Theorem 22.* Let $f_N = \sum_{(k,j) \in E_N} \alpha_{k,j} M_{k,j}^d$ be the approximation of $f$ given in Lemma 23. Then, we have $\|f - f_N\|_{L^r(\Omega)} \lesssim N^{-s/d} \|f\|_{B_{p,q}^s}$. We define

$$E_N' := \left\{ (k,j) \in E_N \mid \left( \operatorname{supp} M_{k,j}^d \right) \cap \Omega' \neq \emptyset \right\},$$

and

$$E_{N,k}' := \left\{ j \in \mathbb{N} \mid (k,j) \in E_N' \right\} = \left\{ j \in \mathbb{N} \mid (k,j) \in E_N, \left( \operatorname{supp} M_{k,j}^d \right) \cap \Omega' \neq \emptyset \right\},$$

for $k \in [K^*]$. Then, we define $f_N'$ as

$$f_N' = \sum_{(k,j) \in E_N'} \alpha_{k,j} M_{k,j}^d = \sum_{k=0}^{K^*} \sum_{j \in E_{N,k}'} \alpha_{k,j} M_{k,j}^d.$$

By the definition of $E_N'$, we have $f_N(x) = f_N'(x)$ for any $x \in \Omega'$.

Next, we evaluate the number of terms in the decomposition of $f_N'$, i.e., $|E_N'|$. The set $\operatorname{supp} M_{k,j}^d$ is the cube with edge length $(m+1)2^{-k}$ as shown in (5). Therefore, it holds $|E_{N,k}'|$ is smaller than the packing number $\mathcal{P}((m+1)2^{-k}; \Omega', \|\cdot\|_2)$. Therefore, we have

$$\begin{aligned} |E_{N,k}'| &\leq \mathcal{P}((m+1)2^{-k}; \Omega', \|\cdot\|_2) \\ &\leq \mathcal{M}((m+1)2^{-k-1}; \Omega', \|\cdot\|_2) \\ &\lesssim 1 + \lambda(\Omega')2^{dk}. \end{aligned}$$

Moreover, for $k \geq K$, we have

$$|E_{N,k}'| \leq n_k \lesssim 2^{d(K-\nu(k-K))}.$$

Hence, for any $K^\circ \geq K$, we have

$$|E_N'| \lesssim \sum_{k=0}^{K^\circ} \left( 1 + \lambda(\Omega')2^{dk} \right) + \sum_{k=K^\circ+1}^{K^*} 2^{d(K-\nu(k-K))}.$$

Let us determine $K^\circ$ to make the right-hand side the minimum. Since $1 + \lambda(\Omega')2^{dk}$ is increasing and $2^{\|K\|_{s'} - \nu(\|k\|_{s'} - \|K\|_{s'})}$ is decreasing with respect to $k$, for the best choice of $K^\circ$, we have

$$\lambda(\Omega')2^{dK^\circ} \simeq 2^{d(K-\nu(K^\circ-K))}.$$

Therefore, we have

$$2^{-(1+\nu)d(K^\circ-K)} \simeq \lambda(\Omega'),$$

which implies $2^{d(K^\circ-K)} \simeq \lambda(\Omega')^{-\frac{1}{1+\nu}}$. For $K^\circ$ satisfying this condition, we have

$$\begin{aligned} |E_N'| &\lesssim \lambda(\Omega')\frac{2^{dK^\circ}}{1 - 2^d} + 2^{d(K-\nu(K^\circ-K))}\frac{1}{1 - 2^{-d\nu}} \\ &\lesssim \lambda(\Omega') \cdot 2^{dK}\lambda(\Omega')^{-\frac{1}{1+\nu}} + 2^{dK}\lambda(\Omega')^{\frac{\nu}{1+\nu}} \\ &\lesssim \lambda(\Omega')^{\frac{\nu}{1+\nu}}2^{dK} = \lambda(\Omega')^{\frac{\nu}{1+\nu}} \cdot N. \end{aligned}$$

The remaining part of the proof is adapted from the proof of Proposition 2 of Suzuki & Nitanda (2021). Specifically, from Lemma 1 of Suzuki (2019), for all $k$ and $j$, there exists an FNN $\widehat{M}_{k,j}^d$ such that $\left\| \widehat{M}_{k,j}^d - M_{k,j}^d \right\|_{L^\infty(\mathbb{R}^d)} \leq \epsilon$, and $\widehat{M}_{k,j}^d = 0$ in $x \notin [0, m+1]^d$. Using these networks, we can construct $\hat{f} \in \Psi(L, W, S, B)$ with $L, W, S, B$ as in the statement of the theorem such that

$$\hat{f}(x) = \sum_{(k,j) \in E_N'} \alpha_{k,j} \widehat{M}_{k,j}^d(x).$$

Then, we have

$$\left| f_N'(x) - \hat{f}(x) \right| \leq \sum_{(k,j) \in E_N'} |\alpha_{k,j}| \cdot \left| M_{k,j}^d(x) - \widehat{M}_{k,j}^d(x) \right|$$

$$\leq \epsilon \sum_{(k,j) \in E_N'} |\alpha_{k,j}| \cdot \mathbb{1}_{\mathrm{supp}\, M_{k,j}^d}(x).$$

For each $x \in \Omega$, the number of $(k,j)$ such that $x \in \mathrm{supp}\, M_{k,j}^d$ is at most $(m+1)^d(1 + K^*)$. Combining with the upper-bound $\max_{(k,j) \in E_N'} |\alpha_{k,j}|$ given in Lemma 23, Therefore, we have

$$\left| f_N'(x) - \hat{f}(x) \right| \leq \epsilon \max_{(k,j) \in E_N'} |\alpha_{k,j}| \cdot (m+1)^d(1 + K^*)$$

$$\lesssim \epsilon 2^{K^* \cdot (d/p - s)_+}(1 + K^*).$$

Since it holds

$$2^{K^*} \simeq 2^{K \cdot (1 + \nu^{-1})} \simeq N^{1 + \nu^{-1}},$$

we have

$$\left| f_N'(x) - \hat{f}(x) \right| \lesssim \epsilon N^{(1 + \nu^{-1})(d/p - s)_+} \log N \leq N^{-s/d}.$$

Here, in the right-most inequality, we utilize the definition of $\epsilon$. Moreover, the absolute values of parameters used in $\widehat{M}_{k,j}^d$ is at most $2^{K^*} \lesssim N^{(1 + \nu^{-1})(d/p - s)_+}$, which completes the proof. $\qquad \square$

## A.4. Distribution-adaptive Approximation Error Bound

For $r^\circ \in B_{p,q}^s$ and $t > 0$, we define

$$r^*(x) := \max_{y \in \Omega_Y} r^\circ(x,y),$$

$$y^*(x) := \arg\max_{y \in \Omega_Y} r^\circ(x,y),$$

$$\overline{S}_t := \{(x,y) \mid r^*(x) - r^\circ(x,y) \geq t\},$$

$$S_t(x) := \{y \mid r^*(x) - r^\circ(x,y) \geq t\}.$$

In the subsequent two subsections, we consider the following assumption.

**Assumption 24.** *Let $C_0, c_0, c_1, T, \mathcal{R} > 0$ and $\beta \in \left[0, \frac{d}{2(s - d/p)}\right)$. For $r^\circ \in B_{p,q}^s$, we assume the following four conditions.*

**(B1)** *For all $\iota \in (0, c_0]$ and $t \in [0, T]$, it holds $\mathcal{M}(\iota; \overline{S}_t, \|\cdot\|_2) \leq C_0\left(1 + \lambda(\overline{S}_t)\iota^{-d}\right)$.*

**(B2)** *For all $t \in [0, T]$, it holds $\lambda(\overline{S}_t) \lesssim t^\beta$.*

**(B3)** *It holds $\mathbb{E}_{x \sim P_X}[r^*(x) - r^\circ(x,y)] \leq \mathcal{R}$.*

**(B4)** *It holds $r^* \in \overline{B}_{p,q}^s(\Omega_X)$.*

Then, we prove the following theorem, which shows that, under Assumption 24, we obtain an improved error bound when $\mathcal{R}$ is small.

**Theorem 25** (Distribution-adaptive Error Bound for Besov Spaces). *Let $f^\circ \in B_{p,q}^s$, and assume that it satisfies Assumption 24. Additionally, let $\varsigma$ be a constant such that $\varsigma \in (0, s - d/p)$ for $p < \infty$, and $\varsigma = s$ for $p = \infty$. Moreover, suppose that $m \in \mathbb{N}$ satisfies $0 < s < \min\{m, m - 1 + 1/p\}$. Then, for sufficiently large real number $N \in \mathbb{R}_{>0}$, there exists an FNN $f \in \Psi(L, W, S, B)$ with*

$$L \lesssim \log N + \log \mathcal{R}^{-1}, \quad W \lesssim N \log \mathcal{R}^{-1} + \mathcal{R}^{-d/s},$$

$$S \lesssim N \log N \log \mathcal{R}^{-1} + \mathcal{R}^{-d/s} \log \mathcal{R}^{-1}, \quad \log B \lesssim \log \mathcal{R}^{-1},$$

*such that $\|f - f^\circ\|_{L^2(\Omega)} \lesssim N^{-s/d} \mathcal{R}^{\beta\varsigma/d}$.*

*Proof.* Applying Theorem 22 with $r = \infty$, we have that, for all $t \in [0, T]$, there exists an FNN $f'_t \in \Psi(L', W'_t, S'_t, B')$ with

$$L' \lesssim \log\left(\epsilon^{-1}\right) \lesssim \log\left(N^{s/d} \log N\right) \lesssim \log N' + \log t^{-1},$$

$$W'_t \lesssim N' W_0 \lesssim N',$$

$$S'_t \lesssim N' \log N + N' \lesssim N'(\log N' + \log t^{-1}),$$

$$B' \lesssim 1,$$

such that

$$\sup_{x \in \Omega_t} |f'_t(x) - f^\circ(x)| \lesssim N^{-s/d} \lesssim (N')^{-s/d} t^{\beta \cdot \frac{\nu s/d}{1+\nu}} \lesssim (N')^{-s/d} t^{\beta \varsigma/d}.$$

Here, we define $\varsigma := \frac{\nu s}{1+\nu}$. Since $\nu$ is the number taking an arbitrary value in $\left(0, \frac{s-d/p}{d/p}\right)$, we have $\varsigma \in (0, s - d/p)$.

Let $a_{-1} = 0$ and $a_i = 2^i \mathcal{R}$ for $i = 0, \ldots, I$ with $I := \lceil \log_2(2F/\mathcal{R}) \rceil$. Then, for $i = 0, \ldots, I$ and any $N \in \mathbb{R}$, we can construct an FNN $f_i \in \Psi(L_i, W_i, S_i, B_i)$ with

$$L_i \lesssim \log N + \log \mathcal{R}^{-1}, \quad W_i \lesssim N, \quad S_i \lesssim N(\log N + \log \mathcal{R}^{-1}), \quad B_i \lesssim 1,$$

such that

$$\sup_{x \in \Omega_{a_i}} |f_i(x) - f^\circ(x)| \lesssim N^{-s/d} (2^i \mathcal{R})^{\beta \varsigma/d}.$$

Next, let $g : \Omega \to \mathbb{R}, (x, y) \mapsto r^*(x) - r^\circ(x, y)$. Applying Theorem 22 for $r^\circ$ with $N \leftarrow \mathcal{R}^{-d/s}$ and $\mathcal{R} \leftarrow T$, and Proposition 1 of Suzuki (2019) for $r^*$ with $N \leftarrow \mathcal{R}^{-d/s}$, we have an FNN $\widetilde{g} \in \Psi(L, W_g, S_g, B_g)$ with

$$L_g \lesssim \log \mathcal{R}^{-1}, \quad W_g \lesssim \mathcal{R}^{-d/s}, \quad S_g \lesssim \mathcal{R}^{-d/s} \log \mathcal{R}^{-1}, \quad \log B_g \lesssim \log \mathcal{R}^{-1},$$

such that $\|g - \widetilde{g}\|_{L^\infty(\Omega)} \lesssim \mathcal{R}/8$.

For $i = 0, \ldots, I$, we can construct an FNN $\phi_i \in \Phi(L, W, S, B)$ with $L, W, S \lesssim 1$ and $\log B \lesssim \log \mathcal{R}^{-1}$ such that

$$\phi_i(x) = \begin{cases} 0 & (x \leq a_{i-1} - \mathcal{R}/4), \\ (x - a_{i-1} + \eta)/(2\eta) & (a_{i-1} - \mathcal{R}/4 < x < a_{i-1} - \mathcal{R}/8), \\ 1 & (a_{i-1} + \eta \leq x \leq a_i - \mathcal{R}/4), \\ (a_i + \eta - x)/(2\eta) & (a_i - \mathcal{R}/4 < x < a_i - \mathcal{R}/8), \\ 0 & (a_i - \mathcal{R}/8 \leq x). \end{cases}$$

Then, we have $\sum_{i=0}^{I} \phi_i(x) = 1$ for all $x \in [0, 2F]$. Moreover, since $\phi_i(x) > 0$ only if $x \in [a_{i-1} - \mathcal{R}/4, a_i - \mathcal{R}/8]$, the necessary condition to $\phi_i(\widetilde{g}(x)) > 0$ is $g(x) \in [a_{i-1} - 3\mathcal{R}/8, a_i]$.

Now, we define $\check{f}$ as

$$\check{f}(x) := \sum_{i=0}^{I} \phi_i(\widetilde{g}(x)) f_i(x).$$

Let us consider $x \in \Omega$ such that $g(x) \in [a_{i-1}, a_i]$ for some $i \in [0, I]$. Then, we have $g(x) \in [a_{i-1} - 3\mathcal{R}/8, a_i]$, which implies that $\phi_j(\widetilde{g}(x)) = 0$ for $j \neq i, i-1$. Therefore, we have

$$\begin{aligned} \left|\check{f}(x) - f^\circ(x)\right| &\leq \phi_i(\widetilde{g}(x))|f_i(x) - f^\circ(x)| + \phi_{i-1}(\widetilde{g}(x))|f_{a_{i-1}}(x) - f^\circ(x)| \\ &\leq \max\{|f_i(x) - f^\circ(x)|, |f_{a_{i-1}}(x) - f^\circ(x)|\} \\ &\lesssim N^{-\widetilde{s}} (2^i \mathcal{R})^{\beta \varsigma/d}. \end{aligned}$$

Moreover, for $x \sim P_X$, the probability of $x \in [a_{i-1}, a_i]$ can be bounded as

$$\mathbb{P}_{x \sim P_X}[g(x) \in [a_{i-1}, a_i]] \leq \mathbb{P}_{x \sim P_X}[g(x) \geq 2^{i-1} \mathcal{R}] \leq \frac{\mathbb{E}_{x \sim P_X}[g(x)]}{2^{i-1} \mathcal{R}} \leq 2^{-(i-1)}.$$

Therefore, we have

$$
\begin{aligned}
\left\| \check{f} - f^\circ \right\|_{L^2(P_X)}^2 &= \mathbb{E}_{x \sim P_X}[|\check{f}(x) - f^\circ(x)|^2] \\
&\lesssim \sum_{i=0}^{I} N^{-2\widetilde{s}} (2^i \mathcal{R})^{2\beta\varsigma/d} \mathbb{P}_{x \sim P_X}[g(x) \in [a_{i-1}, a_i]] \\
&\lesssim N^{-2\widetilde{s}} \mathcal{R}^{2\beta\varsigma/d} \sum_{i=0}^{I} (2^{2\beta\varsigma/d-1})^i \\
&\lesssim N^{-2\widetilde{s}} \mathcal{R}^{2\beta\varsigma/d}.
\end{aligned}
$$

Finally, since $\phi_i(\widetilde{g}(x))$ and $f_i(x)$ are bounded by constants for all $x \in \Omega$ and $i = 0, \ldots, I$, there exists an FNN $f \in \Psi(L, W, S, B)$ with

$$
L \lesssim \log N + \log \mathcal{R}^{-1}, \quad W \lesssim N \log \mathcal{R}^{-1} + \mathcal{R}^{-d/s},
$$
$$
S \lesssim N \log N \log \mathcal{R}^{-1} + \mathcal{R}^{-d/s} \log \mathcal{R}^{-1}, \quad \log B \lesssim \log \mathcal{R}^{-1},
$$

such that $\|f - \check{f}\|_\infty \leq N^{-\widetilde{s}} \mathcal{R}^{\beta\varsigma/d}$. Then, we have

$$
\|f - f^\circ\|_{L^2(P_X)} \leq \|f - \check{f}\|_{L^2(P_X)} + \|\check{f} - f^\circ\|_{L^2(P_X)} \lesssim N^{-s/d} \mathcal{R}^{\beta\varsigma/d},
$$

which completes the proof. $\qquad\square$

## A.5. Distribution-adaptive Estimation Error Bound

We consider a general regression problem for Besov spaces. Specifically, we consider $f \in B_{p,q}^s$, and let $\widehat{f}$ be an estimator of $f^\circ$ defined as

$$
\widehat{f} := \operatorname*{arg\,min}_{f \in \Phi(L,W,S,B)} \sum_{i=1}^{n} (y_i - f(x_i))^2, \tag{11}
$$

where $x_1, \ldots, x_n$ are i.i.d. samples from a distribution $P_X$, and $y_i = f^\circ(x_i) + \xi_i$ with $\xi_i \sim \mathcal{N}(0, \sigma^2)$. We denote $D_n := \{(x_i, y_i)\}_{i=1}^{n}$ as the dataset. This theorem corresponds to Lemma 11 in the main text.

**Theorem 26** (Distribution-adaptive Estimation Error for Besov Spaces). *Let $f^\circ \in B_{p,q}^s$, and assume that it satisfies Assumption 24. Additionally, let $\varsigma$ be a constant such that $\varsigma \in (0, s - d/p)$ for $p < \infty$, and $\varsigma = s$ for $p = \infty$. Moreover, let $\widehat{f} \in \Phi(L, W, S, B)$ be an estimator defined as* (11) *with*

$$
L \lesssim \log N, \quad W \lesssim N, \quad S \lesssim N \log N, \quad \log B \lesssim \log N,
$$

*where $N = n^{\frac{d}{2s+d}} \mathcal{R}^{\frac{2\beta\varsigma}{2s+d}}$. If $\mathcal{R} \gtrsim n^{-\frac{s}{4s+d}}$, it holds*

$$
\mathbb{E}_{D_n}\left[ \left\| \widehat{f} - f^\circ \right\|_{L^2(P_X)}^2 \right] \lesssim \mathcal{R}^{\frac{2\beta\varsigma}{2s+d}} \cdot n^{-\frac{2s}{2s+d}} \log^4(n),
$$

*where $\mathbb{E}_{D_n}$ is the expectation with respect to the dataset $D_n$.*

We utilize the following proposition for the proof.

**Proposition 27** (Schmidt-Hieber (2020); Hayakawa & Suzuki (2020)). *Let $\mathcal{F}$ be a set of functions. Let $\widehat{f}$ be the least-squares estimator in $\mathcal{F}$:*

$$
\widehat{f} := \operatorname*{arg\,min}_{f \in \mathcal{F}} \sum_{i=1}^{n} (y_i - f(x_i))^2,
$$

*Assume that $\|f^\circ\|_\infty \leq F$ and $\|f\|_\infty \leq F$ for all $f \in \mathcal{F}$. If $\delta > 0$ satisfies $\mathcal{M}(\delta; \mathcal{F}, \|\cdot\|_\infty) \geq 3$, then it holds that*

$$
\mathbb{E}_{D_n}\left[ \left\| \widehat{f} - f^\circ \right\|_{L^2(P_X)}^2 \right] \lesssim C\left[ \inf_{f \in \mathcal{F}} \|f - f^\circ\|_{L^2(P_X)}^2 + (F^2 + \sigma^2) \frac{\log \mathcal{M}(\delta; \mathcal{F}, \|\cdot\|_\infty)}{n} + \delta(F + \sigma) \right],
$$

*where $C > 0$ is a universal constant.*

To upper bound the covering number of the function class of FNNs, we use the following result.

**Lemma 28** (Lemma 6 of Suzuki & Nitanda (2021))**.** *The covering number of $\Phi(L, W, S, B)$ can be bounded as*

$$\log \mathcal{M}(\delta; \Phi(L, W, S, B), \|\cdot\|_\infty) \leq 2SL \log((B+1)(W+1)) + S \log(\delta^{-1}L),$$

Now, we prove Theorem 26.

*Proof of Theorem 26.* The proof is basically the same as that of Theorem 2 of Suzuki & Nitanda (2021). The difference is that our proof explicitly provides the dependency on $\mathcal{R}$.

First, using $\mathcal{R} \gtrsim n^{-\frac{s}{4s+d}}$, we have

$$
\begin{aligned}
\mathcal{R}^{\frac{d}{s} + \frac{2\beta(s-d/p)}{2s+d}} &\geq \mathcal{R}^{\frac{d}{s} + \frac{2d}{2s+d}} \\
&\gtrsim n^{-\frac{s}{4s+d}\left(\frac{d}{s} + \frac{2d}{2s+d}\right)} \\
&\geq n^{-\frac{d}{2s+d}\left(\frac{2s+d}{4s+d} + \frac{2s}{4s+d}\right)} \\
&\geq n^{-\frac{d}{2s+d}}.
\end{aligned}
$$

Therefore, we have

$$\mathcal{R}^{-\frac{d}{s}} \lesssim n^{-\frac{d}{2s+d}} \mathcal{R}^{-\frac{2\beta(s-d/p)}{2s+d}} \simeq N.$$

Then, the configuration of $L, W, S, B$ in Theorem 25 can be simplified as

$$L \lesssim \log N, \quad W \lesssim N \log N, \quad S \lesssim N \log^2 N, \quad \log B \lesssim \log N.$$

Let $\mathcal{F} := \Phi(L, W, S, B)$. Then, the covering number of the function class $\mathcal{F}$ can be bounded as

$$
\begin{aligned}
\log \mathcal{N}(\delta, \mathcal{F}, \|\cdot\|_\infty) &\lesssim N \log^3 N(\log N + \log\log N) + N \log^2 N(\log(\delta^{-1}) + \log\log N) \\
&\lesssim N \log^2 N(\log^2 N + \log(\delta^{-1})),
\end{aligned}
$$

Using Proposition 27 and setting $\delta := 1/n$, estimation error can be bounded as

$$
\begin{aligned}
\mathbb{E}_{D_n}\left[\left\|\widehat{f} - f^\circ\right\|^2_{L^2(P_X)}\right] &\lesssim \left\|\widehat{f} - f^\circ\right\|^2_{L^\infty(\mathrm{supp}(P_X))} + \frac{N \log^2 N(\log^2 N + \log(\delta^{-1}))}{n} + \frac{1}{n} \\
&\lesssim N^{-2s/d}\mathcal{R}^{2\beta\varsigma/d} + \frac{N \log^2 N(\log^2 N + \log n)}{n} + \frac{1}{n}.
\end{aligned}
$$

Let us set $N = n^{\frac{s}{2s+d}} \mathcal{R}^{\frac{2\beta\varsigma}{2s+d}}$. Then, if $N \geq \mathcal{R}^{-d/s}$, we have

$$\mathbb{E}_{D_n}\left[\left\|\widehat{f} - f^\circ\right\|^2_{L^2(P_X)}\right] \lesssim n^{-\frac{2s}{2s+d}} \mathcal{R}^{\frac{2\beta\varsigma}{2s+d}} \log^4(n),$$

which completes the proof. $\square$

# B. Proofs for Section 3

In this section, we give the proofs for the theorems in Section 3.

## B.1. Optimal Policy of $\chi^2$-regularized Objective

Before discussing the separation between NN-based estimators, we derive the optimal solution of $\min_{\pi(\cdot|x)} \mathcal{L}_{\chi^2}(r)(x)$. Its expression is already given in Huang et al. (2025b), we provide its deviation to make the proceeding discussion clear.

Recall that the definition of $\chi^2$-divergence $\chi^2(\cdot\|\cdot)$ is given by

$$\chi^2(q\|q_0) := \int \frac{1}{2}\left(\left(\frac{q(x)}{q_0(x)}\right)^2 - 1\right) \cdot q_0(x)\mathrm{d}x$$

$$= \frac{1}{2}\int \frac{(q(x) - q_0(x))^2}{q_0(x)}\mathrm{d}x = \frac{1}{2}\left(\int \frac{q(x)^2}{q_0(x)}\mathrm{d}x - 1\right).$$

Then, the $\chi^2$-regularized objective

$$\max_\pi \mathcal{L}_{\chi^2}(\pi; r)(x), \quad \text{where} \quad \mathcal{L}_{\chi^2}(\pi; r)(x) := \mathbb{E}_{y\sim\pi}[r(x, y)] - \mu \cdot \chi^2(\pi(\cdot \mid x)\|\pi_{\mathrm{ref}}(\cdot \mid x)),$$

is equivalent to

$$\max_\pi \mathcal{L}(\pi; r)(x), \quad \text{where} \quad \mathcal{L}(\pi; r)(x) := \mathbb{E}_{y\sim\pi(\cdot|x)}[r(x, y)] - \frac{\mu}{2} \cdot \int \frac{\pi(y \mid x)^2}{\pi_{\mathrm{ref}}(y \mid x)}\mathrm{d}y.$$

Let $q$ be a density defined as $q_x(y) := \frac{\pi(y|x)}{\pi_{\mathrm{ref}}(y|x)}$. Then, the optimization problem is reduced to

$$\max_{q_x} L_x(q_x), \quad \text{where} \quad L_x(q_x) := \mathbb{E}_{y\sim\pi_{\mathrm{ref}}}[q_x(y)r(x, y)] - \frac{\mu}{2} \cdot \mathbb{E}_{y\sim\pi_{\mathrm{ref}}(\cdot|x)}[q_x(y)^2]. \tag{12}$$

The objective $L_x$ can be rewritten as follows:

$$L_x(q) = -\frac{\mu}{2} \cdot \mathbb{E}_{y\sim\pi_{\mathrm{ref}}(\cdot|x)}\left[\left(q(y) - \frac{r(x, y)}{\mu}\right)^2\right] + \frac{\mathbb{E}_{y\sim\pi_{\mathrm{ref}}(\cdot|x)}[r(x, y)^2]}{2\mu}$$

$$= -\frac{\mu}{2}\left\|q - \mu^{-1}r_x\right\|^2_{L^2(\pi_{\mathrm{ref}}(\cdot|x))} + \frac{\|r_x\|^2_{L^2(\pi_{\mathrm{ref}})}}{2\mu},$$

where $r_x := r(x, \cdot)$. We define $q^*$ as the optimal solution of $\max_q L_x(q)$. Let us get $q^*$ explicitly. KKT condition implies

$$\begin{cases} \text{(a)} & q^*(y) - \mu^{-1} \cdot r_x(y) + \xi(y) \cdot (-1) + \lambda \cdot 1 = 0, \\ \text{(b)} & q^*(y) \geq 0, \\ \text{(c)} & \mathbb{E}_{y\sim\pi_{\mathrm{ref}}(\cdot|x)}[q^*(y)] = 1, \\ \text{(d)} & \xi(y) \geq 0 \\ \text{(e)} & \xi(y) \cdot (-q^*(y)) = 0, \end{cases}$$

where $\xi : [0, 1]^{d_Y} \to \mathbb{R}$ and $\zeta \in \mathbb{R}$ are slack variables corresponding to the constraints $q(y) \geq 0$ and $\mathbb{E}_{y\sim\pi_{\mathrm{ref}}(\cdot|x)}[q(y)] = 1$, respectively. The item (a) implies that

$$q^*(y) = \mu^{-1} \cdot r_x(y) - \lambda + \xi(y).$$

If $y \in \Omega_Y$ satisfies $q^*(y) > 0$, it holds $\xi(y) = 0$ by (e), and we have

$$q^*(y) = \mu^{-1} \cdot r_x(y) - \lambda.$$

Otherwise, by (b), we have $q^*(y) = 0$, and we have

$$\xi^*(y) = \lambda - \mu^{-1} \cdot r_x(y),$$

Then, (d) implies $\mu^{-1} \cdot r_x(y) \leq \lambda$. Summarizing above, we have

$$\begin{cases} q^*(y) > 0 & \Rightarrow & q^*(y) = \mu^{-1} \cdot r_x(y) - \lambda, \\ q^*(y) \leq 0 & \Rightarrow & \mu^{-1} \cdot r_x(y) \leq \lambda. \end{cases}$$

which implies

- If $\mu^{-1} \cdot r_x(y) > \lambda$, we have $q^*(y) > 0$, which implies $q^*(y) = \mu^{-1} \cdot r_x(y) - \lambda$.

- Otherwise (i.e., $\mu^{-1} \cdot r_x(y) \leq \lambda$), we have $0 \leq q^*(y) \leq \xi(y)$, and (e) implies $q^*(y) = 0$.

This indicates that

$$q^*(y) = [\mu^{-1} \cdot r(x, y) - \lambda_x]_+,$$

where the constant $\lambda_x$ is determined to satisfy (c), i.e,

$$\mathbb{E}_{y \sim \pi_{\mathrm{ref}}}[[\mu^{-1} \cdot r(x, y) - \lambda_x]_+] = 1.$$

### B.2. Proof of Theorem 3

Next, we prove Theorem 3. The proof technique is similar to that of Theorem 3.2 in Zhao et al. (2026): we reduce the regularized loss with a strongly convex divergence to the reward estimation error.

*Proof.* Let $\mathcal{H}_x := L^2(\pi_{\mathrm{ref}}(\cdot \mid x))$, and for $f, g \in \mathcal{H}_x$, we define

$$\langle f, g \rangle_{\mathcal{H}_x} := \int f(y) g(y) \pi_{\mathrm{ref}}(y \mid x) \, \mathrm{d}y,$$

$$\|f\|_{\mathcal{H}_x} := \sqrt{\langle f, f \rangle_{\mathcal{H}_x}}.$$

Then, the objective $L_x$ defined in (12) can be rewritten as

$$L_x(q) = \langle q, r_x \rangle_{\mathcal{H}_x} - \frac{\mu}{2} \|q\|_{\mathcal{H}_x}.$$

We define $L_x^\circ$ and $\widehat{L}_x$ as

$$L_x^\circ(q) := \langle q, r_x^\circ \rangle_{\mathcal{H}_x} - \frac{\mu}{2} \|q\|_{\mathcal{H}_x}, \quad \widehat{L}_x(q) := \langle q, \widehat{r}_x \rangle_{\mathcal{H}_x} - \frac{\mu}{2} \|q\|_{\mathcal{H}_x}.$$

Let $\mathcal{Q} := \{q \in \mathcal{H}_x \mid q \geq 0, \mathbb{E}_{y \sim \pi_{\mathrm{ref}}(\cdot \mid x)}[q(y)] = 1\}$ be the set of densities in $\mathcal{H}$. We define the map $V : \mathcal{H}_x \to \mathbb{R}$ by

$$V(r) := \max_{q \in \mathcal{Q}} L_x(q) = \max_{q \in \mathcal{Q}} \left( \langle q, r \rangle_{\mathcal{H}_x} - \frac{\mu}{2} \|q\|_{\mathcal{H}_x}^2 \right).$$

Then, we have

$$L_x^\circ(q) := V(r^\circ).$$

Let $g : \mathcal{H}_x \to \mathbb{R}$ be a map defined as

$$g(q) = \frac{\mu}{2} \|q\|_{\mathcal{H}_x}^2 + \mathbb{1}_{\mathcal{Q}}(q),$$

where $\mathbb{1}_{\mathcal{Q}}$ is a indicator function, and $g^*$ be its Fenchel's duality. Then, we have

$$
\begin{aligned}
g^*(r) &:= \sup_{q \in \mathcal{H}_x} \left\{ \langle q, r \rangle_{\mathcal{H}_x} - g(q) \right\} \\
&= \sup_{q \in \mathcal{H}_x} \left\{ \langle q, r \rangle_{\mathcal{H}_x} - \frac{\mu}{2} \|q\|_{\mathcal{H}_x}^2 - \mathbb{1}_{\mathcal{Q}}(q) \right\} \\
&= \sup_{q \in \mathcal{Q}} \left\{ \langle q, r \rangle_{\mathcal{H}_x} - \frac{\mu}{2} \|q\|_{\mathcal{H}_x}^2 \right\} \\
&= V(r).
\end{aligned}
$$

Since $g$ is $\mu$-strong convex, its Fenchel's dual $g^* = V$ is differentiable, and its gradient is $\mu^{-1}$-Lipschitz. Therefore, for $r \in \mathcal{H}_x$ and $q_r := \arg\min_{q \in \mathcal{H}} g(q) = \arg\min_{q \in \mathcal{Q}} g(q)$, we have $\nabla V(r) = q_r$, and it is $\mu^{-1}$-Lipschitz. In particular, for $\widehat{q} := q_{\widehat{r}}$, we have $\nabla V(\widehat{r}) = \widehat{q}$, and

$$V(r^\circ) \le V(\widehat{r}) + \langle \nabla V(\widehat{r}), r^\circ - \widehat{r} \rangle_{\mathcal{H}_x} + \frac{1}{2\mu}\|\widehat{r} - r^\circ\|_{\mathcal{H}_x}^2$$

$$\le V(\widehat{r}) + \langle \widehat{q}, r^\circ - \widehat{r} \rangle_{\mathcal{H}_x} + \frac{1}{2\mu}\|\widehat{r} - r^\circ\|_{\mathcal{H}_x}^2.$$

Since it holds $L_x^\circ(q^\circ) = V(r^\circ)$ and

$$V(\widehat{r}) = \langle \widehat{q}, \widehat{r} \rangle_{\mathcal{H}_x} - \frac{\mu}{2}\|\widehat{q}\|_{\mathcal{H}_x}^2$$

$$= \langle \widehat{q}, r^\circ \rangle_{\mathcal{H}_x} - \frac{\mu}{2}\|\widehat{q}\|_{\mathcal{H}_x}^2 + \langle \widehat{q}, \widehat{r} - r^\circ \rangle_{\mathcal{H}_x}$$

$$= L_x^\circ(\widehat{q}) + \langle \widehat{q}, \widehat{r} - r^\circ \rangle_{\mathcal{H}_x},$$

we have

$$L_x^\circ(q^\circ) - L_x^\circ(\widehat{q}) = V(r^\circ) - V(\widehat{r}) - \langle \widehat{q}, \widehat{r} - r^\circ \rangle$$

$$\le \frac{1}{2\mu}\|\widehat{r}_x - r_x^\circ\|_{\mathcal{H}_x}^2$$

$$\le \frac{\pi_{\max}}{2\mu}\|\widehat{r}(x,\cdot) - r^\circ(x,\cdot)\|_2^2.$$

Therefore, we have

$$\overline{\mathcal{L}}(r^\circ) - \overline{\mathcal{L}}(\widehat{r}) = \int_{\Omega_X} (L_x^\circ(q_x^\circ) - L_x^\circ(\widehat{q}_x))\,\mathrm{d}x$$

$$\le \frac{\pi_{\max}}{2\mu} \int_{\Omega_X} \|\widehat{r}(x,\cdot) - r^\circ(x,\cdot)\|_2^2\,\mathrm{d}x$$

$$= \frac{\pi_{\max}}{2\mu}\|\widehat{r} - r^\circ\|_2^2.$$

Theorem 2 in Suzuki (2019) implies

$$\overline{\mathcal{L}}(r^\circ) - \overline{\mathcal{L}}(\widehat{r}) \lesssim \mu^{-1} n^{-\frac{2s}{2s+d}} \log^2 n.$$

This completes the proof. $\qquad\square$

## B.3. Proof of Theorem 4

To prove Theorem 4, we first show the lower bound of estimation error for linear estimators. Specifically, we adapt the results for the standard Besov space (Theorem 1 in Zhang et al. (2002)) to our setting, which consider the Besov space with anisotropic integrability.

**Lemma 29.** *Let $X^n := (x_1, \ldots, x_n)$, $Y^n := (y_1, \ldots, y_n)$, and $\xi^n = (\xi_1, \ldots, \xi_n)$. For the class of linear estimators defined in (4), we have*

$$\inf_{\widehat{r}_{\mathrm{lin}}: linear} \sup_{r^\circ \in U(B_{p,q}^s)} \mathbb{E}_{X^n, Y^n, \xi^n}\left[\|\widehat{r}_{\mathrm{lin}} - r^\circ\|^2\right] \gtrsim n^{-\frac{2s-v}{2s-v+d}},$$

*for sufficiently large $n \in \mathbb{Z}_{>0}$, where $v := 2(d_Y/p - d/2)_+$.*

*Proof.* The proof strategy follows that of Theorem 1 in Zhang et al. (2002), which in turn is based on the argument of Donoho & Johnstone (1998). We define

$$R^*(\widehat{r}_{\mathrm{lin}}) := \sup_{r^\circ \in U(B_{p,q}^s)} \mathbb{E}_{X^n, Y^n, \xi^n}\left[\|\widehat{r}_{\mathrm{lin}} - r^\circ\|^2\right].$$

**Upper bound of** $\sum_i \varphi_i^2$ **using** $R^*(\widehat{r}_{\mathrm{lin}})$    Let $k$ be a non-negative integer which will be specified later. We define an event $\mathcal{E}$ as

$$\mathcal{E} := \left\{ \#\{i : (x_i, y_i) \in D_{\boldsymbol{j}}\} \leq C_1 \frac{n}{2^{kd}} \quad \text{for all } \boldsymbol{j} \in \mathcal{J}_k \right\},$$

for some constant $C_1 > 0$. That is, $\mathcal{E}$ is the event that no dyadic cube contains more than the average number of observation points up to a constant factor. Then, for sufficiently large $n$, we have $\mathbb{P}(\mathcal{E}) \geq 1/2$.

By the definition of linear estimators, we have

$$\mathbb{E}_{\xi^n}\left[(\widehat{r}_{\mathrm{lin}} - r^\circ)^2 \mid X^n, Y^n\right] = \left[\sum_{i=1}^n r^\circ(x_i, y_i)\varphi_i(x, y; X^n, Y^n) - r^\circ(x, y)\right]^2 + \sigma^2 \sum_{i=1}^n \varphi_i(x, y; X^n, Y^n)^2.$$

We take the integral over $(x, y) \in \Omega$, which yields

$$R^*(\widehat{r}_{\mathrm{lin}}; X^n, Y^n) := \mathbb{E}_{\xi^n}\left[\|\widehat{r}_{\mathrm{lin}} - r^\circ\|_{L^2}^2 \mid X^n, Y^n\right]$$

$$= \int_{\Omega_X} \int_{\Omega_Y} \left[\sum_{i=1}^n r^\circ(x_i, y_i)\varphi_i(x, y; X^n, Y^n) - r^\circ(x, y)\right]^2 \mathrm{d}y\, \mathrm{d}x + \sigma^2 \int_{\Omega_X} \int_{\Omega_Y} \sum_{i=1}^n \varphi_i(x, y; X^n, Y^n)^2 \,\mathrm{d}y\, \mathrm{d}x$$

$$\geq \sigma^2 \int_{\Omega_X} \int_{\Omega_Y} \sum_{i=1}^n \varphi_i(x, y; X^n, Y^n)^2 \,\mathrm{d}y\, \mathrm{d}x \,.$$

Therefore, we have

$$R^*(\widehat{r}_{\mathrm{lin}}) \geq \mathbb{P}(\mathcal{E})\mathbb{E}[R^*(\widehat{r}_{\mathrm{lin}}; X^n, Y^n) \mid \mathcal{E}]$$

$$\geq \frac{\sigma^2}{2} \int_{\Omega_X} \int_{\Omega_Y} \mathbb{E}\left[\sum_{i=1}^n \varphi_i(x, y; X^n, Y^n)^2 \,\middle|\, \mathcal{E}\right] \mathrm{d}y\, \mathrm{d}x \,.$$

Hence, there exists $\boldsymbol{j}^* \in \mathcal{J}_k$ such that

$$\frac{\sigma^2}{2} \int_{D_{\boldsymbol{j}^*}^X} \int_{D_{\boldsymbol{j}^*}^Y} \mathbb{E}\left[\sum_{i=1}^n \varphi_i(x, y; X^n, Y^n)^2 \,\middle|\, \mathcal{E}\right] \mathrm{d}y\, \mathrm{d}x \leq 2^{-kd} R^*(\widehat{r}_{\mathrm{lin}}),$$

where $D_{\boldsymbol{j}}^X$ and $D_{\boldsymbol{j}}^Y$ are the projections of $D_{\boldsymbol{j}}$ onto the $X$- and $Y$-coordinates, respectively, so that $D_{\boldsymbol{j}} = D_{\boldsymbol{j}}^X \times D_{\boldsymbol{j}}^Y$.

**Constructing** $r^\circ$ **and defining subsets** $\Omega_A, \Omega_B$ **on its support**    Suppose that $r^\circ = C_2 2^{-k(s-d_Y/p)} \cdot M_{k,\boldsymbol{j}^*}^d$ for some constant $C_2 > 0$. Then, using Theorem 20, we have

$$\|r^\circ\|_{B_{p,q}^s} \simeq 2^{k(s-d_Y/p)} \cdot C_2 2^{-k(s-d_Y/p)} = C_2.$$

Therefore, we have $r^\circ \in U(B_{p,q}^s)$ for sufficiently small $C_2 > 0$.

Since $M_{0,0}^d$ is a nonzero function, there exists $F > 0$ satisfying

$$\lambda\left(\{z \in \Omega : M_{0,0}^d(z) \geq F\}\right) \geq F.$$

Then, we define a set $\Omega_A \subseteq D_{\boldsymbol{j}^*}$ as

$$\Omega_A := \left\{z \in \Omega \,\middle|\, r^\circ(z) \geq C_2 2^{-k(s-d_Y/p)} \cdot F\right\}$$

Therefore, we have

$$\lambda(\Omega_A) \geq \lambda\left(\{z \in \Omega : M_{k,0}^d(z) \geq F\}\right)$$
$$\geq 2^{-kd}\lambda\left(\{z \in \Omega : M_{0,0}^d(z) \geq F\}\right)$$
$$\geq 2^{-kd} F.$$

Let $\Omega_B \subseteq D_{\boldsymbol{j}^*}$ be a set defined as

$$\Omega_B := \left\{ z \in D_{\boldsymbol{j}^*} \;\middle|\; \mathbb{E}\left[ \sum_{i=1}^n \varphi_i(x, y; X^n, Y^n)^2 \;\middle|\; \mathcal{E} \right] \leq \frac{4}{F\sigma^2} R^*(\widehat{r}_{\mathrm{lin}}) \right\}.$$

Then, Markov's inequality implies that, if $C_2$ is sufficiently large, we have

$$\lambda(\Omega_B) \geq 2^{-kd} - \frac{\int_{D_{\boldsymbol{j}^*}^X} \int_{D_{\boldsymbol{j}^*}^Y} \mathbb{E}\left[ \sum_{i=1}^n \varphi_i(x, y; X^n, Y^n)^2 \;\middle|\; \mathcal{E} \right] \mathrm{d}y\,\mathrm{d}x}{\frac{4}{F\sigma^2} R^*(\widehat{r}_{\mathrm{lin}})}$$

$$\geq 2^{-kd} - \frac{F}{2} 2^{-kd} = \left(1 - \frac{F}{2}\right) 2^{-kd}.$$

Therefore, we have

$$\lambda(\Omega_A \cap \Omega_B) = \frac{F}{2} 2^{-kd}.$$

**Lower bound of the difference between $\widehat{r}_{\mathrm{lin}}$ and $r^\circ$**  We have $\operatorname{supp} M_{k,\boldsymbol{j}^*}^d = \cup_{D \in \mathcal{D}'} D$ for some $\mathcal{D}' \subset \mathcal{D}$ with $|\mathcal{D}| = m^d$. Moreover, it holds

$$\|r^\circ\|_\infty = C_2 2^{-k(s - d_Y/p)} \|M_{k,\boldsymbol{j}}^d\|_\infty = C_2 2^{-k(s - d_Y/p)} \|\mathcal{K}_m\|_\infty^d.$$

Cauchy–Schwarz implies that, under the event $\mathcal{E}$, we have

$$\left(\mathbb{E}_{\xi^n}\left[\widehat{r}_{\mathrm{lin}}(x, y) \mid X^n, Y^n\right]\right)^2 = \left( \sum_{i=1}^n r^\circ(x_i, y_i) \varphi_i(x, y; X^n, Y^n) \right)^2$$

$$= \left( \sum_{i:(x_i, y_i) \in \operatorname{supp} M_{k,\boldsymbol{j}^*}^d} r^\circ(x_i, y_i) \varphi_i(x, y; X^n, Y^n) \right)^2$$

$$\leq \left( \#\{i : (x_i, y_i) \in \operatorname{supp} M_{k,\boldsymbol{j}^*}^d\} \right) \cdot \sum_{i:(x_i, y_i) \in \operatorname{supp} M_{k,\boldsymbol{j}^*}^d} r^\circ(x_i, y_i)^2 \varphi_i(x, y; X^n, Y^n)^2$$

$$\leq C_1 \frac{m^d n}{2^{kd}} \cdot \|r^\circ\|_\infty^2 \sum_{i=1}^n \varphi_i(x, y; X^n, Y^n)^2$$

$$\leq C_1 C_2 \|\mathcal{K}_m\|_\infty^{2d} \cdot \frac{m^d n 2^{-2k(s - d_Y/p)}}{2^{kd}} \sum_{i=1}^n \varphi_i(x, y; X^n, Y^n)^2.$$

Taking expectation over the event $\mathcal{E}$, we have

$$\mathbb{E}\left[ \left(\mathbb{E}_{\xi^n}\left[\widehat{r}_{\mathrm{lin}}(x, y) \mid X^n, Y^n\right]\right)^2 \;\middle|\; \mathcal{E} \right] \leq C_1 C_2 \|\mathcal{K}_m\|_\infty^{2d} \cdot \frac{m^d n 2^{-2k(s - d_Y/p)}}{2^{kd}} \mathbb{E}\left[ \sum_{i=1}^n \varphi_i(x, y; X^n, Y^n)^2 \;\middle|\; \mathcal{E} \right].$$

In particular, for $(x, y) \in \Omega_B$, we have

$$\mathbb{E}\left[ \left(\mathbb{E}_{\xi^n}\left[\widehat{r}_{\mathrm{lin}}(x, y) \mid X^n, Y^n\right]\right)^2 \;\middle|\; \mathcal{E} \right] \leq C_1 C_2 \|\mathcal{K}_m\|_\infty^{2d} \cdot \frac{m^d n 2^{-2k(s - d_Y/p)}}{2^{kd}} R^*(\widehat{r}_{\mathrm{lin}}).$$

Therefore, we have

$$\left\{ \mathbb{E}\left[ \left(\mathbb{E}_{\xi^n}\left[\widehat{r}_{\mathrm{lin}}(x, y) \mid X^n, Y^n\right] - r^\circ(x, y)\right)^2 \;\middle|\; \mathcal{E} \right] \right\}^{1/2} \geq |r^\circ(x, y)| - \left\{ \mathbb{E}\left[ \left(\mathbb{E}_{\xi^n}\left[\widehat{r}_{\mathrm{lin}}(x, y) \mid X^n, Y^n\right]\right)^2 \;\middle|\; \mathcal{E} \right] \right\}^{1/2}$$

$$\geq |r^\circ(x, y)| - C_3^{1/2} \cdot \frac{\sqrt{n} 2^{-k(s - d_Y/p)}}{2^{kd/2}} R^*(\widehat{r}_{\mathrm{lin}})^{1/2},$$

where we define $C_3 := C_1 C_2 \|\mathcal{K}_m\|_\infty^{2d} m^d$. In particular, for $(x, y) \in \Omega_A \cap \Omega_B$, we have

$$\left\{ \mathbb{E}\left[ \left(\mathbb{E}_{\xi^n}\left[\widehat{r}_{\mathrm{lin}}(x, y) \mid X^n, Y^n\right] - r^\circ(x, y)\right)^2 \;\middle|\; \mathcal{E} \right] \right\}^{1/2} \geq C_2 2^{-k(s - d_Y/p)} \cdot F - C_3^{1/2} \cdot \frac{\sqrt{n} 2^{-k(s - d_Y/p)}}{2^{kd/2}} R^*(\widehat{r}_{\mathrm{lin}})^{1/2}.$$

**Obtain the lower bound** Suppose that $R^*(\widehat{r}_{\lin}) < \frac{C_2^2 F^2}{4 C_3 n} 2^{kd}$. Then, we have $(C_3 n R^*(\widehat{r}_{\lin}))^{1/2} < \frac{C_2 F}{2} 2^{kd/2}$. Then, we have

$$\left\{ \mathbb{E}\left[ \left(\mathbb{E}_{\xi^n}\left[\widehat{r}_{\lin}(x, y) \mid X^n, Y^n\right] - r^\circ(x, y)\right)^2 \mid \mathcal{E} \right] \right\}^{1/2} \geq C_2 2^{-k(s - d_Y/p)} \cdot F - \frac{C_2 F}{2} 2^{-k(s - d_Y/p)}$$

$$= \frac{C_2 F}{2} 2^{-k(s - d_Y/p)}.$$

Therefore, it holds

$$R^*(\widehat{r}_{\lin}) \geq \int_{\Omega_A \cap \Omega_B} \mathbb{E}_{X^n, Y^n, \xi^n}\left[ \left(\widehat{r}_{\lin}(x, y) - r^\circ(x, y)\right)^2 \right] dz$$

$$\geq \int_{\Omega_A \cap \Omega_B} \mathbb{E}\left[ \left(\mathbb{E}_{\xi^n}\left[\widehat{r}_{\lin}(x, y) \mid X^n, Y^n\right] - r^\circ(x, y)\right)^2 \right] dz$$

$$\geq \int_{\Omega_A \cap \Omega_B} \mathbb{P}(\mathcal{E}) \mathbb{E}\left[ \left(\mathbb{E}_{\xi^n}\left[\widehat{r}_{\lin}(x, y) \mid X^n, Y^n\right] - r^\circ(x, y)\right)^2 \mid \mathcal{E} \right] dz$$

$$\geq \lambda(\Omega_A \cap \Omega_B) \cdot \left( \frac{C_2 F}{2} 2^{-k(s - d_Y/p)} \right)^2 = \frac{C_2^2 F^3}{16} 2^{-2k(s + d/2 - d_Y/p)}.$$

Hence, we have $R^*(\widehat{r}_{\lin}) \geq \frac{C_2^2 F^2}{4 C_3 n} 2^{kd}$ or $R^*(\widehat{r}_{\lin}) \geq \frac{C_2^2 F^3}{16} 2^{-2k(s + d/2 - d_Y/p)}$, which implies

$$R^*(\widehat{r}_{\lin}) \geq \min\left\{ \frac{C_2^2 F^2}{4 C_3 n} 2^{kd}, \frac{C_2^2 F^3}{16} 2^{-2k(s + d/2 - d_Y/p)} \right\}.$$

Choosing $k \in \mathbb{Z}_{\geq 0}$ to satisfy

$$\frac{2^{kd}}{n} \simeq 2^{-2k(s + d/2 - d_Y/p)}, \quad \text{i.e.,} \quad 2^k \simeq n^{\frac{1}{2(s + d - d_Y/p)}},$$

we have

$$R^*(\widehat{r}_{\lin}) \gtrsim n^{-\frac{2(s + d/2 - d_Y/p)}{2(s + d - d_Y/p)}} = n^{-\frac{2s - v}{2s + d - v}},$$

where $v := 2 d_Y/p - d = 2(d_Y/p - d/2)$. Combining the minimax optimal rate of estimation error, we obtain the assertion. $\qquad\square$

*Proof of Theorem 4.* We fix $\phi_1, \ldots, \phi_n$. Let $r^\star \in U(B_{p,q}^s(\Omega))$ that attains the supremum of $\sup_{r^\circ \in U(B_{p,q}^s(\Omega))} \mathbb{E}_{D^n}[\|\widehat{r} - r^\circ\|_2^2]$. Lemma 29 implies that

$$\mathbb{E}_{D^n}[\|\widehat{r} - r^\star\|_2^2] = \sup_{r^\circ \in U(B_{p,q}^s(\Omega))} \mathbb{E}_{D^n}[\|\widehat{r} - r^\circ\|_2^2] \gtrsim n^{-\frac{2s - v}{2s + d - v}}.$$

From the assumption, there exists $\delta \in (0, 1]$ and $c \in (0, 1]$ such that

$$\|\widehat{r} - r^\star\|_\infty \xrightarrow{\mathbb{P}} 0, \tag{13}$$

$$\mathbb{P}_{D^n}\left[ \|\widehat{r} - r^\star\|_2^2 \gtrsim \mathbb{E}_{D^n}[\|\widehat{r} - r^\star\|_2^2] \right] \geq \delta, \tag{14}$$

$$\mathbb{P}_{D^n}\left[ \left\| \mathbb{E}_{y \sim \pi_{\ref}(\cdot | x)}[r^\star(\cdot, y) - \widehat{r}(\cdot, y)] \right\|_{L^2(\Omega_X)}^2 \ll \mathbb{E}_{D^n}[\|\widehat{r} - r^\star\|_2^2] \right] \geq 1 - \delta/4. \tag{15}$$

Here, to derive (15), we used Markov's inequality.

(13) implies that, for sufficiently large $n \in \mathbb{Z}_{>0}$, we have

$$\mathbb{P}[\mathcal{A}_1] \geq 1 - \frac{\delta}{2}, \quad \text{where} \quad \mathcal{A}_1 := \left\{ \|\widehat{r} - r^\star\|_\infty \leq \frac{R}{\mu} \right\}.$$

Moreover, using (14), we have

$$\mathbb{P}[\mathcal{A}_2] \geq \delta, \quad \text{where} \quad \mathcal{A}_2 := \{\|\widehat{r} - r^\star\|_2 \gtrsim \mathbb{E}_{D^n}[\|\widehat{r} - r^\star\|_2^2]\}.$$

Furthermore, using (15), we have

$$\mathbb{P}[\mathcal{A}_3] \geq 1 - \frac{\delta}{4}, \quad \text{where} \quad \mathcal{A}_3 := \left\{ \left\|\mathbb{E}_{y \sim \pi_{\mathrm{ref}}(\cdot|x)}[r^\star(\cdot, y) - \widehat{r}(\cdot, y)]\right\|_{L^2(\Omega_X)}^2 \ll \mathbb{E}_{D^n}[\|\widehat{r} - r^\star\|_2^2] \right\}.$$

We get explicit forms of $q$ for $r', \check{r}$ defined as follows:

$$r' := \frac{\mu}{4R}(r^\star - R), \quad \check{r} := \frac{\mu}{4R}(\widehat{r} - R).$$

Then, for any $x, y$, we have $r'(x, y), \check{r}(x, y) \in [0, 1/2]$. Then it holds $\mu^{-1} \int_{y \in \Omega_Y} r'(x, y)\, \mathrm{d}y \leq 1/2 < 1$, which yields

$$\lambda^\star(x) = \mu^{-1} \int_{y \in \Omega_Y} r'(x, y)\, \mathrm{d}y - 1, \quad \text{and} \quad q^\circ(x, y) = \mu^{-1} r'(x, y) - \lambda^\star(x).$$

Additionally, under the event $\mathcal{A}_1$, we have

$$\|\check{r} - r'\|_\infty \leq \frac{\mu}{4R}\|r^\circ - \widehat{r}\|_\infty \leq \frac{1}{4}.$$

Therefore, we have

$$\mu^{-1} \int_{y \in \Omega_Y} \check{r}(x, y)\, \mathrm{d}y \leq \int_{y \in \Omega_Y} \mu^{-1}\check{r}(x, y)\, \mathrm{d}y + \|\check{r} - r'\|_\infty \leq \frac{3}{4},$$

which yields

$$\lambda'(x) = \mu^{-1} \int_{y \in \Omega_Y} \check{r}(x, y)\, \mathrm{d}y - 1, \quad \text{and} \quad \widehat{q}(x, y) = \mu^{-1}\check{r}(x, y) - \lambda'(x).$$

We next evaluate the value of objective value and their difference. We have

$$L_x^\circ(q_x^\circ) = -\frac{\mu}{2} \cdot (-\lambda^\star(x))^2 + \frac{\|r'(x, \cdot)\|^2}{2\mu}$$

$$= -\frac{\mu}{2} \cdot (\lambda^\star(x))^2 + \frac{\|r'(x, \cdot)\|^2}{2\mu},$$

$$L_x^\circ(\widehat{q}_x) = -\frac{\mu}{2} \cdot \|\mu^{-1}\check{r}(x, \cdot) - \lambda'(x) - \mu^{-1}r'(x, \cdot)\|^2 + \frac{\|r'(x, \cdot)\|^2}{2\mu}$$

$$= -\frac{\mu}{2} \cdot (\lambda'^2 - 2\mu^{-1}\lambda'\mathbb{E}_y[\check{r}(x, y) - r'(x, y)] + \mu^{-2}\|\check{r}(x, \cdot) - r'(x, \cdot)\|^2) + \frac{\|r'(x, \cdot)\|^2}{2\mu}.$$

Therefore, we have

$$L_x^\circ(q_x^\circ) - L_x^\circ(\widehat{q}_x)$$
$$= -\frac{\mu}{2}\left((\lambda^\star(x))^2 - \lambda'^2(x) + 2\mu^{-1}\lambda'(x) \cdot \mathbb{E}_y[\check{r}(x, y) - r'(x, y)] - \mu^{-2}\|\check{r}(x, \cdot) - r'(x, \cdot)\|^2\right)$$
$$= \frac{\mu}{2}(\lambda'^2(x) - (\lambda^\star(x))^2) - \lambda'(x) \cdot \mathbb{E}_y[\check{r}(x, y) - r'(x, y)] + \frac{1}{2\mu}\|\check{r}(x, \cdot) - r'(x, \cdot)\|^2.$$

Now, it holds

$$\frac{\mu}{2}(\lambda'(x)^2 - (\lambda^\star(x))^2) = \frac{\mu}{2}(\lambda^\star(x) + \lambda'(x)) \cdot \mu^{-1}\mathbb{E}_y[\check{r}(x, y) - r'(x, y)]$$

$$= \frac{\lambda^\star(x) + \lambda'(x)}{2} \cdot \mathbb{E}_y[\check{r}(x, y) - r'(x, y)],$$

which implies

$$L_x^\circ(q_x^\circ) - L_x^\circ(\widehat{q}_x) = \frac{\lambda^\star - \lambda'}{2} \cdot \mathbb{E}_y[\check{r}(x,y) - r'(x,y)] + \frac{1}{2\mu}\|\check{r}(x,\cdot) - r'(x,\cdot)\|^2$$

$$= \frac{1}{2\mu}\Big\{ \|\check{r}(x,\cdot) - r'(x,\cdot)\|^2 - (\mathbb{E}_y[\check{r}(x,y) - r'(x,y)])^2 \Big\}.$$

Taking expectation over $x \sim P_X$, we have

$$\overline{\mathcal{L}}(r') - \overline{\mathcal{L}}(\check{r}) = \frac{1}{2\mu}\Big\{ \|\check{r} - r'\|^2 - \mathbb{E}_x\Big[ (\mathbb{E}_y[\check{r}(x,y) - r'(x,y)])^2 \Big] \Big\}$$

$$= \frac{1}{2\mu}\frac{\mu^2}{16R^2}\Big\{ \|\widehat{r} - r^\star\|^2 - \big\|\mathbb{E}_{y\sim\pi_{\mathrm{ref}}(\cdot|x)}[r^\star(\cdot,y) - \widehat{r}(\cdot,y)]\big\|^2_{L^2(\Omega_X)} \Big\}.$$

Under the event $\mathcal{A}_2 \cap \mathcal{A}_3$, we have

$$\overline{\mathcal{L}}(r') - \overline{\mathcal{L}}(\check{r}) \gtrsim \mathbb{E}_{D^n}[\|\widehat{r} - r^\star\|_2^2] - \big\|\mathbb{E}_{y\sim\pi_{\mathrm{ref}}(\cdot|x)}[r^\star(\cdot,y) - \widehat{r}(\cdot,y)]\big\|^2_{L^2(\Omega_X)}.$$

Using (15), we have

$$\overline{\mathcal{L}}(r') - \overline{\mathcal{L}}(\check{r}) \gtrsim \mathbb{E}_{D^n}[\|\widehat{r} - r^\star\|_2^2].$$

Since it holds that

$$\mathbb{P}(\mathcal{A}_1 \cap \mathcal{A}_2 \cap \mathcal{A}_3) \geq \mathbb{P}(\bar{\mathcal{A}}_1) - \mathbb{P}(\bar{\mathcal{A}}_2) - \mathbb{P}(\bar{\mathcal{A}}_3) \geq \delta - \frac{\delta}{2} - \frac{\delta}{4} \geq \frac{\delta}{4},$$

we have

$$\mathbb{E}_{D^n}[\overline{\mathcal{L}}(r^\star) - \overline{\mathcal{L}}(\check{r})] \gtrsim \mathbb{P}(\mathcal{A}_1 \cap \mathcal{A}_2 \cap \mathcal{A}_3) \cdot \mathbb{E}_{D^n}[\|\widehat{r} - r^\star\|_2^2]$$

$$\gtrsim \mathbb{E}_{D^n}[\|\widehat{r} - r^\star\|_2^2].$$

Since $\check{r}$ is a linear estimator of $r^\star$, this gives the assertion. $\qquad\square$

## B.4. Assumptions for Linear Estimators in Theorem 4

In this subsection, we discuss the validity of the assumptions in Theorem 4.

We first note that the condition (a) $\|\widehat{r}_{\mathrm{lin}} - r^\circ\|_\infty \xrightarrow{\mathbb{P}} 0$ $(n \to \infty)$ is satisfied for many linear estimators, including Nadaraya–Watson estimator, $k$-nearest neighbor estimator, Sieve estimator with spline basis and kernel ridge estimator with Matérn kernel satisfies this condition. See, for example, Hansen (2008); Jiang (2019); Chen & Christensen (2015); Tuo et al. (2020) for the details.

Next, we discuss the condition (b) under a specific setting. To do that, we introduce the bias-variance decomposition of linear estimators. We consider a linear estimator defined as $\widehat{r}(x,y) = \sum_{i=1}^n r_i^\dagger \phi_i(x,y;X^n,Y^n)$. For the simplicity of notation, we define $\Phi_i(x,y) := \phi_i(x,y;X^n,Y^n)$. First, we decompose the risk $R(X^n,Y^n) := \mathbb{E}_{\xi^n}[\|\widehat{r} - r^\circ\|_2^2]$ as follows:

$$\mathbb{E}_{\xi^n}[\|\widehat{r} - r^\circ\|_2^2]$$

$$= \mathbb{E}_{\xi^n}\left[ \left\| \left( \sum_{i=1}^n r^\circ(x_i,y_i)\Phi_i - r^\circ \right) + \sum_{i=1}^n \xi_i \Phi_i \right\|^2_{L^2(\Omega)} \right]$$

$$= \left\| \sum_{i=1}^n r^\circ(x_i,y_i)\Phi_i - r^\circ \right\|^2_{L^2(\Omega)} + \mathbb{E}_{\xi^n}\left[ \left\langle \sum_{i=1}^n r^\circ(x_i,y_i)\Phi_i - r^\circ, \sum_{i=1}^n \xi_i \Phi_i \right\rangle_{L^2(\Omega)} \right] + \mathbb{E}_{\xi^n}\left[ \left\| \sum_{i=1}^n \xi_i \Phi_i \right\|^2_{L^2(\Omega)} \right]$$

$$= \left\| \sum_{i=1}^n r^\circ(x_i,y_i)\Phi_i - r^\circ \right\|^2_{L^2(\Omega)} + \left\langle \sum_{i=1}^n r^\circ(x_i,y_i)\Phi_i - r^\circ, \sum_{i=1}^n \mathbb{E}_{\xi_i}[\xi_i]\Phi_i \right\rangle_{L^2(\Omega)} + \mathbb{E}_{\xi^n}\left[ \sum_{i=1}^n \xi_i^2 \|\Phi_i\|^2_{L^2(\Omega)} \right]$$

$$= \left\| \sum_{i=1}^n r^\circ(x_i,y_i)\Phi_i - r^\circ \right\|^2_{L^2(\Omega)} + \sigma^2 \sum_{i=1}^n \|\Phi_i\|^2_{L^2(\Omega)},$$

where the expectation is conditioned on $X^n$ and $Y^n$, which we omitted here. By taking expectation over $X^n$ and $Y^n$, we have

$$R^* := \mathbb{E}_{\xi^n, X^n, Y^n}\left[\|\widehat{r} - r^\circ\|_2^2\right] = \underbrace{\mathbb{E}_{X^n, Y^n}\left[\left\|\sum_{i=1}^n r^\circ(x_i, y_i)\Phi_i - r^\circ\right\|_{L^2(\Omega)}^2\right]}_{\text{Bias}} + \underbrace{\sigma^2 \mathbb{E}_{\xi^n, X^n, Y^n}\left[\sum_{i=1}^n \|\Phi_i\|_{L^2(\Omega)}^2\right]}_{\text{Variance}}. \quad (16)$$

It is common to set the hyperparameters to balance the bias and variance, i.e.,

$$R^* = \mathbb{E}_{X^n, Y^n}[R(X^n, Y^n)] \simeq \mathbb{E}_{X^n, Y^n}\left[\left\|\sum_{i=1}^n r^\circ(x_i, y_i)\Phi_i - r^\circ\right\|_{L^2(\Omega)}^2\right] \simeq \mathbb{E}_{X^n, Y^n}\left[\sum_{i=1}^n \|\Phi_i\|_{L^2(\Omega)}^2\right].$$

Based on this, we suppose that there exists a constant $C > 0$ such that

$$R^* \leq C \cdot \mathbb{E}_{X^n, Y^n}\left[\sum_{i=1}^n \|\Phi_i\|_{L^2(\Omega)}^2\right]. \quad (17)$$

Now, we assume that the following condition holds:

$$\mathbb{E}_{\xi^n, X^n, Y^n}\left[\left(\sum_{i=1}^n \|\Phi_i\|_{L^2(\Omega)}^2\right)^2\right] \leq c\left(\mathbb{E}_{\xi^n, X^n, Y^n}\left[\sum_{i=1}^n \|\Phi_i\|_{L^2(\Omega)}^2\right]\right)^2, \quad (18)$$

for some constant $c > 0$. This can be validated, for $k$-NN estimators, for example. Indeed, for $k$-NN estimators, the map $\phi_i$ can be written as

$$\phi_i(x, y; X^n, Y^n) = \frac{1}{k}\mathbb{1}_{R_i}(x, y),$$

where $R_i := \{(x, y) \in \Omega \mid i \in N_k(x; X^n)\}$ and $N_k(x; X^n)$ is $k$-nearest neighbors of $x$. Then, we have

$$\|\Phi_i\|_{L^2(\Omega)}^2 = \int_\Omega \frac{1}{k^2}\mathbb{1}_{R_i}(u)\,\mathrm{d}u = \frac{\lambda(R_i)}{k^2}.$$

Therefore, we have

$$\begin{aligned}
\sum_{i=1}^n \|\Phi_i\|_{L^2(\Omega)}^2 &= \sum_i \frac{\lambda(R_i)}{k^2} \\
&= \frac{1}{k^2}\int_\Omega \sum_i (1 \text{ if } i \in N_k(x; X^n) \text{ else } 0)\,\mathrm{d}x \\
&= \int_\Omega k\,\mathrm{d}x = k, \quad \text{(a.s.)}.
\end{aligned}$$

This implies that (18) holds with $c = 1$.

Under (18), using Paley–Zygmund inequality, for any $c_0 \in (0, 1)$, we have

$$\mathbb{P}\left[\sum_{i=1}^n \|\Phi_i\|_{L^2(\Omega)}^2 \geq (1 - c_0)\mathbb{E}_{X^n, Y^n}\left[\sum_{i=1}^n \|\Phi_i\|_{L^2(\Omega)}^2\right]\right] \geq c_0^2 \frac{\left(\mathbb{E}_{X^n, Y^n}\left[\sum_{i=1}^n \|\Phi_i\|_{L^2(\Omega)}^2\right]\right)^2}{\mathbb{E}_{X^n, Y^n}\left[\left(\sum_{i=1}^n \|\Phi_i\|_{L^2(\Omega)}^2\right)^2\right]} \geq cc_0^2.$$

Combining with (16) and (17), we have

$$\begin{aligned}
\mathbb{P}\left[R(X^n, Y^n) \geq \frac{(1 - c_0)R^*}{C}\right] &\geq \mathbb{P}\left[\sum_{i=1}^n \|\Phi_i\|_{L^2(\Omega)}^2 \geq \frac{(1 - c_0)R^*}{C}\right] \\
&\geq \mathbb{P}\left[\sum_{i=1}^n \|\Phi_i\|_{L^2(\Omega)}^2 \geq (1 - c_0)\mathbb{E}_{X^n, Y^n}\left[\sum_{i=1}^n \|\Phi_i\|_{L^2(\Omega)}^2\right]\right] \\
&\geq cc_0^2.
\end{aligned}$$

Next, we can bound the variance of $\|\widehat{r} - r^\circ\|_2^2$ conditioned on $X^n, Y^n$ can be calculated as

$$\mathbb{V}(\|\widehat{r} - r^\circ\|_2^2 \mid X^n, Y^n) = 2 \sum_{i=1}^n \sum_{j=1}^n \langle \Phi_i, \Phi_j \rangle^2 + 4 \sum_{i=1}^n \left\langle \Phi_i, f - \sum_{j=1}^n f(x_j)\Phi_j \right\rangle^2.$$

Each term can be upper-bounded as follows:

$$\sum_{i=1}^n \sum_{j=1}^n \langle \Phi_i, \Phi_j \rangle^2 \leq \left( \sum_i \|\Phi_i\|^2 \right)^2 \leq R(X^n, Y^n)^2,$$

$$\sum_{i=1}^n \left\langle \Phi_i, f - \sum_{j=1}^n f(x_j)\Phi_j \right\rangle^2 \leq \left( \sum_i \|\Phi_i\|_2^2 \right) \left\| f - \sum_{j=1}^n f(x_j)\Phi_j \right\|_2^2$$

$$\leq \frac{1}{4} \left( \sum_i \|\Phi_i\|_2^2 + \left\| f - \sum_{j=1}^n f(x_j)\Phi_j \right\|_2 \right)^2$$

$$= \frac{1}{4} R(X^n, Y^n)^2.$$

This yields

$$\mathbb{V}(\|\widehat{r} - r^\circ\|_2^2 \mid X^n, Y^n) \leq 3R(X^n, Y^n)^2.$$

Therefore, using Cantelli's inequality, for any $c_1 \in (0, 1)$, we have

$$\mathbb{P}\big[\|\widehat{r} - r^\circ\|_2^2 \leq (1 - c_1)R(X^n, Y^n)\big] = \mathbb{P}\big[\|\widehat{r} - r^\circ\|_2^2 - \mathbb{E}[\|\widehat{r} - r^\circ\|_2^2 \mid X^n, Y^n] \leq -c_1 R(X^n, Y^n)\big]$$

$$\leq \frac{\mathbb{V}(\|\widehat{r} - r^\circ\|_2^2 \mid X^n, Y^n)}{\mathbb{V}(\|\widehat{r} - r^\circ\|_2^2 \mid X^n, Y^n) + (c_1 R(X^n, Y^n))^2}$$

$$\leq \frac{3R(X^n, Y^n)^2}{3R(X^n, Y^n)^2 + c_1^2 R(X^n, Y^n)^2}$$

$$= \frac{3}{3 + c_1^2}.$$

Combining above, we have

$$\mathbb{P}\left[\|\widehat{r} - r^\circ\|_2^2 \geq \frac{(1 - c_1)(1 - c_0)R^*}{C}\right] \geq cc_0^2 - \frac{3}{3 + c_1^2},$$

if the right-hand side is positive. If $c_0 \to 1, c_1 \to 1$, the right-hand side converges to $c - \frac{3}{4}$. To summarize, if the bias and the variance are balanced, and Equation (18) holds with $c \in (3/4, 1]$, we see that $\|\widehat{r} - r^\circ\|_2^2 \gtrsim R^*$ with constant probability. We leave it to future work to check whether the assumption in Theorem 4 holds in other configurations.

## C. Proofs for Section 4

In this section, we prove Theorem 6.

First, we prove Lemma 8, which constructs a comparator policy that attains the $\epsilon$-optimal reward for $\epsilon > 0$ with bounded coverage from the reference policy $\pi_{\text{ref}}$.

*Proof of Lemma 8.* If we set the policy $\pi_\epsilon^*$ as

$$\pi_\epsilon^*(y \mid x) := \frac{\mathbb{1}_{S_\epsilon(x)}(y)}{\lambda(S_\epsilon(x))}$$

then the two conditions are satisfied. Indeed, the condition (i) is satisfied since

$$
\mathbb{E}_{x \sim P_X}\left[r^*(x) - \mathbb{E}_{y \sim \pi_\epsilon^*(\cdot|x)}[r^\circ(x,y)]\right] = \mathbb{E}_{x \sim P_X}\left[\mathbb{E}_{y \sim \pi_\epsilon^*(\cdot|x)}[r^*(x) - r^\circ(x,y)]\right]
$$
$$
\leq \mathbb{E}_{x \sim P_X}\left[\epsilon \cdot \frac{1}{\lambda(S_\epsilon(x))} \cdot \lambda(S_\epsilon(x))\right]
$$
$$
= \epsilon,
$$

and the condition (ii) is satisfied since

$$
\mathcal{C}(\pi_\epsilon^*, \pi_{\mathrm{ref}}) = \mathbb{E}_{x \sim P_X}\left[\mathbb{E}_{y \sim \pi_\epsilon^*(\cdot|x)}\left[\frac{\pi_\epsilon^*(y\mid x)}{\pi_{\mathrm{ref}}(y\mid x)}\right]\right] = \mathbb{E}_{x \sim P_X}\left[\int \frac{\pi_\epsilon^*(y\mid x)^2}{\pi_{\mathrm{ref}}(y\mid x)}\,\mathrm{d}y\right]
$$
$$
\leq \mathbb{E}_{x \sim P_X}\left[\int \frac{1/\lambda(S_\epsilon(x))^2}{\pi_{\min}}\,\mathrm{d}y\right] \lesssim \mathbb{E}_{x \sim P_X}\left[\frac{1}{\lambda(S_\epsilon(x))}\right] \lesssim \epsilon^{-\gamma}.
$$

This completes the proof. □

Now, we prove Theorem 6.

*Proof.* As in the proof sketch in Section 4, we have

$$
\mathbb{E}_{D^n}\left[\mathbb{E}_{x \sim P_X}\left[\mathbb{E}_{\widehat{y}_{\mathrm{NN}}(x) \sim \pi_{\mu,N}^{\mathrm{Pes}}}[r^*(x) - r^\circ(x, \widehat{y}_{\mathrm{NN}}(x))]\right]\right]
$$
$$
\lesssim \mathbb{E}_{x \sim P_X}\left[r^*(x) - \mathbb{E}_{y \sim \pi_\epsilon^*(\cdot|x)}[r^\circ(x,y)]\right] + \mathbb{E}_{D^n, x \sim P_X}\left[\mathbb{E}_{y \sim \pi_\epsilon^*(\cdot|x)}[r^\circ(x,y)] - \mathbb{E}_{y \sim \pi_{\mu,N}^{\mathrm{Pes}}(\cdot|x)}[r^\circ(x,y)]\right].
$$

As for the first term, Lemma 8 implies that

$$
\mathbb{E}_{x \sim P_X}\left[r^*(x) - \mathbb{E}_{y \sim \pi_\epsilon^*(\cdot|x)}[r^\circ(x,y)]\right] \leq \epsilon.
$$

Moreover, using Proposition 7, Theorem 26, and the coverage bound in Lemma 8, we can upper-upper bound the second term as

$$
\mathbb{E}_{D^n}\left[\mathbb{E}_{x \sim P_X}\left[\mathbb{E}_{y \sim \pi_\epsilon^*(\cdot|x)}[r^\circ(x,y)] - \mathbb{E}_{y \sim \pi_{\mu,N}^{\mathrm{Pes}}(\cdot|x)}[r^\circ(x,y)]\right]\right]
$$
$$
\lesssim \mu \cdot \mathbb{E}_{x \sim P_X}[\mathcal{C}(x; \pi^*, \pi_{\mathrm{ref}})] + \mu^{-1} \cdot \mathbb{E}_{D^n}[\mathbb{E}_{x \sim P_X}[\mathcal{E}^2(x)]] + \mu^{-1}\mathbb{E}_{D^n}[\mathbb{E}_{x \sim P_X}[\mathcal{E}(x)]] \cdot \mathrm{e}^{-\frac{\mu N}{c(R+\mu)}}
$$
$$
\leq \mu \cdot \mathbb{E}_{x \sim P_X}[\mathcal{C}(x; \pi^*, \pi_{\mathrm{ref}})] + \mu^{-1} \cdot \mathbb{E}_{D^n}[\mathbb{E}_{x \sim P_X}[\mathcal{E}^2(x)]] + \mu^{-1}\left(\mathbb{E}_{D^n}[\mathbb{E}_{x \sim P_X}[\mathcal{E}^2(x)]]\right)^{1/2} \cdot \mathrm{e}^{-\frac{\mu N}{c(R+\mu)}}
$$
$$
\lesssim \mu \cdot \epsilon^{-\gamma} + \mu^{-1} \cdot n^{-\frac{2s}{2s+d}}\log^4 n + \mu^{-1}\left(n^{-\frac{2s}{2s+d}}\log^4 n\right)^{1/2} \cdot \mathrm{e}^{-\frac{\mu N}{c(R+\mu)}}.
$$

Therefore, if we set $N \gtrsim \mu^{-1}\log n$, we have

$$
\mathbb{E}_{D^n}\left[\mathbb{E}_{x \sim P_X}\left[\mathbb{E}_{\widehat{y}_{\mathrm{NN}}(x) \sim \pi_{\mu,N}^{\mathrm{Pes}}}[r^*(x) - r^\circ(x, \widehat{y}_{\mathrm{NN}}(x))]\right]\right] \lesssim \epsilon + \mu \cdot \epsilon^{-\gamma} + \mu^{-1} \cdot n^{-\frac{2s}{2s+d}}\log^4 n.
$$

To minimize the right-hand side, we set

$$
\begin{cases}
\epsilon \simeq \mu \cdot \epsilon^{-\gamma} \\
\epsilon \simeq \mu^{-1} \cdot n^{-\frac{2s}{2s+d}}\log^4 n.
\end{cases}
$$

This is equivalent to

$$
\epsilon \simeq \left(n^{-\frac{2s}{2s+d}}\log^4 n\right)^{\frac{1}{2+\gamma}} = n^{-\frac{s}{2s+d}\frac{2}{2+\gamma}}\log^{\frac{4}{2+\gamma}} n,
$$
$$
\mu \simeq \epsilon^{1+\gamma} \simeq n^{-\frac{s}{2s+d}\frac{2(1+\gamma)}{2+\gamma}}\log^{\frac{4(1+\gamma)}{2+\gamma}} n.
$$

Hence, we have

$$
\mathbb{E}_{D^n}\left[\mathbb{E}_{x \sim P_X}\left[\mathbb{E}_{\widehat{y}_{\mathrm{NN}}(x) \sim \pi_{\mu,N}^{\mathrm{Pes}}}[r^*(x) - r^\circ(x, \widehat{y}_{\mathrm{NN}}(x))]\right]\right] \lesssim n^{-\frac{s}{2s+d}\frac{2}{2+\gamma}}\log^{\frac{4}{2+\gamma}} n,
$$

which completes the proof. □

# D. Smoothing over Prompts for $L^\infty$-approximation Guarantee

Remind that the definition of the smoothing kernel $K$ is given as follows:

$$K(x) := \sum_{j=1}^{r} \binom{r}{j} (-1)^{1-j} \frac{1}{j^{d_X}} \left(\frac{2}{\gamma^2 \pi}\right)^{d_X/2} K_{\frac{j\gamma}{\sqrt{2}}}(x),$$

where $K_\gamma(x) := \exp(-\gamma^{-2}\|x\|_2^2)$. In this section, we prove the following theorem, which demonstrates that we can bound prompt-uniform estimation error $\sup_{x \in \Omega_X^{[\epsilon]}} \|\widetilde{r}(x, \cdot) - r^\circ(x, \cdot)\|_{L^2(\Omega_Y^{[\epsilon']})}$ using the upper bound of $L^2$ estimation error $\|\widehat{r} - r^\circ\|_{L^2(\Omega)}$.

**Theorem 30.** *Let $\epsilon > 0$, $f \in U(B_{p,q}^s(\Omega)) \cap L^\infty(\Omega)$, $\widehat{f} \in L^\infty(\Omega)$, and, let $\Omega_X^{[\epsilon]} := [\epsilon, 1-\epsilon]^{d_X}$ and $\Omega_Y^{[\epsilon']} := [\epsilon', 1-\epsilon']^{d_Y}$. Moreover, we define $\overline{f} : \mathbb{R}^d \to \mathbb{R}$ as*

$$\overline{f}(x) = \begin{cases} \widehat{f}(x) & (x \in \Omega) \\ 0 & (otherwise). \end{cases}$$

*Additionally, for each $x \in \Omega_X$, let $\rho_x$ be a probability density on $\Omega_Y$, and let $\rho(x,y) := \rho_x(y)$. Suppose that $\underline{\rho} \le \rho_x(y) \le \overline{\rho}$ for all $x \in \Omega_X, y \in \Omega_Y$ for some $\underline{\rho}, \overline{\rho} > 0$. Then, it holds*

$$\sup_{x \in \Omega_X^{[\epsilon]}} \left\|(K * \overline{f})(x,y) - f(x,y)\right\|_{L^2(\rho_x | \Omega_Y^{[\epsilon']})}$$

$$\lesssim \underline{\rho}^{-1/2}\left(\gamma^{-d_X/2} \cdot |f - \widehat{f}\|_{L^2(\rho)} + \overline{\rho}^{1/2}\gamma^{s'}\right) + \exp\left(-\frac{2\epsilon^2}{r^2\gamma^2}\right),$$

*where $s' := s\min\{1, p/2\}$. In particular, by setting $\epsilon \gg \gamma = \overline{\rho}^{-\frac{1}{2s'+d}} \|\widehat{f} - f\|_{L^2(\Omega)}^{\frac{2}{2s'+d}}$, we have*

$$\sup_{x \in \Omega_X^{[\epsilon]}} \left\|(K * \overline{f})(x,y) - f(x,y)\right\|_{L^2(\rho_x | \Omega_Y^{[\epsilon']})} \lesssim \underline{\rho}^{-1/2}\overline{\rho}^{\frac{d/2}{2s'+d}} \|\widehat{f} - f\|_{L^2(\Omega)}^{\frac{2s'}{2s'+d}}.$$

**A result without prompt-truncation.** We first introduce a linear operator $\mathfrak{E}$ that maps $f : \Omega \to \mathbb{R}$ to $\mathfrak{E}f : \mathbb{R}^d \to \mathbb{R}$ satisfying the following three properties:

- $(\mathfrak{E}f)_{|\Omega} = f$, that is, $\mathfrak{E}$ is an extension operator.

- For any $p \in [1, \infty]$, there exists a constant $a_p \ge 0$ satisfying $\|\mathfrak{E}f\|_{L^p(\mathbb{R}^d)} \le a_p\|f\|_{L^p(\Omega)}$.

- For any $s > 0, p \in [1, \infty], q \in (0, \infty]$, there exists a constant $b_p \ge 0$ satisfying $\|\mathfrak{E}f\|_{B_{p,q}^s(\mathbb{R}^d)} \le b_{p,q,s}\|f\|_{B_{p,q}^s(\Omega)}$.

For the existence of such an operator, see Stein (1970), Adams & Fournier (2003) and Triebel (2006). Then, we prove the following theorem, where we removed the truncation of the space of the prompt in Theorem 30 while replacing $\overline{f}$ by $\mathfrak{E}\widehat{f}$.

**Theorem 31.** *Let $f \in U(B_{p,q}^s(\Omega)) \cap L^\infty(\Omega)$ and $\widehat{f} \in L^\infty(\Omega)$. Moreover, for each $x \in \Omega_X$, let $\rho_x$ be a probability density on $\Omega_Y$. Suppose that $\underline{\rho} \le \rho_x(y) \le \overline{\rho}$ for all $x \in \Omega_X, y \in \Omega_Y$ for some $\underline{\rho}, \overline{\rho} > 0$. Then, it holds*

$$\sup_{x \in \Omega_X} \left\|(K * \mathfrak{E}\widehat{f})(x,y) - f(x,y)\right\|_{L^2(\rho_x | \Omega_Y^{[\epsilon]})} \lesssim \underline{\rho}^{-1/2}\left(\gamma^{-d/2}|f - \widehat{f}\|_{L^2(\rho)} + \overline{\rho}^{1/2}\gamma^{s\min\{1,p/2\}}\right),$$

To prove the theorem, we first show the following lemma, which upper-bounds the penalty of smoothing by $K$.

**Lemma 32.** *Let $f \in L^\infty(\mathbb{R}^d)$ and $r \in \mathbb{Z}_{>0}$. Moreover, for each $x \in \Omega_X$, let $\rho_x$ be a probability density on $\Omega_Y$. Suppose that $\rho_x(y) \le \overline{\rho}$ for all $x \in \Omega_X, y \in \Omega_Y$ for some $\overline{\rho} > 0$. Then, there exists a constant $C_r > 0$ that depends only on $r$ and $\|f\|_\infty$ such that*

$$\sup_{x \in \Omega_X} \|(K * f)(x, \cdot) - f(x, \cdot)\|_{L^2(\rho_x)} \le C_r \overline{\rho}^{1/2} \cdot w_{r,\infty}(f, \gamma/2)^{\min\{1,p/2\}}.$$

*Proof.* We adapt the proof of Theorem 2.2 in Eberts & Steinwart (2013). Similar to the first part of their proof, we have

$$(K * f)(x, y) = \int_{\mathbb{R}^d} \left(\frac{2}{\gamma^2 \pi}\right)^{d/2} K_{\frac{\gamma}{\sqrt{2}}}(h) \left(\sum_{j=1}^{r} \binom{r}{j} (-1)^{1-j} f(x + jh_X, y + jh_Y)\right) dh, \quad (19)$$

where $h_X := [h_1, \ldots, h_{d_X}]^\top \in \mathbb{R}^{d_X}$, $h_Y := [h_{d_X+1}, \ldots, h_d]^\top \in \mathbb{R}^{d_Y}$. With this and

$$\int_{\mathbb{R}^d} \left(\frac{2}{\gamma^2 \pi}\right)^{d/2} K_{\frac{\gamma}{\sqrt{2}}}(h) \, dh = 1,$$

we have

$$(K * f)(x, y) - f(x, y)$$

$$= \int_{\mathbb{R}^d} \left(\frac{2}{\gamma^2 \pi}\right)^{d/2} K_{\frac{\gamma}{\sqrt{2}}}(h) \left(\sum_{j=1}^{r} \binom{r}{j} (-1)^{1-j} f(x + jh_X, y + jh_Y)\right) dh - f(x)$$

$$= \int_{\mathbb{R}^d} \left(\frac{2}{\gamma^2 \pi}\right)^{d/2} K_{\frac{\gamma}{\sqrt{2}}}(h) \left(\sum_{j=1}^{r} \binom{r}{j} (-1)^{2r+1-j} f(x + jh_X, y + jh_Y) - f(x, y)\right) dh$$

$$= \int_{\mathbb{R}^d} \left(\frac{2}{\gamma^2 \pi}\right)^{d/2} K_{\frac{\gamma}{\sqrt{2}}}(h) \left(\sum_{j=0}^{r} \binom{r}{j} (-1)^{2r+1-j} f(x + jh_X, y + jh_Y)\right) dh$$

$$= \int_{\mathbb{R}^d} (-1)^{r+1} \left(\frac{2}{\gamma^2 \pi}\right)^{d/2} K_{\frac{\gamma}{\sqrt{2}}}(h) \left(\sum_{j=0}^{r} \binom{r}{j} (-1)^{r-j} f(x + jh_X, y + jh_Y)\right) dh.$$

Then, recalling the definition of $\Delta_h^r$, we have

$$(K * f)(x, y) - f(x, y) = \int_{\mathbb{R}^d} (-1)^{r+1} \left(\frac{2}{\gamma^2 \pi}\right)^{d/2} K_{\frac{\gamma}{\sqrt{2}}}(h) \Delta_h^r(f)(x, y) \, dh,$$

which yields

$$\|(K * f)(x, \cdot) - f(x, \cdot)\|_{L^2(\rho_x)} = \left\|\int_{\mathbb{R}^d} \left(\frac{2}{\gamma^2 \pi}\right)^{d/2} K_{\frac{\gamma}{\sqrt{2}}}(h) |\Delta_h^r(f)(x, \cdot)| \, dh\right\|_{L^2(\rho_x)}$$

$$\leq \sqrt{\overline{\rho}} \int_{\mathbb{R}^d} \left(\frac{2}{\gamma^2 \pi}\right)^{d/2} K_{\frac{\gamma}{\sqrt{2}}}(h) \|\Delta_h^r(f)(x, \cdot)\|_{L^2(\Omega_Y)} \, dh.$$

For $p \geq 2$, we have

$$\|\Delta_h^r(f)(x, \cdot)\|_{L^2(\Omega_Y)} \leq \|\Delta_h^r(f)(x, \cdot)\|_{L^p(\Omega_Y)}.$$

Additionally, for $p \in [1, 2)$, we have

$$\|\Delta_h^r(f)(x, \cdot)\|_{L^2(\Omega_Y)}^2 = \int_{\Omega_Y} (\Delta_h^r(f)(x, y))^2 \, dy$$

$$= \int_{\Omega_Y} (\Delta_h^r(f)(x, y))^p (\Delta_h^r(f)(x, y))^{2-p} \, dy$$

$$\leq \|\Delta_h^r(f)(x, \cdot)\|_{L^p(\Omega_Y)}^p \|\Delta_h^r(f)(x, \cdot)\|_{L^\infty(\Omega_Y)}^{2-p}.$$

Since it holds that

$$\|\Delta_h^r(f)(x, \cdot)\|_{L^\infty(\Omega_Y)} \leq \|f\|_\infty \sum_{j=0}^{r} \binom{r}{j} \leq \|f\|_\infty \cdot 2^r =: C_r^1,$$

we have

$$\|\Delta_h^r(f)(x,\cdot)\|_{L^2(\Omega_Y)} \leq C_r^1 \|\Delta_h^r(f)(x,\cdot)\|_{L^p(\Omega_Y)}^{p/2}.$$

Combining above, we have

$$\|\Delta_h^r(f)(x,\cdot)\|_{L^2(\Omega_Y)} \leq \max\{C_r^1, 1\}\|\Delta_h^r(f)(x,\cdot)\|_{L^p(\Omega_Y)}^{p'},$$

where $p' := \min\{1, p/2\}$. Therefore, we have

$$\sup_{x\in\Omega_X} \|(K*f)(x,\cdot) - f(x,\cdot)\|_{L^2(\Omega_Y)}$$

$$\leq \sqrt{\rho} \sup_{x\in\Omega_X} \int_{\mathbb{R}^d} \left(\frac{2}{\gamma^2\pi}\right)^{d/2} K_{\frac{\gamma}{\sqrt{2}}}(h)\|\Delta_h^r(f)(x,\cdot)\|_{L^2(\Omega_Y)}\,\mathrm{d}h$$

$$\leq \sqrt{\rho} \int_{\mathbb{R}^d} \left(\frac{2}{\gamma^2\pi}\right)^{d/2} K_{\frac{\gamma}{\sqrt{2}}}(h)\left(\sup_{x\in\Omega_X} \|\Delta_h^r(f)(x,\cdot)\|_{L^2(\Omega_Y)}\right)\mathrm{d}h$$

$$\leq \max\{C_r^1, 1\} \cdot \sqrt{\rho} \int_{\mathbb{R}^d} \left(\frac{2}{\gamma^2\pi}\right)^{d/2} K_{\frac{\gamma}{\sqrt{2}}}(h)\left(\sup_{x\in\Omega_X} \|\Delta_h^r(f)(x,\cdot)\|_{L^p(\Omega_Y)}\right)^{p'}\mathrm{d}h.$$

By the definition of $w_{r,\infty}$, we have

$$\sup_{x\in\Omega_X} \|\Delta_h^r(f)(x,\cdot)\|_{L^p(\Omega_Y)} \leq \left\|\sup_{x\in\Omega_X}|\Delta_h^r(f)(x,y)|\right\|_{L^p(\Omega_Y)}$$
$$\leq w_{r,p}(f, \|h\|_{L^2(\mathbb{R}^d)}) = w_{r,p}(f, \|h\|_{L^2(\mathbb{R}^d)}).$$

By the property of modulus of smoothness, for all $t > 0$, we have

$$w_{r,p}(f, \|h\|_{L^2(\mathbb{R}^d)}) \leq \left(1 + \frac{\|h\|_{L^2(\mathbb{R}^d)}}{t}\right)^r w_{r,p}(f, t).$$

Therefore, we have

$$\sup_{x\in\Omega_X} \|(K*f)(x,\cdot) - f(x,\cdot)\|_{L^2(\Omega_Y)}$$

$$\leq \max\{C_r^1, 1\} \cdot \sqrt{\rho} \int_{\mathbb{R}^d} \left(\frac{2}{\gamma^2\pi}\right)^{d/2} K_{\frac{\gamma}{\sqrt{2}}}(h)\left(1 + \frac{\|h\|_2}{\gamma/2}\right)^{rp'} w_{r,\infty}(f, \gamma/2)^{p'}\,\mathrm{d}h$$

$$= \max\{C_r^1, 1\} \cdot \sqrt{\rho} \cdot w_{r,\infty}(f, \gamma/2)^{p'} \cdot \int_{\mathbb{R}^d} \left(\frac{2}{\gamma^2\pi}\right)^{d/2} K_{\frac{\gamma}{\sqrt{2}}}(h)\left(1 + \frac{\|h\|_2}{\gamma/2}\right)^{rp'}\mathrm{d}h.$$

The proof of Theorem 2.2 in Eberts & Steinwart (2013) implies

$$\int_{\mathbb{R}^d} \left(\frac{2}{\gamma^2\pi}\right)^{d/2} K_{\frac{\gamma}{\sqrt{2}}}(h)\left(1 + \frac{\|h\|_2}{\gamma/2}\right)^{rp'}\mathrm{d}h \leq \int_{\mathbb{R}^d} \left(\frac{2}{\gamma^2\pi}\right)^{d/2} K_{\frac{\gamma}{\sqrt{2}}}(h)\left(1 + \frac{\|h\|_2}{\gamma/2}\right)^{r}\mathrm{d}h$$

$$\leq \sum_{i=0}^{\lceil r\rceil} \left(\binom{\lceil r\rceil}{i}(2d)^{\frac{i}{2}}\prod_{j=1}^{i}\left(j - \frac{1}{2}\right)^{\frac{1}{2}}\right) =: C_r^2,$$

where we used the fact that $p' \leq 1$ and $1 + \frac{\|h\|_2}{\gamma/2} > 1$ in the first inequality. Thus, we obtain

$$\sup_{x\in\Omega_X} \|(K*f)(x,\cdot) - f(x,\cdot)\|_{L^2(\Omega_Y)} \leq \max\{C_r^1, 1\}C_r^2 \cdot \sqrt{\rho} \cdot w_{r,\infty}(f, \gamma/2)^{\min\{1, p/2\}},$$

which completes the proof. $\qquad\square$

Using the lemma above, we prove Theorem 31.

*Proof of Theorem 31.* We define $\widetilde{f} := \mathfrak{E}f$ and $\check{f} := \mathfrak{E}\widehat{f}$. Then, the value to be bound can be decomposed as follows:

$$
\sup_{x \in \Omega_X} \|(f - \mathcal{K}\widehat{f})(x, \cdot)\|_{L^2(\rho_x | \Omega_Y^{[\varepsilon]})}
$$

$$
= \sup_{x \in \Omega_X} \|(\widetilde{f} - \mathcal{K}\check{f})(x, \cdot)\|_{L^2(\rho_x | \Omega_Y^{[\varepsilon]})}
$$

$$
\leq \sup_{x \in \Omega_X} \|(\widetilde{f} - K * \widetilde{f})(x, \cdot)\|_{L^2(\rho_x | \Omega_Y^{[\varepsilon]})} + \sup_{x \in \Omega_X} \|(K * (\widetilde{f} - \check{f}))(x, \cdot)\|_{L^2(\rho_x | \Omega_Y^{[\varepsilon]})}.
$$

We first upper-bound the second term. Let $\mathcal{H}_\gamma$ be the RKHS of the Gaussian RBF kernel $k_\gamma$ on $\mathbb{R}^d$. Then, using Theorem 2.3 in Eberts & Steinwart (2013), we have

$$
\|K * (\widetilde{f} - \check{f})\|_{\mathcal{H}_\gamma} \leq (\gamma\sqrt{\pi})^{-d/2}(2^r - 1)\|\widetilde{f} - \check{f}\|_{L^2(\mathbb{R}^d)}.
$$

Using the reproducing property of $\mathcal{H}_\gamma$, we have

$$
|(K * (\widetilde{f} - \check{f}))(x, y)| = \left\langle K * (\widetilde{f} - \check{f}), k_\gamma((x, y), \cdot) \right\rangle_{\mathcal{H}_\gamma} \leq \|K * (\widetilde{f} - \check{f})\|_{\mathcal{H}_\gamma} \|k_\gamma((x, y), \cdot)\|_{\mathcal{H}_\gamma}.
$$

Since it holds that

$$
\|k_\gamma(\cdot, (x, y))\|_{\mathcal{H}_\gamma} = \sqrt{\langle k_\gamma(\cdot, (x, y)), k_\gamma(\cdot, (x, y))\rangle_{\mathcal{H}_\gamma}} = \sqrt{k_\gamma((x, y), (x, y))} = 1,
$$

we have

$$
|(K * (\widetilde{f} - \check{f}))(x, y)| \leq \|K * (\widetilde{f} - \check{f})\|_{\mathcal{H}_\gamma} \leq (\gamma\sqrt{\pi})^{-d/2}(2^r - 1)\|\widetilde{f} - \check{f}\|_{L^2(\mathbb{R}^d)}.
$$

Using the property of the operator $\mathfrak{E}$, we have

$$
\|\widetilde{f} - \check{f}\|_{L^2(\mathbb{R}^d)} = \|\mathfrak{E}(f - \widehat{f})\|_{L^2(\mathbb{R}^d)} \lesssim \|f - \widehat{f}\|_{L^2(\Omega)},
$$

which yields

$$
\sup_{x \in \Omega_X} \|(K * (\widetilde{f} - \check{f}))(x, \cdot)\|_{L^2(\rho_x | \Omega_Y^{[\varepsilon]})} \leq \sup_{x \in \Omega_X, y \in \Omega_Y} |(K * (\widetilde{f} - \check{f}))(x, y)|
$$

$$
\lesssim \gamma^{-d/2}\|\widetilde{f} - \check{f}\|_{L^2(\mathbb{R}^d)}
$$

$$
\lesssim \gamma^{-d/2}\|f - \widehat{f}\|_{L^2(\Omega)}
$$

$$
\leq \underline{\rho}^{-1/2} \cdot \gamma^{-d/2}\|f - \widehat{f}\|_{L^2(\rho)}.
$$

Next, we bound the first term. Since $w_{r,p}(\widetilde{f}, t)$ is non-decreasing with respect to $t$, for any $t > 0$ and $u \in [t/2, t]$, it holds

$$
u^{-s}w_{r,p}(\widetilde{f}, t) \geq (t/2)^{-s}w_{r,p}(\widetilde{f}, t/2).
$$

Therefore, we have

$$
\int_{t/2}^{t} (u^{-s}w_{r,p}(\widetilde{f}, u))^q \frac{\mathrm{d}u}{u} \geq (t/2)^{-sq}w_{r,p}(\widetilde{f}, t/2)^q \int_{t/2}^{t} \frac{\mathrm{d}u}{u}
$$

$$
= (t/2)^{-sq}w_{r,p}(\widetilde{f}, t/2)^q[\log u]_{t/2}^t
$$

$$
= (\log 2) \cdot (t/2)^{-sq}w_{r,p}(\widetilde{f}, t/2)^q.
$$

The definition of semi-norm $|\cdot|_{B^s_{p,q}}$, we have

$$
\begin{aligned}
|\widetilde{f}|_{B^s_{p,q}} &= \left( \int_0^\infty (u^{-s} w_{r,p}(\widetilde{f}, u))^q \frac{\mathrm{d}u}{u} \right)^{1/q} \\
&\geq \left( \int_{t/2}^t (u^{-s} w_{r,p}(\widetilde{f}, u))^q \frac{\mathrm{d}u}{u} \right)^{1/q} \\
&\geq (\log 2)^{1/q} \cdot (t/2)^{-s} w_{r,p}(\widetilde{f}, t/2).
\end{aligned}
$$

Hence, it holds

$$
w_{r,p}(\widetilde{f}, t) \leq w_{r,p}(\widetilde{f}, t/2) \leq (\log 2)^{-1/q} |\widetilde{f}|^q_{B^s_{p,q}} \cdot (t/2)^s.
$$

Moreover, for any $x \in \Omega_X, y \in \Omega_Y$, we have

$$
(\rho_x \mid \Omega_Y^{[\varepsilon]})(y) \leq \frac{\rho_x(y)}{\mathbb{P}[y \in \Omega_Y^{[\varepsilon]}]} \leq \frac{\overline{\rho}}{(1 - 2\epsilon)^d \underline{\rho}} \lesssim \overline{\rho} \underline{\rho}^{-1}.
$$

Using Lemma 32, we have

$$
\sup_{x \in \Omega_X} \left\| (K * \widetilde{f})(x, \cdot) - \widetilde{f}(x, \cdot) \right\|_{L^2(\rho_x)} \lesssim \overline{\rho}^{1/2} \underline{\rho}^{-1/2} \gamma^{s \min\{1, p/2\}}.
$$

Combining above, we have

$$
\sup_{x \in \Omega_X} \| (f - K * \widehat{f})(x, \cdot) \|_{L^2(\rho_x | \Omega_Y^{[\varepsilon]})} \lesssim \underline{\rho}^{-1/2} \left( \gamma^{-d/2} \| f - \widehat{f} \|_{L^2(\mathbb{R}^d)} + \overline{\rho}^{1/2} \gamma^{s \min\{1, p/2\}} \right),
$$

which gives this assertion. $\qquad \square$

**Proof of Theorem 30.**  Finally, we prove Theorem 30.

*Proof.*  Using (19), we have

$$
\begin{aligned}
&\left| (K * \mathfrak{E}\widehat{f})(x, y) - (K * \overline{f})(x, y) \right| \\
&= \left| \int_{\mathbb{R}^{d_X}} \left( \frac{2}{\gamma^2 \pi} \right)^{d_X/2} K_{\frac{\gamma}{\sqrt{2}}}(h) \left( \sum_{j=1}^r \binom{r}{j} (-1)^{1-j} \left( \mathfrak{E}\widehat{f}(x + jh, y) - \overline{f}(x + jh, y) \right) \right) \mathrm{d}h \right| \\
&\leq \int_{\mathbb{R}^{d_X}} \left( \frac{2}{\gamma^2 \pi} \right)^{d_X/2} K_{\frac{\gamma}{\sqrt{2}}}(h) \left( \sum_{j=1}^r \binom{r}{j} \left| \mathfrak{E}\widehat{f}(x + jh, y) - \overline{f}(x + jh, y) \right| \right) \mathrm{d}h \\
&\leq \int_{\mathbb{R}^{d_X}} \left( \frac{2}{\gamma^2 \pi} \right)^{d_X/2} K_{\frac{\gamma}{\sqrt{2}}}(h) \left( \sum_{j=1}^r \binom{r}{j} \| \widehat{f} \|_\infty \mathbb{1}_{\mathbb{R}^{d_X} \setminus \Omega_X}(x + jh) \right) \mathrm{d}h.
\end{aligned}
$$

Since it holds $x \in [\epsilon, 1 - \epsilon]^{d_X}$, $x + jh \notin \Omega_X$ when $r\|h\|_\infty \geq \|jh\|_\infty \geq \epsilon$.

$$
\begin{aligned}
\left| (K * \mathfrak{E}\widehat{f})(x, y) - (K * \overline{f})(x, y) \right| &\leq \| \widehat{f} \|_\infty \int_{\|h\|_\infty \geq \epsilon/r} \left( \frac{2}{\gamma^2 \pi} \right)^{d_X/2} K_{\frac{\gamma}{\sqrt{2}}}(h) \left( \sum_{j=1}^r \binom{r}{j} \right) \mathrm{d}h \\
&\lesssim \int_{\|h\|_\infty \geq \epsilon/r} \left( \frac{2}{\gamma^2 \pi} \right)^{d_X/2} \exp\left( -\frac{2\|h\|^2}{\gamma^2} \right) \mathrm{d}h
\end{aligned}
$$

The component in the integral is the density of $\mathcal{N}\left( 0, \frac{\gamma^2}{4} I_{d_X} \right)$. Therefore, we have

$$
\left| (K * \mathfrak{E}\widehat{f})(x, y) - (K * \overline{f})(x, y) \right| \lesssim 2 d_X \exp\left( -\frac{2\epsilon^2}{r^2 \gamma^2} \right),
$$

which completes the proof. $\qquad \square$

# E. Proofs for Section 5

In this section, we provide the proof of Theorem 10.

Before proving the theorem, we clarify the configuration of the hyper-parameters used in Algorithm 1. Specifically, the parameter $\mu^{(\tau)}$ of Inference-time Pessimism, $\sigma^{(\tau)}$ of Gaussian mollification, $\kappa^{(\tau)}$ of the scale of mixture with uniform distribution have to be set for each step. They are updated as follows:

$$\mu^{(\tau)} \simeq \left(\mu^{(\tau-1)}\right)^u (\mathcal{E}_\star)^{\frac{2(1+\gamma)}{2+\gamma} - \beta \frac{2s'}{2s'+d} \frac{2}{2+\gamma}}, \quad \sigma^{(\tau)} \simeq \left(\sigma^{(\tau-1)}\right)^u (\mathcal{E}_\star)^{\frac{2s'}{2s'+d} \frac{2}{2+\gamma} \frac{\beta}{d}}, \quad \kappa^{(\tau)} \simeq \left(\kappa^{(\tau-1)}\right)^u (\mathcal{E}_\star)^{\frac{2s'}{2s'+d} \frac{2}{2+\gamma}},$$

and they are initialized as

$$\mu^{(1)} = (\mathcal{E}_\star)^{\frac{2(1+\gamma)}{2+\gamma}}, \quad \sigma^{(1)} = (\mathcal{E}_\star)^{\frac{2s'}{2s'+d} \frac{2}{2+\gamma} \frac{\beta}{d}}, \quad \kappa^{(1)} = (\mathcal{E}_\star)^{\frac{2s'}{2s'+d} \frac{2}{2+\gamma}},$$

where $\mathcal{E}_\star$ is a typical estimation error of nonparametric regression for Besov spaces defined as

$$\mathcal{E}_\star := n_0^{\frac{2s}{2s+d}} \log^2(n_0).$$

We first prove the following lemma, which constructs a comparator policy that attains the $\epsilon$-optimal reward for $\epsilon > 0$ with bounded coverage from the policy that appears in the algorithm. This lemma corresponds to Lemma 12 in the main text.

**Lemma 33** (Existence of $\epsilon$-comparator policy). *Suppose that $r^\circ \in B_{p,q}^s(\Omega)$ satisfies Assumption 9. Moreover, let $\widehat{\pi}$ be a policy satisfying*

$$\sup_{x \in \Omega_X} \mathbb{E}_{y \sim \widehat{\pi}(\cdot|x)}[r^*(x) - r^\circ(x,y)] \leq \mathcal{R},$$

*for some $\mathcal{R} \in (0, 1/2)$. Additionally, we define the policy $\pi_G, \pi, \pi_{\mathrm{trnc}}$ as follows:*

$$\pi_G(\cdot \mid x) := \left(\pi(\cdot \mid x) * \mathcal{N}(0, \sigma^2 I_d)\right) \mid \Omega_Y,$$
$$\pi(\cdot \mid x) := \mathcal{R} \, \mathrm{Unif}(\Omega_Y) + (1 - \mathcal{R})\pi_G(\cdot \mid x),$$
$$\pi_{\mathrm{trnc}}(\cdot \mid x) := \pi(\cdot \mid x) \mid \Omega_Y^{[\eta/2]}.$$

*Then, for any $\epsilon \in (0, \mathcal{R})$, if $\sigma^2 \simeq \mathcal{R}^{2\beta/d}$, there exists a comparator policy $\pi_\epsilon^*$ satisfying the following two conditions for all $x \in \Omega_X$:*

*(i) $r^*(x) - \mathbb{E}_{y \sim \pi_\epsilon^*(\cdot|x)}[r^\circ(x,y)] \leq \epsilon$, \qquad (ii) $\mathcal{C}(x; \pi_\epsilon^*, \pi_{\mathrm{trnc}}) \leq \epsilon^{-\gamma} \mathcal{R}^\beta$.*

*Proof.* As same as Lemma 8, we set the policy $\pi_\epsilon^*$ as

$$\pi_\epsilon^*(y \mid x) := \frac{\mathbb{1}_{S_\epsilon(x)}(y)}{\lambda(S_\epsilon(x))}.$$

The condition (i) can be confirmed by the totally same calculation as Lemma 8. To discuss the condition (ii), we first lower bound the density of $\pi_G$. For $y \in S_\epsilon(x)$, we have

$$\pi_G(y \mid x) \geq \frac{1}{(2\pi\sigma^2)^{d/2}} \int \widehat{\pi}(z \mid x) \exp\left(-\frac{\|y - z\|^2}{2\sigma^2}\right) \mathrm{d}z$$

$$\geq \frac{1}{(2\pi\sigma^2)^{d/2}} \int_{S_{2\mathcal{R}}(x)} \widehat{\pi}(z \mid x) \exp\left(-\frac{\|y - z\|^2}{2\sigma^2}\right) \mathrm{d}z.$$

For $y \in S_\epsilon(x), z \in S_{2\mathcal{R}}(x) \subset \Omega_Y$, it holds that

$$\|y - z\| = \|(y - y^*(x)) + (y^*(x) - z)\|$$
$$\leq \|y\| + \|z\|$$
$$\leq \epsilon^{\beta/d} + (2\mathcal{R})^{\beta/d},$$

by **(A1)**. Moreover, it holds

$$\mathbb{P}_{y\sim\widehat{\pi}(\cdot|x)}[r^*(x) - r^\circ(x,y) \le 2\mathcal{R}] = 1 - \mathbb{P}_{y\sim\widehat{\pi}(\cdot|x)}[r^*(x) - r^\circ(x,y) > 2\mathcal{R}]$$
$$\ge 1 - \frac{\mathbb{E}_{y\sim\widehat{\pi}(\cdot|x)}[r^*(x) - r^\circ(x,y)]}{2\mathcal{R}}$$
$$\ge 1 - \frac{\mathcal{R}}{2\mathcal{R}} = \frac{1}{2}.$$

Therefore, we have

$$\pi_G(y \mid x) \ge \frac{1}{(2\pi\sigma^2)^{d/2}} \exp\left(-\frac{(\epsilon^{\beta/d} + (2\mathcal{R})^{\beta/d})^2}{2\sigma^2}\right) \int_{S_{2\mathcal{R}}(x)} \widehat{\pi}(z \mid x) \, \mathrm{d}z$$
$$\ge \frac{1}{(2\pi\sigma^2)^{d/2}} \exp\left(-\frac{(\epsilon^{\beta/d} + (2\mathcal{R})^{\beta/d})^2}{2\sigma^2}\right) \cdot \mathbb{P}_{z\sim\widehat{\pi}(\cdot|x)}[r^*(x) - r^\circ(x,z) \le 2\mathcal{R}]$$
$$\ge \frac{1/2}{(2\pi\sigma^2)^{d/2}} \exp\left(-\frac{9\mathcal{R}^{2\beta/d}}{2\sigma^2}\right).$$

By setting $\sigma^2 = 9\mathcal{R}^{2\beta/d}/2$, for any $y \in S_{2\mathcal{R}}(x) \subset \Omega_Y^{[\eta/2]}$, we have

$$\pi_G(y \mid x) \gtrsim \frac{1}{(2\pi\sigma^2)^{d/2}} \exp(-1) \gtrsim \mathcal{R}^{-\beta}.$$

This implies that

$$\pi_{\mathrm{trnc}}(y \mid x) \gtrsim \pi(y \mid x) \gtrsim \pi_G(y \mid x) \gtrsim \mathcal{R}^{-\beta}.$$

Therefore, we have

$$\mathcal{C}(x; \pi_\epsilon^*, \pi_{\mathrm{trnc}}) = \mathbb{E}_{y\sim\pi_\epsilon^*(\cdot|x)}\left[\frac{\pi_\epsilon^*(y \mid x)}{\pi_{\mathrm{trnc}}(y \mid x)}\right]$$
$$= \int \frac{\pi_\epsilon^*(y \mid x)^2}{\pi_{\mathrm{trnc}}(y \mid x)} \, \mathrm{d}y$$
$$\lesssim \int \frac{\mathbb{1}_{S_\epsilon(x)}(y)/\lambda(S_\epsilon(x))^2}{\mathcal{R}^{-\beta}} \, \mathrm{d}y$$
$$\lesssim \frac{\mathcal{R}^\beta}{\lambda(S_\epsilon(x))} \lesssim \mathcal{R}^\beta \epsilon^{-\gamma},$$

which completes the proof. $\qquad\square$

*Proof of Theorem 10.* We first provide some definitions of the symbols. Let $\overline{\mathcal{E}}^{(\tau)}, \overline{\mathcal{E}}_\infty^{(\tau)}$ be the error of the reward estimators $\widehat{r}^{(\tau)}, \widetilde{r}^{(\tau)}$, respectively, defined as

$$\overline{\mathcal{E}}^{(\tau)} := \left(\mathbb{E}_{x\sim P_X}\mathbb{E}_{y\sim\pi^{(\tau-1)}(\cdot|x)}\left[(\widehat{r}^{(\tau-1)}(x,y) - r^\circ(x,y))^2\right]\right)^{1/2},$$
$$\overline{\mathcal{E}}_\infty^{(\tau)} := \sup_{x\in\Omega_X^{(\tau)}} \left(\mathbb{E}_{y\sim\pi_{\mathrm{trnc}}^{(\tau-1)}(\cdot|x)}\left[(\widetilde{r}^{(\tau)}(x,y) - r^\circ(x,y))^2\right]\right)^{1/2}.$$

Additionally, let $\pi_{\mathrm{Pes}}^{(\tau)}$ be the policy which is the pure output of Inference-Time Pessimism with reference policy $\pi_{\mathrm{trnc}}^{(\tau-1)}$ before adding Gaussian noises:

$$\pi_G^{(\tau)}(\cdot \mid x) = \left(\pi_{\mathrm{Pes}}^{(\tau)}(\cdot \mid x) * \mathcal{N}(0, I_{d_Y})\right) \mid \Omega_Y,$$
$$\pi^{(\tau)}(\cdot \mid x) = \kappa^{(\tau)} \mathrm{Unif}(\Omega_Y) + (1 - \kappa^{(\tau)})\pi_G^{(\tau)}(\cdot \mid x),$$
$$\pi_{\mathrm{trnc}}^{(\tau)}(\cdot \mid x) = \pi^{(\tau)}(\cdot \mid x) \mid \Omega_Y^{[\eta/2]}.$$

Moreover, let $\mathcal{R}_{\text{Pes}}^{(\tau)}(x), \overline{\mathcal{R}}_{\text{Pes}}^{(\tau)}, \overline{\mathcal{R}}_{\text{Pes},\infty}^{(\tau)}$ be the regret of $\pi_{\text{Pes}}^{(\tau)}$ defined as

$$\mathcal{R}_{\text{Pes}}^{(\tau)}(x) := \mathbb{E}_{y \sim \pi_{\text{Pes}}^{(\tau)}}[r^*(x) - r^\circ(x, y)],$$

$$\overline{\mathcal{R}}_{\text{Pes}}^{(\tau)} := \mathbb{E}_{x \sim P_X}\mathbb{E}_{y \sim \pi_{\text{Pes}}^{(\tau)}}[r^*(x) - r^\circ(x, y)],$$

$$\overline{\mathcal{R}}_{\text{Pes},\infty}^{(\tau)} := \sup_{x \in \Omega_X} \mathbb{E}_{y \sim \pi_{\text{Pes}}^{(\tau)}}[r^*(x) - r^\circ(x, y)],$$

and $\mathcal{R}^{(\tau)}(x), \overline{\mathcal{R}}_\infty^{(\tau)}$ be the regret of $\pi^{(\tau)}$ defined as

$$\mathcal{R}^{(\tau)}(x) := \mathbb{E}_{y \sim \pi^{(\tau)}}[r^*(x) - r^\circ(x, y)],$$

$$\overline{\mathcal{R}}_\infty^{(\tau)} := \sup_{x \in \Omega_X} \mathbb{E}_{y \sim \pi^{(\tau)}}[r^*(x) - r^\circ(x, y)].$$

We also define $\mathcal{R}_G^{(\tau)}(x)$ and $\overline{\mathcal{R}}_{G,\infty}^{(\tau)}$ for $\pi_G^{(\tau)}$:

$$\mathcal{R}_G^{(\tau)}(x) := \mathbb{E}_{y \sim \pi_G^{(\tau)}}[r^*(x) - r^\circ(x, y)],$$

$$\overline{\mathcal{R}}_{G,\infty}^{(\tau)} := \sup_{x \in \Omega_X} \mathbb{E}_{y \sim \pi_G^{(\tau)}}[r^*(x) - r^\circ(x, y)].$$

**Bounding $\overline{\mathcal{E}}_\infty^{(\tau)}$** We first obtain the prompt-uniform error bound for the reward estimator of $\tau$-th step using Theorem 30. Since it holds that

$$\|\pi_G^{(\tau-1)}(\cdot \mid x)\|_{L^\infty(\Omega_Y)} \lesssim \sup_{y \in \Omega_Y} \int \frac{2}{(2\pi(\sigma^{(\tau)})^2)^{d/2}} \exp\left(-\frac{\|x - z\|^2}{2(\sigma^{(\tau)})^2}\right) \pi_{\text{Pes}}^{(\tau-1)}(\cdot \mid z)\,\mathrm{d}z$$

$$\leq \frac{2}{(2\pi(\sigma^{(\tau)})^2)^{d/2}} \int \pi_{\text{Pes}}^{(\tau-1)}(\cdot \mid z)\,\mathrm{d}z$$

$$\lesssim (\sigma^{(\tau)})^{-d} \lesssim \left(\overline{\mathcal{R}}_{\text{Pes},\infty}^{(\tau-1)}\right)^{-\beta},$$

$$\|\pi^{(\tau-1)}(\cdot \mid x)\|_{L^\infty(\Omega_Y)} \lesssim \kappa^{(\tau-1)} + \|\pi_G^{(\tau-1)}(\cdot \mid x)\|_{L^\infty(\Omega_Y)} \lesssim \left(\overline{\mathcal{R}}_{\text{Pes},\infty}^{(\tau-1)}\right)^{-\beta},$$

and

$$\pi^{(\tau-1)}(y \mid x) \geq \kappa^{(\tau-1)},$$

for all $x \in \Omega_X, y \in \Omega_Y$, we have

$$\overline{\mathcal{E}}_\infty^{(\tau)} \lesssim (\kappa^{(\tau)})^{-1/2}\left\{\left(\overline{\mathcal{R}}_{\text{Pes},\infty}^{(\tau-1)}\right)^{-\beta}\right\}^{\frac{d/2}{2s'+d}}\left(\overline{\mathcal{E}}^{(\tau)}\right)^{\frac{2s'}{2s'+d}}.$$

By setting $\kappa^{(\tau-1)} \simeq \overline{\mathcal{R}}_{\text{Pes},\infty}^{(\tau-1)}$, we have

$$\overline{\mathcal{E}}_\infty^{(\tau)} \lesssim \left(\overline{\mathcal{R}}_{\text{Pes},\infty}^{(\tau-1)}\right)^{-\frac{1}{2}\left(1+\beta\cdot\frac{d}{2s+d}\right)}\left(\overline{\mathcal{E}}^{(\tau)}\right)^{\frac{2s'}{2s'+d}}.$$

Since $\beta \geq 1 + \frac{d}{2s'}$, we have $1 + \beta \cdot \frac{d}{2s'+d} \leq \beta$, which yields

$$\overline{\mathcal{E}}_\infty^{(\tau)} \lesssim \left(\overline{\mathcal{R}}_{\text{Pes},\infty}^{(\tau-1)}\right)^{-\beta/2}\left(\overline{\mathcal{E}}^{(\tau)}\right)^{\frac{2s'}{2s'+d}}.$$

**Bounding $\overline{\mathcal{R}}_{\text{Pes},\infty}^{(\tau)}$**  Applying Lemma 33 with $\widehat{\pi} \leftarrow \pi_{\text{Pes}}^{(\tau-1)}$, if we set $\sigma^{(\tau-1)} \simeq \left(\overline{\mathcal{R}}_{\text{Pes},\infty}^{(\tau-1)}\right)^{\beta/d}$, there exists a comparator policy $\pi_{\epsilon,\tau}^*$ satisfying

$$r^*(x) - \mathbb{E}_{y \sim \pi_{\epsilon,\tau}^*(\cdot|x)}[r^\circ(x,y)] \le \epsilon, \qquad \mathcal{C}(x; \pi_{\epsilon,\tau}^*, \pi_{\text{Pes}}^{(\tau)}) \le \epsilon^{-\gamma} \left(\overline{\mathcal{R}}_{\text{Pes},\infty}^{(\tau-1)}\right)^{\beta}.$$

Therefore, using Proposition 7, we have

$$
\begin{aligned}
&\mathcal{R}_{\text{Pes}}^{(\tau)}(x) \\
&= r^*(x) - \mathbb{E}_{y \sim \pi_{\text{Pes}}^{(\tau)}(\cdot|x)}[r^\circ(x,y)] \\
&\le \left(r^*(x) - \mathbb{E}_{y \sim \pi_{\epsilon,\tau}^*}[r^\circ(x,y)]\right) + \left(\mathbb{E}_{y \sim \pi_{\epsilon,\tau}^*}[r^\circ(x,y)] - \mathbb{E}_{y \sim \pi_{\text{Pes}}^{(\tau)}}[r^\circ(x,y)]\right) \\
&\lesssim \epsilon + \mu^{(\tau)} \epsilon^{-\gamma} \left(\overline{\mathcal{R}}_{\text{Pes},\infty}^{(\tau-1)}\right)^{\beta} + \left(\mu^{(\tau)}\right)^{-1} \left(\overline{\mathcal{E}}_\infty^{(\tau)}\right)^2 + \left(\mu^{(\tau)}\right)^{-1} \left(\overline{\mathcal{E}}_\infty^{(\tau)}\right) \exp\left(-\frac{\mu^{(\tau)} N}{C_1(R + \mu^{(\tau)})}\right) \\
&\lesssim \epsilon + \mu^{(\tau)} \epsilon^{-\gamma} \left(\overline{\mathcal{R}}_{\text{Pes},\infty}^{(\tau-1)}\right)^{\beta} + \left(\mu^{(\tau)}\right)^{-1} \left(\overline{\mathcal{R}}_{\text{Pes},\infty}^{(\tau-1)}\right)^{-\beta} \left(\overline{\mathcal{E}}^{(\tau)}\right)^{\frac{2s'}{2s'+d}\cdot 2} \\
&\qquad\qquad\qquad\qquad + \left(\mu^{(\tau)}\right)^{-1} \left(\overline{\mathcal{E}}_\infty^{(\tau)}\right) \exp\left(-\frac{\mu^{(\tau)} N}{C_1(R + \mu^{(\tau)})}\right).
\end{aligned}
\tag{20}
$$

The right-hand side is minimized when

$$\epsilon \simeq \left(\overline{\mathcal{E}}^{(\tau)}\right)^{\frac{2s'}{2s'+d}\frac{2}{2+\gamma}}, \quad \mu^{(\tau)} \simeq \frac{\left(\overline{\mathcal{E}}^{(\tau)}\right)^{\frac{2s'}{2s'+d}}}{\epsilon^{-\gamma/2}\left(\overline{\mathcal{R}}_{\text{Pes},\infty}^{(\tau-1)}\right)^{\beta}} \simeq \left(\overline{\mathcal{E}}^{(\tau)}\right)^{\frac{2(1+\gamma)}{2+\gamma}} \left(\overline{\mathcal{R}}_{\text{Pes},\infty}^{(\tau-1)}\right)^{-\beta}.$$

Thus, we have

$$\overline{\mathcal{R}}_{\text{Pes},\infty}^{(\tau)} \lesssim \left(\overline{\mathcal{E}}^{(\tau)}\right)^{\frac{2s'}{2s'+d}\frac{2}{2+\gamma}}. \tag{21}$$

**Bounding $\overline{\mathcal{R}}_\infty^{(\tau)}$**  We upper-bound $\mathcal{R}^{(\tau)}(x)$ by $\mathcal{R}_{\text{Pes}}^{(\tau)}(x)$; in other words, we evaluate how much the regret increases by mollification and mixture with a uniform distribution. To this end, we first upper bound $\mathcal{R}_G^{(\tau)}(x)$ by $\mathcal{R}_{\text{Pes}}^{(\tau)}(x)$. The assumption

$$B(y^*(x), C_2 \epsilon^{\gamma/d}) \subseteq S_\epsilon(x) \subseteq B(y^*(x), C_1 \epsilon^{\beta/d}),$$

with $S_\epsilon(x) := \{y \mid f^*(x) - f^\circ(x,y) \le \epsilon\}$ implies that

$$C_2 \cdot \epsilon^{\gamma/d} \le \|y - y^*(x)\| \le C_1 \cdot \epsilon^{\beta/d},$$

which yields

$$\frac{1}{C_1}\|y - y^*(x)\|^{d/\beta} \le f^*(x) - f^\circ(x,y) \le \frac{1}{C_2}\|y^*(x) - y\|^{d/\gamma}.$$

Taking expectation with respect to $y \sim \pi_{\text{Pes}}^{(\tau)}(\cdot \mid x)$, we have

$$\frac{1}{C_1}\mathbb{E}_{y \sim \pi_{\text{Pes}}^{(\tau)}(\cdot|x)}[\|y - y^*(x)\|^{d/\beta}] \le \mathcal{R}_{\text{Pes}}^{(\tau)}. \tag{22}$$

Moreover, the expectation with respect to $y \sim \pi^{(\tau)}(\cdot \mid x)$ implies

$$\mathcal{R}^{(\tau)} \le \frac{1}{C_2}\mathbb{E}_{y \sim \pi_G^{(\tau)}(\cdot|x)}[\|y - y^*(x)\|^{d/\gamma}]. \tag{23}$$

The right-hand side of (23) can be bounded as

$$
\begin{aligned}
\mathbb{E}_{y \sim \pi_G^{(\tau)}(\cdot|x)}[\|y - y^*(x)\|^{d/\gamma}] &= \mathbb{E}_{y \sim \pi_G^{(\tau)}(\cdot|x)}[(\|y - y^*(x)\|^{d/\beta})^{\beta/\gamma}] \\
&\le \left(\mathbb{E}_{y \sim \pi_G^{(\tau)}(\cdot|x)}[\|y - y^*(x)\|^{d/\beta}]\right)^{\beta/\gamma} \quad (\because \beta/\gamma \le 1).
\end{aligned}
$$

Moreover, we have

$$
\mathbb{E}_{y \sim \pi_G^{(\tau)}(\cdot \mid x)}[\|y - y^*(x)\|^{d/\beta}]
$$

$$
= \int_{y \in \Omega_Y} \|y - y^*(x)\|^{d/\beta} \pi_G^{(\tau)}(y \mid x) \, \mathrm{d}y
$$

$$
\lesssim \int_{y \in \Omega_Y} \int_{z \in \mathbb{R}^d} \|y - y^*(x)\|^{d/\beta} \cdot \frac{1}{(2\pi(\sigma^{(\tau)})^2)^{d/2}} \exp\left(-\frac{\|y - z\|^2}{2(\sigma^{(\tau)})^2}\right) \pi_{\mathsf{Pes}}^{(\tau)}(z \mid x) \, \mathrm{d}z \, \mathrm{d}y \,.
$$

Since it holds

$$
\|y - y^*(x)\|^{d/\beta} \le c_{d/\beta} \left(\|y - z\|^{d/\beta} + \|z - y^*(x)\|^{d/\beta}\right)
$$

with

$$
c_{d/\beta} = \begin{cases} 1 & (d/\beta \le 1), \\ 2^{d/\beta - 1} & (d/\beta > 1), \end{cases}
$$

we have

$$
\frac{1}{c_{d/\beta}} \mathbb{E}_{y \sim \pi_G^{(\tau)}(\cdot \mid x)}[\|y - y^*(x)\|^{d/\beta}]
$$

$$
\lesssim \underbrace{\int_{y \in \Omega_Y} \int_{z \in \mathbb{R}^d} \|y - z\|^{d/\beta} \cdot \frac{1}{(2\pi(\sigma^{(\tau)})^2)^{d/2}} \exp\left(-\frac{\|y - z\|^2}{2(\sigma^{(\tau)})^2}\right) \pi_{\mathsf{Pes}}^{(\tau)}(z \mid x) \, \mathrm{d}z \, \mathrm{d}y}_{(a)}
$$

$$
+ \underbrace{\int_{y \in \Omega_Y} \int_{z \in \mathbb{R}^d} \|z - y^*(x)\|^{d/\beta} \cdot \frac{1}{(2\pi(\sigma^{(\tau)})^2)^{d/2}} \exp\left(-\frac{\|y - z\|^2}{2(\sigma^{(\tau)})^2}\right) \pi_{\mathsf{Pes}}^{(\tau)}(z \mid x) \, \mathrm{d}z \, \mathrm{d}y}_{(b)} \,.
$$

As for the first term (a), by setting $w := y - z$, it can be evaluated as

$$
(a) = \int_{w \in \mathbb{R}^d} \int_{z \in \mathbb{R}^d} \|w\|^{d/\beta} \cdot \frac{1}{(2\pi(\sigma^{(\tau)})^2)^{d/2}} \exp\left(-\frac{\|w\|^2}{2(\sigma^{(\tau)})^2}\right) \pi_{\mathsf{Pes}}^{(\tau)}(z \mid x) \, \mathrm{d}z \, \mathrm{d}w
$$

$$
= \int_{z \in \mathbb{R}^d} \pi_{\mathsf{Pes}}^{(\tau)}(z \mid x) \, \mathrm{d}z \int_{w \in \mathbb{R}^d} \|w\|^{d/\beta} \cdot \frac{1}{(2\pi(\sigma^{(\tau)})^2)^{d/2}} \exp\left(-\frac{\|w\|^2}{2(\sigma^{(\tau)})^2}\right) \mathrm{d}w
$$

$$
\le C(\sigma^{(\tau)})^{d/\beta},
$$

for some constant $C > 0$. Additionally, the second term (b) can be evaluated as

$$
(b) = \int_{y \in \Omega_Y} \frac{1}{(2\pi(\sigma^{(\tau)})^2)^{d/2}} \exp\left(-\frac{\|y - z\|^2}{2(\sigma^{(\tau)})^2}\right) \mathrm{d}y \int_{z \in \mathbb{R}^d} \|z - y^*(x)\|^{d/\beta} \pi_{\mathsf{Pes}}^{(\tau)}(z \mid x) \, \mathrm{d}z
$$

$$
= \mathbb{E}_{y \sim \pi_{\mathsf{Pes}}^{(\tau)}(\cdot \mid x)}\left[\|y - y^*(x)\|^{d/\beta}\right].
$$

Hence, we have

$$
\mathbb{E}_{y \sim \pi_G^{(\tau)}(\cdot \mid x)}[\|y - y^*(x)\|^{d/\gamma}]
$$

$$
\le \left(c_{d/\beta} \cdot \left(C(\sigma^{(\tau)})^{d/\beta} + \mathbb{E}_{y \sim \pi_{\mathsf{Pes}}^{(\tau)}(\cdot \mid x)}\left[\|y - y^*(x)\|^{d/\beta}\right]\right)\right)^{\beta/\gamma}
$$

$$
\le c_{\beta/\gamma} c_{d/\beta}^{\beta/\gamma} \left(C(\sigma^{(\tau)})^{d/\gamma} + \mathbb{E}_{y \sim \pi_{\mathsf{Pes}}^{(\tau)}(\cdot \mid x)}\left[\|y - y^*(x)\|^{d/\beta}\right]^{\beta/\gamma}\right).
$$

Combining this and (22), (23), we have

$$
\begin{aligned}
\mathcal{R}_G^{(\tau)}(x) &\le \frac{1}{C_2}\mathbb{E}_{y\sim\pi_G^{(\tau)}(\cdot|x)}[\|y - y^*(x)\|^{d/\gamma}] \\
&\le \frac{c_{\beta/\gamma}c_{d/\beta}^{\beta/\gamma}}{C_2}\left(C(\sigma^{(\tau)})^{d/\gamma} + \mathbb{E}_{y\sim\pi_{\mathrm{Pes}}^{(\tau)}(\cdot|x)}\Big[\|y - y^*(x)\|^{d/\beta}\Big]^{\beta/\gamma}\right) \\
&\le \frac{c_{\beta/\gamma}c_{d/\beta}^{\beta/\gamma}}{C_2}\left(C(\sigma^{(\tau)})^{d/\gamma} + C_1^{\beta/\gamma}(\overline{\mathcal{R}}_{\mathrm{Pes}}^{(\tau)}(x))^{\beta/\gamma}\right) \\
&\lesssim (\sigma^{(\tau)})^{d/\gamma} + (\overline{\mathcal{R}}_{\mathrm{Pes}}^{(\tau)}(x))^{\beta/\gamma}.
\end{aligned}
$$

We can bound $\mathcal{R}^{(\tau)}(x)$ as follows:

$$
\mathcal{R}^{(\tau)}(x) \lesssim \kappa^{(\tau)}\cdot 1 + (1 - \kappa^{(\tau)})\cdot\mathcal{R}_G^{(\tau)}(x) \lesssim \kappa^{(\tau)} + (\sigma^{(\tau)})^{d/\gamma} + (\overline{\mathcal{R}}_{\mathrm{Pes}}^{(\tau)}(x))^{\beta/\gamma}.
$$

which implies

$$
\mathcal{R}_\infty^{(\tau)} \lesssim \kappa^{(\tau)} + (\sigma^{(\tau)})^{d/\gamma} + (\overline{\mathcal{R}}_{\mathrm{Pes},\infty}^{(\tau)})^{\beta/\gamma}.
$$

By setting $\kappa^{(\tau)} \simeq \overline{\mathcal{R}}_{\mathrm{Pes},\infty}^{(\tau)}$ and $\sigma^{(\tau)} \simeq (\overline{\mathcal{R}}_{\mathrm{Pes},\infty}^{(\tau)})^{\beta/d}$ and taking supremum over $x \in \Omega_X$, we have

$$
\overline{\mathcal{R}}_\infty^{(\tau)} \lesssim \left(\overline{\mathcal{R}}_{\mathrm{Pes},\infty}^{(\tau)}\right)^{\frac{\beta}{\gamma}}. \tag{24}
$$

**Bounding $\overline{\mathcal{E}}^{(\tau+1)}$.** Using Theorem 26 and (24), we have

$$
\begin{aligned}
\overline{\mathcal{E}}^{(\tau+1)} &= \mathbb{E}_{x\sim P_X}\mathbb{E}_{y\sim\pi^{(\tau+1)}(\cdot|x)}[(r^{(\tau+1)}(x,y) - r^\circ(x,y))^2] \\
&\lesssim \left(\overline{\mathcal{R}}_\infty^{(\tau)}\right)^{\frac{2\beta\varsigma}{2s+d}}\mathcal{E}_\star \lesssim \left(\overline{\mathcal{R}}_{\mathrm{Pes},\infty}^{(\tau)}\right)^{\frac{2\beta\varsigma}{2s+d}\frac{\beta}{\gamma}}\mathcal{E}_\star. \tag{25}
\end{aligned}
$$

Note that this is also satisfied for $\tau = 0$ by setting $\overline{\mathcal{R}}_{\mathrm{Pes},\infty}^{(0)} = R \simeq 1$. (Recall that the estimation error for the first step can be bounded by $\mathcal{E}_\star$ since its procedure is a standard least-square regression.)

**Obtaining the convergence rate.** Combining (21) and (25), we have

$$
\overline{\mathcal{E}}^{(\tau+1)} \lesssim \left(\overline{\mathcal{E}}^{(\tau)}\right)^u \mathcal{E}_\star,
$$

where $u := \frac{2\beta\varsigma}{2s+d}\frac{\beta}{\gamma}\frac{2s'}{2s'+d}\frac{2}{2+\gamma}$. Therefore, we have

$$
\begin{aligned}
\overline{\mathcal{E}}^{(T-1)} &\lesssim \left(\overline{\mathcal{E}}_{\mathrm{Pes},\infty}^{(T-2)}\right)^u\mathcal{E}_\star \lesssim \left(\overline{\mathcal{E}}_{\mathrm{Pes},\infty}^{(T-3)}\right)^{u^2}\mathcal{E}_\star^{1+u} \\
&\lesssim \left(\overline{\mathcal{E}}_{\mathrm{Pes},\infty}^{(T-4)}\right)^{u^3}\mathcal{E}_\star^{1+u+u^2} \lesssim \cdots \\
&\lesssim \left(\overline{\mathcal{E}}^{(1)}\right)^{u^{T-2}}\mathcal{E}_\star^{1+u+\cdots+u^{T-3}} \\
&\lesssim \mathcal{E}_\star^{1+u+\cdots+u^{T-3}+u^{T-2}} \\
&= (\mathcal{E}_\star)^{\frac{1-u^{T-1}}{1-u}} = (\mathcal{E}_\star)^{\frac{1}{1-u}}\left(\mathcal{E}_\star^{u^T}\right)^{-\frac{1/u}{1-u}}.
\end{aligned}
$$

The factor $\mathcal{E}_\star^{u^T}$ is bounded by a constant independent to $n$. Indeed, for any $a,b \in (0,1)$, $n^{a\cdot b^{\log n}} = e^{a\cdot n^{-\log b^{-1}}\cdot\log n}$ is convergent to 1 as $n \to \infty$, which implies that there exist some constant $c_1, c_2 > 0$ such that $c_1 < (\frac{1}{n})^{a\cdot b^{\log n}} < c_2$. Therefore, we have

$$
\overline{\mathcal{E}}^{(T-1)} \lesssim (\mathcal{E}_\star)^{\frac{1}{1-u}}.
$$

Using (21) and (25) again, we have

$$\overline{\mathcal{R}}_{\text{Pes},\infty}^{(T-1)} \lesssim (\mathcal{E}_\star)^{\frac{1}{1-u}\frac{2s'}{2s'+d}\frac{2}{2+\gamma}}, \quad \overline{\mathcal{E}}^{(T)} \lesssim (\mathcal{E}_\star)^{\frac{1}{1-u}}.$$

Similarly to (20), we have

$$\mathcal{R}_{\text{Pes}}^{(T)}(x) \lesssim \epsilon + \mu^{(T)}\epsilon^{-\gamma}\left(\overline{\mathcal{R}}_{\text{Pes},\infty}^{(T-1)}\right)^\beta + \left(\mu^{(T)}\right)^{-1}\left(\mathcal{E}^{(T)}(x)\right)^2.$$

Taking expectation over $x \sim P_X$ for the both side, we have

$$\overline{\mathcal{R}}_{\text{Pes}}^{(T)} \lesssim \epsilon + \mu^{(T)}\epsilon^{-\gamma}\left(\overline{\mathcal{R}}_{\text{Pes},\infty}^{(T-1)}\right)^\beta + \left(\mu^{(\tau)}\right)^{-1}\left(\overline{\mathcal{E}}^{(T)}\right)^2.$$

The right-hand side is minimized when

$$\epsilon \simeq \left((\overline{\mathcal{R}}_{\text{Pes},\infty}^{(T-1)})^{\frac{\beta}{2}}\cdot\overline{\mathcal{E}}^{(T)}\right)^{\frac{2}{2+\gamma}} = \left(\overline{\mathcal{R}}_{\text{Pes},\infty}^{(T-1)}\right)^{\frac{\beta}{2+\gamma}}\left(\overline{\mathcal{E}}^{(T)}\right)^{\frac{2}{2+\gamma}}$$

$$\mu^{(T)} \simeq \frac{\overline{\mathcal{E}}^{(T)}}{\epsilon^{-\gamma/2}\left(\overline{\mathcal{R}}_{\text{Pes},\infty}^{(T-1)}\right)^{\beta/2}} \simeq \left(\overline{\mathcal{E}}^{(T)}\right)^{\frac{2(1+\gamma)}{2+\gamma}}\left(\overline{\mathcal{R}}_{\text{Pes},\infty}^{(T-1)}\right)^{-\frac{\beta}{2+\gamma}}.$$

Then, we have

$$\overline{\mathcal{R}}_{\text{Pes}}^{(T)} \lesssim \left(\overline{\mathcal{R}}_{\text{Pes},\infty}^{(T-1)}\right)^{\frac{\beta}{2+\gamma}}\left(\overline{\mathcal{E}}^{(T)}\right)^{\frac{2}{2+\gamma}} \lesssim \left(\mathcal{E}_\star^{\frac{u'}{1-u}}\mathcal{E}_\star^{\frac{1}{1-u}}\right)^{\frac{2}{2+\gamma}} = \mathcal{E}_\star^{\frac{1+u'}{1-u}\frac{2}{2+\gamma}},$$

where $u' := \frac{2s'}{2s'+d}\frac{\beta}{2+\gamma}$. Therefore, we have

$$\overline{\mathcal{R}}_{\text{Pes}}^{(T)} \lesssim \left(\frac{n}{\log n}\right)^{-\frac{1+u'}{1-u}\frac{2}{2+\gamma}\frac{2s}{2s+d}}\log^{\frac{1+u'}{1-u}\frac{4}{2+\gamma}}\left(\frac{n}{\log n}\right),$$

which completes the proof. $\square$

## F. Differences between Assumption 5 and Assumption 9

In this section, we explain the differences between Assumption 5 and Assumption 9. The former is the assumption imposed for the analysis of the single-step algorithm, whereas the latter is the assumption imposed for the analysis of the multi-step algorithm, and the latter is stronger. Under this stronger assumption, we have shown that the multi-step algorithm achieves better regret than the single-step algorithm. We believe that this improvement is not merely a consequence of using stronger assumptions for the multi-step algorithm, but rather reflects an essential advantage of the multi-step algorithm itself. Indeed, as explained below, we believe that even if Assumption 9 were imposed on the single-step algorithm, its regret would not improve.

First, both assumptions share the conditions that the true reward $r^\circ$ belongs to $B_{p,q}^s$ and that $||r^\circ||_\infty$ is bounded.

Assumption 9(A1) imposes two conditions on the $\epsilon$-optimal set of the true reward $r^\circ$: **(i)** it contains a ball of radius $\Omega(\epsilon^{\gamma/d})$ centered at $y^*(x)$, and **(ii)** it is contained in a ball of radius $O(\epsilon^{\beta/d})$ centered at $y^*(x)$.

Among these, **(i)** is a strengthening of Assumption 9. Assumption 5 only requires the weaker condition that the volume of the $\epsilon$-optimal set is $\Omega(\epsilon^\gamma)$. In the multi-step analysis, however, we need the stronger geometric condition in Assumption 9(A1)**(i)** to ensure that, after Gaussian mollification, the maximizer remains inside the support of the updated policy. Since *the reward estimator at the next step is learned from samples drawn from the updated policy*, the support of that policy must continue to cover the maximizer. The condition **(i)** assumes that the $\epsilon$-optimal set contains an isotropic ball, which ensures that Gaussian mollification works to guarantee this. By contrast, *the single-step algorithm only learns a reward estimator from samples drawn from $\pi_{\text{ref}}$. Since the sampling distribution does not change across steps, this geometric*

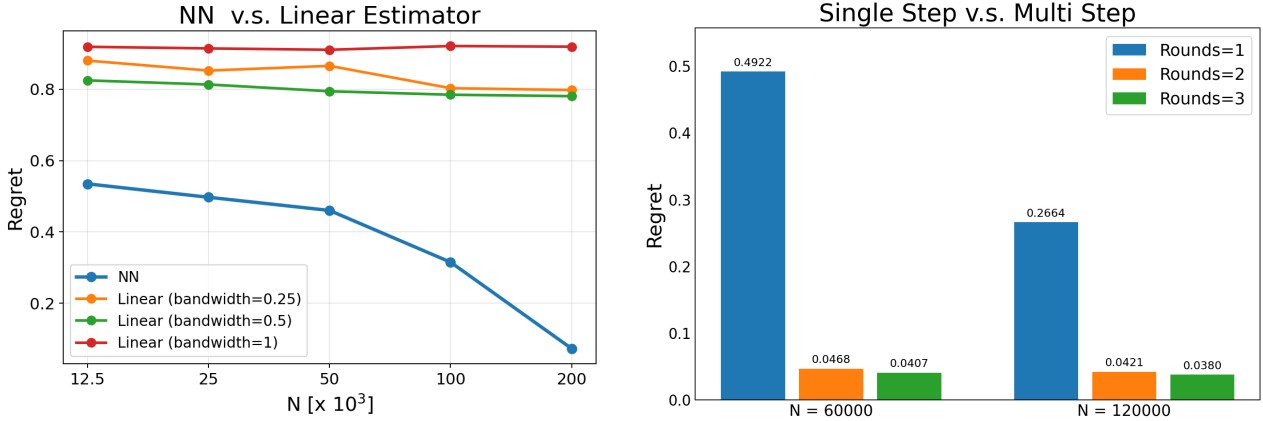

*Figure 2.* Experimental results. **Left:** Regret comparison between NN-based reward estimators and linear estimators under the single-step algorithm. **Right:** Regret comparison between the single-step and multi-step algorithms using NN-based reward estimators.

information cannot be exploited there. Therefore, replacing Assumption 5 by Assumption 9(A1)**(i)** does not improve the single-step regret analysis.

Assumption 9(A1)**(ii)** is used, roughly speaking, to show that *the multi-step algorithm can progressively shrink the support of the policy around the maximizer*. Because the policy in each step identifies a region with higher reward, this assumption allows us to conclude that the next policy has smaller support around the optimal response. The single-step algorithm has no such iterative support-shrinking mechanism: it only learns once from $\pi_{\text{ref}}$ and outputs a policy. Hence this assumption is useful specifically for the multi-step refinement, and does not provide an advantage for the single-step method.

(A2) and (A3) in Assumption 9 are used to obtain *sharper function-approximation guarantees on the smaller support produced by previous updates*. Condition (A2) ensures low complexity of the function class on the restricted domain. In addition, condition (A3) assumes that functions defined on the small support have regularity in the $x$-direction: since the narrowed response support is different for each prompt $x$, one must properly define the function class on this small support. Thus, these two conditions *are introduced to make full use of the benefit of the fact that the support of the policy has been narrowed and thereby obtain sharper approximation accuracy*, which in turn allows the support to be narrowed down further at the next step. Hence, they are not relevant to the single-step setting and therefore do not lead to improved regret there.

Finally, Assumption 9(A4) requires that $y^*(x)$ not lie too close to the boundary. This is imposed so that the multi-step algorithm can collect enough information around the maximizer. Even if this assumption was imposed on the single-step algorithm, *it would only reduce the search range of the response space by a constant factor*, so it seems unlikely to improve the regret rate.

Overall, while Assumption 9 is indeed stronger than Assumption 5, the additional strength is closely tied to the specific mechanism of the multi-step algorithm that iteratively narrows the policy support and improves the reward estimation on the small support. The single-step algorithm only uses samples from $\pi_{\text{ref}}$ for estimator learning, and therefore *cannot effectively benefit from these stronger assumptions*. For this reason, we expect that imposing Assumption 9 on the single-step algorithm would still not improve its regret rate.

## G. Numerical Experiments

In this section, we present the results of numerical experiments. We validate two main claims of our paper: (i) the importance of feature learning ability of NN-based reward estimators; and (ii) the advantage of the multi-step algorithm over the single-step algorithm.

We consider the true reward $r^\circ$ defined as follows:

$$r^\circ(x,y) = \frac{1}{d_Y}\left\{-\|y - y^*(x)\|^2 + \sum_{i=1}^{d_Y}\cos(2\pi q_i(y_i - y_i^*(x)))\right\}$$

where $y_i^*(x) = \frac{1}{2} + \frac{1}{2}\sin(2\pi w_i^\top x)$. Here, $q_i$ and $w_i$ are drawn independently from the uniform distribution on $\{1, \dots, 8\}$ and the sphere of radius $1/\sqrt{d_X}$ centered at the origin, respectively. In our experiments, we set $d_X = d_Y = 2$. We also set $\pi_{\text{ref}}$ to be the uniform distribution on $[0,1]^{d_X} \times [0,1]^{d_Y}$.

**Comparison of NNs and linear estimators**   We first compare NN-based reward estimators with linear estimators. We conduct this comparison in the single-step setting. That is, we first train each reward estimator using $N$ i.i.d. samples of prompt-response pairs, and then construct the policy by minimizing the $\chi^2$-regularized objective in (2). We set the coefficient of the regularization term to $\mu = 10^{-5}$. For the linear estimators, we use Nadaraya–Watson estimators with bandwidths in $\{0.25, 0.5, 1\}$. For the neural networks, we use fully connected networks of depth 3 and width 128. We optimize the networks for 200 epochs using the Adam optimizer and a cosine learning-rate scheduler with maximum learning rate $2 \times 10^{-3}$ and warmup during the first 10% of training.

The results are shown on the left side of Figure 2. We observe that, for all values of $N$, the neural network achieves smaller regret than the linear estimators. Moreover, the regret of the neural network decreases substantially as $N$ increases. These results indicate that, in inference-time alignment, neural networks with feature-learning ability outperform linear estimators without feature-learning ability.

**Comparison of single-step and multi-step algorithms**   We next compare the single-step and multi-step algorithms using NN-based reward estimators. Specifically, instead of training the reward estimator only on prompt-response pairs sampled from $\pi_{\text{ref}}$, we repeatedly retrain the reward estimator using samples from the constructed policy and then use the updated estimator to construct a new policy. We investigate whether the regret improves over this multiple-round procedure.

We conduct experiments with $N = 60000$ and $N = 120000$ available reward-oracle queries. We set the number of rounds to 1, 2 or 3, and examine whether increasing the number of rounds improves the regret under the same reward-oracle budget. Note that the total number of reward-oracle queries is kept fixed as the number of rounds varies: for example, when $N = 60000$, if the number of rounds is 1, 2, or 3, then the number of oracle queries available per round is 60000, 30000, or 20000, respectively.

We decay the regularization coefficient $\mu$ in the $\chi^2$-regularized objective across rounds: the initial value is set to $4 \times 10^{-3}$ for $N = 60000$ and $1 \times 10^{-3}$ for $N = 120000$, and it is halved from the previous round at each subsequent round. In this experiment, we skip the smoothing and mollification steps in Algorithm 1. The neural-network architecture and training settings are the same as those in the first experiment.

We present the results in the right half of Figure 2. For the same reward-oracle budget $N$, we observe that using more rounds leads to smaller regret. This empirical result supports our theory in Theorem 10: as the number of rounds increases, the reward estimator becomes more accurate around the optimal response, and the policy increasingly concentrates near the optimal response, which together lead to improved regret.

