# OpenReview forum: "Inference-time Alignment with Rewards in Besov Spaces: Provable Advantages of Feature Learning and Multi-Step Policy Updates"
_ICML.cc/2026/Conference — ICML 2026 regular_

### Official Review · Reviewer_F3q4 · 2026-03-01

**Soundness:** 3
**Presentation:** 3
**Significance:** 3
**Originality:** 3
**Overall Recommendation:** 4
**Confidence:** 3

**Summary:**

This paper investigates the theoretical foundations of inference-time alignment for large language models. Assuming that the true token advantage function resides within an anisotropic Besov space (to capture non-uniform smoothness), the authors rigorously prove that neural networks (NNs), leveraging their feature learning capabilities, can bypass the fundamental limitations of traditional linear estimators to achieve minimax optimal estimation rates. Furthermore, the paper proposes and mathematically bounds a multi-step iterative algorithm that alternates between policy sampling and refitting the NN token advantage model, demonstrating that this multi-step approach progressively tightens the policy and improves the overall regret bound.

**Compliance With Llm Reviewing Policy:**

Affirmed.

**Final Justification:**

I read the responses and I am satisfied.

**Key Questions For Authors:**

1. The analysis relies heavily on modeling prompts and responses in continuous vector spaces ($[0,1]^d$). While this is a standard theoretical abstraction, the gap to discrete, autoregressive token generation is substantial. Beyond acknowledging this as a limitation, could you elaborate on specific mathematical pathways to bridge this gap? For instance, would it involve redefining the token advantage function over discrete metric spaces, or adapting the Besov space assumptions to discrete graph structures?

2. In Algorithm 1, the optimal choice of the noise parameter $\sigma^{(\tau)}$ appears to depend on the unknown smoothness of the true token advantage function. How should this parameter be practically chosen when this smoothness is unknown? Furthermore, how much does the theoretical regret bound degrade if a suboptimal $\sigma^{(\tau)}$ is used?

**Limitations:**

Yes

**Strengths And Weaknesses:**

Stengths:

1. Novel Theoretical Perspective: Previous theoretical works in this area typically assume fixed token advantage models or oversimplify them as linear estimators. This paper successfully extends the analysis to NNs with feature learning, filling a significant theoretical gap.

2. Appropriate Mathematical Framing: The introduction of anisotropic Besov spaces to model the "rugged" landscape of the token advantage function is highly appropriate and aligns well with the complex local structures inherent in generation tasks.

3. Rigorous Algorithmic Justification: The theoretical progression from single-step estimation advantages to the design and rigorous regret bounding of the multi-step iterative algorithm is structurally elegant and logically sound.

Weaknesses

1. High Abstraction Gap:  The paper abstracts prompts and responses as continuous vectors $x \in [0,1]^{d_X}$ and $y \in [0,1]^{d_Y}$. This represents a significant gap from the practical, discrete autoregressive token generation process in actual LLMs. There are no empirical experiments or benchmark evaluations to ground the theoretical bounds.

---

> ### Author Rebuttal · Authors · 2026-03-31
>
> We thank the reviewer for the helpful feedback and for recognizing the strengths of our paper.
> We address the specific concerns and questions below.
>
> > 1. High Abstraction Gap: The paper abstracts prompts and responses as continuous vectors .... This represents a significant gap from the practical, discrete autoregressive token generation process
>
> Thank you for pointing this out.
> Our setting can be interpreted as assuming that prompts and responses are embedded into a single vector by a pretrained LM. (We will make this interpretation explicit in the revised paper.)
> Although texts are formally discrete, the space of possible candidates is enormous, and we believe it is reasonable to view them as being embedded in an effectively continuous space.
>
> At the same time, we agree that this leaves a substantial gap to modern autoregressive LLMs.
> We consider the abstraction above to isolate core theoretical issues: (i) whether and how the feature-learning capability of NNs can provably improve inference-time alignment beyond linear estimators, and (ii) how small a regret can be achieved using NN-based reward estimators and how that regret can be further reduced.
> This abstraction enables us to formalize local reward geometry and quantify the regularized objective and the regret bound in terms of the number of oracle queries $n$.
> We believe that these assumptions, which isolate the underlying theoretical challenges, are appropriate as a first step toward addressing the new problem of learning rewards for inference-time alignment.
>
> > 1. The analysis relies heavily on modeling prompts and responses in continuous vector spaces ... could you elaborate on specific mathematical pathways to bridge this gap?
>
> One possible bridge is to view the discrete autoregressive generation tree as embedded in a continuous latent space.
> In autoregressive generation, text can be represented as a branching tree that expands token by token.
> For example, with vocabulary size $W$ and prompt length $L$, the set of prompts is a discrete set of size $W^L$.
> One can imagine embedding these sequences into $[0,1]^d$ so that sequences with long common prefixes are mapped to nearby points.
>
> Under such a formulation, the gap between realistic discrete prompt/response spaces and our Besov-space assumption becomes more explicit and mathematically interpretable.
> In particular, the continuous regularity assumption can be viewed as a structural assumption on how autoregressive sequences are embedded in a latent vector space.
> We believe that formalizing this type of autoregressive generation together with its embedding into a continuous space would likely require a Transformer-based architecture, and we view this as an interesting direction for future work.
>
> > 2. In Algorithm 1, the optimal choice of the noise parameter $\sigma^{(\tau)}$ appears to depend on the unknown smoothness ... How should this parameter be practically chosen
>
> > Furthermore, how much does the theoretical regret bound degrade if a suboptimal $\sigma^{(\tau)}$ is used?
>
> We agree that the theoretically optimal choice of $\sigma^{(\tau)}$ depends on unknown problem parameters. Our current theory therefore provides a principled scaling law rather than a fully adaptive selector. The role of $\sigma^{(\tau)}$ is to balance exploitation of the current reward estimator and coverage of the near-optimal region: if it is too small, the policy may over-concentrate around a biased maximizer and lose coverage; if it is too large, the samples become too diffuse and fail to exploit the learned reward signal.
>
> The update of $\sigma^{(\tau)}$ in our theory is given on line 1813 of Appendix E.
> In simple terms, we initialize $\sigma^{(1)}$ as a decaying function of $n$, and update it as $\sigma^{(\tau)} = a_n (\sigma^{(\tau-1)})^u$, where $a_n$ is an $n$-dependent value (but independent of $\tau$), and $u$ is defined in Theorem 10.
> In practice, a natural implementation would be to tune $\sigma^{(1)}$, $a_n$, and $u$ over a small logarithmic grid. Designing an adaptive choice rule under unknown smoothness is an interesting future direction.
>
> We do not currently provide a separate theorem quantifying regret degradation under misspecified $\sigma^{(\tau)}$. However, the proof makes the direction clear: overly small $\sigma^{(\tau)}$ harms coverage, while overly large $\sigma^{(\tau)}$ weakens exploitation. Since our regret bound is driven by the balance between estimation error and coverage, misspecifying $\sigma^{(\tau)}$ worsens the bound through one of these two channels. Providing an explicit bound for misspecified $\sigma^{(\tau)}$ would be a valuable extension.
>
> ---
> We would be happy to clarify any concerns or answer any questions that may come up during the discussion period. We would greatly appreciate it if you could consider increasing the score once all concerns have been resolved.

---

> > ### Author Rebuttal · Reviewer_F3q4 · 2026-04-01
> >
> > my concerns have been addressed.

---

### Official Review · Reviewer_N3Hg · 2026-03-10

**Soundness:** 3
**Presentation:** 2
**Significance:** 2
**Originality:** 2
**Overall Recommendation:** 4
**Confidence:** 4

**Summary:**

This paper studies inference-time alignment. The authors assume the true reward function lies in the Besov space and analyze learning the reward model using neural networks. They show that neural-network reward estimators can achieve better approximation error than linear estimators. The paper also proposes a multi-step algorithm that repeatedly updates the reward model and policy, and proves that this iterative procedure can reduce regret compared to a single-step approach.

**Compliance With Llm Reviewing Policy:**

Affirmed.

**Final Justification:**

In the rebuttal, the authors provide a detailed comparison of the assumptions used for single- and multi-step analysis. Thanks to this, I'm now convinced the paper does prove something important that the multi-step algorithm improves regret over a simple approach. Although, I still think the message that feature learning plays a good role in inference-time alignment not very surprising, I acknowledge this paper is overall good, and I lean towards acceptance.

**Key Questions For Authors:**

What is the key novel message of this paper, except that neural network can approximate better than linear estimators?

**Limitations:**

The paper writes a section on limitations, which describes the extension to token-sequence models or transformers. However, even when we only consider the theory with the NN of MLP with Relu, there are still limitations that have not been well specified.

**Strengths And Weaknesses:**

**Strengths**:

1. The studied problem: inference-time alignment is very important and interesting.

2. It works on the ability of feature learning on the inference-time generation problem, and proposes new regret bounds.

3. It proposes a new algorithm of multi-step training for the reward model training, which is a new recipe for this field.

**Weaknesses**:

1. The main weakness of this paper is the novelty. Although the paper introduces several concepts, such as Besov spaces, and presents results with complicated expressions, the key message can be summarized in one sentence: In the literature of inference time alignment, existing works usually make an assumption on the accuracy of the reward function. This paper replaces this assumption with the existing approximation theory in the Besov space. To see this, we can check

(1) The proof of Theorem 3: although the claimed theorem seems complex and involves the hyperparameters of neural networks, the proof is very straightforward, and the only step is to reduce the loss to the estimated reward error. Once this is done, it remains to call Theorem 2 in Suzuki (2018).

(2) The argument of the lower bound for the linear estimator (Theorem 4) can be found in parallel in Suzuki (2018). Moreover, it does not seem quite related to the remaining part of the paper.

(3) The proof of Theorem 6: the proof just replaces the estimated reward error $\mathcal{E}^2(x)$ with the approximation error obtained with the Besov space.

Overall, while connecting inference-time alignment with approximation theory is not problematic in itself, the way this connection is used in the paper appears somewhat isolated across different parts. The analysis mainly relies on existing approximation results rather than introducing new techniques or insights. As a result, the paper does not clearly demonstrate any novel conceptual message or methodological novelty.

2. The mentioned algorithm, Inference-time pessimism, is an algorithm proposed in Huang et al. (2025b). The keyword is used before the introduction.

3. The claim in Theorem 6 is not accurate. In its proof, it uses Theorem 17, which only says there **exists** a function that can approximate the ground truth. However, in the algorithm, the reward function is obtained by empirical risk minimization, which does not have such guarantees. Therefore, the bound on $\pi^{pes}$ is wrong.

4. The exact same equation on the empirical risk minimization (or least squares estimators) is written twice in the paper.

5. The proof of Theorem 3 is ambiguous. For example, what is the norm and inner product in Line 1175? Is it the Euclidean norm or the $H_x$ norm? Moreover, this result of reducing regularized loss to the square loss of the reward function is not new, which can be seen, e.g., Theorem 3.2 in [1].

6. The assumptions are strong and do not seem realistic.

---

[1] Towards a Sharp Analysis of Offline Policy Learning for f-Divergence-Regularized Contextual Bandits, Zhao et al. ICLR2026

---

> ### Author Rebuttal · Authors · 2026-03-31
>
> We thank the reviewer for the helpful feedback.
> > 1. The main weakness of this paper is the novelty ... replaces this assumption with the existing approximation theory
>
> We agree that parts of our analysis build on existing approximation/estimation results for Besov spaces.
> However, our contribution is **not simply to replace an accuracy assumption by existing approximation theory**.
>
> We study reward-model training for inference-time alignment under a finite reward-oracle budget, **an important but underexplored setting**.
> In this setting, **we provide two messages**: (i) feature learning, known to matter in regression, also matters in inference-time alignment; and (ii) the multi-step algorithm improves regret over a simple approach.
>
> > To see this, we can check (1) The proof of Theorem 3 ...
>
> > (2) The argument ... (Theorem 4) can be found in parallel in Suzuki (2018).
>
> Theorems 3 and 4 are not mere restatements of prior regression-error results.
> Our goal there is to show that the known separation between NNs and linear estimators in regression **translates into a separation in the regularized alignment objective**.
> Thus, the contribution is not the Besov approximation theory itself, but the observation that **feature-learning advantages persist in inference-time alignment**.
>
> > (3) The proof of Theorem 6
>
> We agree that the technical novelty of Theorem 6 is modest. However, **it still serves as a baseline for our new setting**. More importantly, **our contribution goes beyond providing this baseline**: we propose and analyze a multi-step algorithm and prove an improved regret bound over the single-step method (Theorem 10). This part is technically novel, and its value is highlighted by the comparison with the single-step baseline.
>
> > Moreover, it does not seem quite related to the remaining part
>
> > Overall, ... isolated across different parts.
>
> Rather than being isolated, **our theorems establish two complementary insights**.
> Theorems 3 and 4 show the importance of feature learning in inference-time alignment, while Theorems 6 and 10 analyze NN-based algorithms and compare the single-step and multi-step methods.
>
> >  the paper does not clearly demonstrate any novel conceptual message or methodological novelty
>
> As clarified above, the paper has two messages: the importance of feature learning in inference-time alignment, and the advantage of the multi-step policy update. The first point relies partly on prior results, but **is still meaningful because it transfers them to a new setting**. The second has **novelty in both methodology and analysis**.
>
> > 2. Inference-time pessimism, is an algorithm proposed in Huang et al. (2025b). The keyword is used before the introduction.
>
> The first mention of Inference-time Pessimism appears in the final paragraph of Section 2.1, where it is introduced as an algorithm of Huang et al. (2025b). We will also add the citation at the later occurrence in Line 242.
>
> > 3. The claim in Theorem 6 is not accurate.
>
> Thank you for pointing this out. The current proof of Theorem 6 cites Theorem 17, which is only an approximation result. The estimation guarantee should instead rely on Theorem 21. We will correct this in the revision. We also note that the logarithmic factor should be $\log^4$, and we will revise the statement accordingly.
>
> > 4. The exact same equation on the empirical risk
>
> We agree and will remove the duplication.
>
> > 5. The proof of Theorem 3 is ambiguous.
>
> Thank you for pointing out this ambiguity.
> The norms and inner products in Lines 1175--1191 should be those of $\mathcal H_x$.
> We will clarify them in the revision.
> > Moreover, ... reducing regularized loss to the square loss of the reward function is not new ...
>
> Thank you for pointing out this connection. We will add a citation to [1] and clarify that the proof idea for reducing the regularized loss to the estimation error is similar. Our contribution is not the proof technique itself, but showing that feature learning matters not only for regression but also for inference-time alignment, by bringing techniques from the two settings together.
>
> > 6. The assumptions are strong...
>
> The assumptions are chosen to cover a broad class of reward landscapes, and **their interpretation and roles are discussed in the remarks following each assumption**. Besov spaces include many standard smoothness classes such as Hölder and Sobolev spaces. Assumption 5 and Assumption 9 (A1) formalize the local geometry around the maximizer and can be viewed as generalizations of standard local curvature conditions. Assumption 9 (A2)–(A4) are technical conditions for the multi-step analysis, and the paper gives interpretations and standard sufficient cases.
>
> > What is the key novel message of this paper,
>
> Please see the responses to Weakness 1.
>
> ---
> We would be happy to clarify any remaining concerns during the discussion period. We would appreciate it if you could reconsider the score once these concerns have been resolved.

---

> > ### Author Rebuttal · Reviewer_N3Hg · 2026-04-02
> >
> > Thanks for your detailed responses. After reading them, I still hold some questions.
> > 1. "**feature learning, known to matter in regression, also matters in inference-time alignment**" Given knowledge of reducing the regularized loss to the estimation error, the results are not so surprising, as regression and (inference-time) alignment are already connected and therefore it's a little straightforward to see that the results that work in regression will continue to work for alignment.
> > 2. "**the advantage of the multi-step policy update**" I did not write the questions on this part clearly in the original review by justing saying "the assumptions are strong". Now I can make it clearer. The goal of section 5 (and the claimed main contribution) is to show that multi-step algorithms have a better regret bound than the single-step ones. However, it also relies on different assumptions. In particular, Section 5 depends on Assumption 9 which has 4 conditions, while Section 4 depends on Assumption 5 with 2 conditions. I do not quite understand the comparison of these assumptions. If Assumption 9 is much more stringent than Assumption 5, the advantage may not come from the algorithm design, but from the benefit of another assumption. Could you clarify more on the relations of these assumptions?
> >
> > I'm open to change my mind if you could further resolve these questions, especially the second one. Thank you.
> >
> > ---
> > Post-rebuttal: I appreciate the response by the authors, and I'm satisfied with the comparison of assumptions. I suggest that the authors should add the discussions in the revision, in the main text or the appendix. Thus, I increase my score to reflect this.

---

> > > ### Author Response · Authors · 2026-04-05
> > >
> > > Thank you for your additional comments.
> > >
> > > > Given knowledge of reducing the regularized loss to the estimation error, the results are not so surprising
> > >
> > > As you pointed out, the main technical contribution of our paper does not lie in Theorem 3 (and 4).
> > > However, we believe there is value in **bringing classical techniques into the new setting of inference-time alignment and showing that feature learning is important there as well**.
> > > In our view, the key point is not Theorem 3 in isolation, but rather the fact that, together with Theorem 4, it clarifies the importance of feature learning in this setting.
> > >
> > > > Section 5 depends on Assumption 9 which has 4 conditions, while Section 4 depends on Assumption 5 with 2 conditions. I do not quite understand the comparison of these assumptions.
> > >
> > > Thank you for raising this important question.
> > > Assumption 9 is indeed stronger than Assumption 5, and the multi-step algorithm does require Assumption 9.
> > > However, **we do not believe that imposing Assumption 9 on the single-step algorithm would improve its regret**.
> > >
> > > First, both assumptions share the conditions that the true reward $r^\circ$ belongs to $B_{p,q}^s$ and that $||r^\circ||_\infty$ is bounded.
> > >
> > > Assumption 9 (A1) imposes two conditions on the $\epsilon$-optimal set of the true reward $r^\circ$:
> > > **(i)** it contains a ball of radius $\Omega(\epsilon^{\gamma/d})$ centered at $y^\ast(x)$, and
> > > **(ii)** it is contained in a ball of radius $O(\epsilon^{\beta/d})$ centered at $y^\ast(x)$.
> > >
> > > Among these, **(i)** is a strengthening of Assumption 5.
> > > Assumption 5 only requires the weaker condition that the volume of the $\epsilon$-optimal set is $\Omega(\epsilon^\gamma)$.
> > > In the multi-step analysis, however, we need the stronger geometric condition in Assumption 9(A1)(i) to ensure that, after Gaussian mollification, the maximizer remains inside the support of the updated policy.
> > > Since **the reward estimator at the next step is learned from samples drawn from the updated policy**, the support of that policy must continue to cover the maximizer.
> > > (A1)(i) assumes that the $\epsilon$-optimal set contains an isotropic ball, which ensures that Gaussian mollification works to guarantee this.
> > > By contrast, **the single-step algorithm only learns a reward estimator from samples drawn from $\pi_{\mathrm{ref}}$**.
> > > Since the sampling distribution does not change across steps, this geometric information cannot be exploited there.
> > > Therefore, replacing Assumption 5 by Assumption 9(A1)(i) does not improve the single-step regret analysis.
> > >
> > > Assumption 9(A1)(ii) is used, roughly speaking, to show that **the multi-step algorithm can progressively shrink the support of the policy around the maximizer**.
> > > Because the policy in each step identifies a region with higher reward, this assumption allows us to conclude that the next policy has smaller support around the optimal response.
> > > **The single-step algorithm has no such iterative support-shrinking mechanism**: it only learns once from $\pi_{\mathrm{ref}}$ and outputs a policy.
> > > Hence this assumption is useful specifically for the multi-step refinement, and does not provide an advantage for the single-step method.
> > >
> > > Assumptions 9(A2) and (A3) are used to obtain **sharper function-approximation guarantees on the smaller support produced by previous updates**.
> > > Condition (A2) ensures low complexity of the function class on the restricted domain.
> > > In addition, condition (A3) assumes that functions defined on the small support have regularity in the $x$-direction: since the narrowed response support is different for each prompt $x$, one must properly define the function class on this small support.
> > > Thus, these two conditions **are introduced to make full use of the benefit of the fact that the support of the policy has been narrowed and thereby obtain sharper approximation accuracy**, which in turn allows the support to be narrowed down further at the next step.
> > > Hence, they are not relevant to the single-step setting and therefore do not lead to improved regret there.
> > >
> > > Finally, Assumption 9(A4) requires that $y^\ast(x)$ not lie too close to the boundary. This is imposed so that the multi-step algorithm can collect enough information around the maximizer. Even if this assumption was imposed on the single-step algorithm, **it would only reduce the search range of the response space by a constant factor**, so it seems unlikely to improve the regret rate.
> > >
> > > Overall, while Assumption 9 is indeed stronger than Assumption 5, the additional strength is closely tied to the specific mechanism of the multi-step algorithm that iteratively narrows the policy support and improves the reward estimation on the small support.
> > > The single-step algorithm only uses samples from $\pi_{\mathrm{ref}}$ for estimator learning, and therefore **cannot effectively benefit from these stronger assumptions**.
> > > For this reason, we expect that imposing Assumption 9 on the single-step algorithm would still not improve its regret rate.

---

### Official Review · Reviewer_aqPY · 2026-03-12

**Soundness:** 3
**Presentation:** 2
**Significance:** 3
**Originality:** 3
**Overall Recommendation:** 4
**Confidence:** 2

**Summary:**

This paper studies the impact of model selection to represent the true reward function during inference-time alignment. In particular, the paper compares neural networks and linear estimators. The contributions list shows that neural networks are superior to linear estimators in that regard in the setting of a true reward function in Besov spaces. Two results for regret are also proposed: an upper bound, and an analysis of the regret through a multi-step algorithm. The paper proposes a vast theoretical framework for future implementation in transformer inference-time alignment.

**Compliance With Llm Reviewing Policy:**

Affirmed.

**Key Questions For Authors:**

1. Besides the future direction for research on transformer-based inference-time alignment, do you plan on adding experimental results?

2. Do you expect the Besov space assumption to hold as the model you are trying to align grows?

3. Are there any stronger assumption(s) that you will have to make in order to bridge the gap with a realistic setting?

**Limitations:**

Yes

**Strengths And Weaknesses:**

Strength:

- Formally proving the superiority of neural network in the setting of inference-time alignment is directly beneficial to future research.

- The contrast between the singlet step algorithm and the multi-step proposed is clearly established in section 4 and 5.

- The link between inference-time alignment and the work done on reward modeling is clearly stated through regret analysis.

- In the case where the Besov assumption holds true the multi-step process should be to my understanding a direct upgrade to the single-step algorithm.

Weaknesses:

- The entire paper hinges on the assumption that the true reward lies in a Besov space. Whether this assumption is reasonable is up to debate. The capacity to encode both smoothness and roughness is a convenient property to justify this choice.

- The paper lacks an experimental setting in which this assumption can be verified empirically.

- Similarly, the superiority of NN over linear estimators is demonstrated theoretically but the lack of experimental results weakens the claim.

- The paper is really dense in notations. This is expected given the nature of the theoretical objects manipulated but I can imagine the implementation of this method from a third party would be arduous.

---

> ### Author Rebuttal · Authors · 2026-03-31
>
> We thank the reviewer for the helpful feedback.
> We also thank the reviewer for recognizing the strengths of our paper.
> We address the specific concerns and questions below.
>
> > The entire paper hinges on the assumption that the true reward lies in a Besov space.
>
> Thank you for raising this concern. We use Besov spaces not merely for convenience, but because they provide a natural function class for studying reward landscapes with spatially non-uniform smoothness, which is precisely the regime where feature learning can affect the statistical rate and alignment performance. At the same time, **we do not view this assumption as overly specialized**: Besov spaces are a standard and broad class that includes familiar smoothness spaces such as Hölder- and Sobolev-type classes. We therefore view the Besov assumption as a principled analytical model rather than a narrowly tailored technical choice.
>
> > The paper lacks an experimental setting ...
>
> > Similarly, the superiority of NN over linear estimators ... the lack of experimental results
>
> > 1. Besides ..., do you plan on adding experimental results?
>
> Thank you for this constructive suggestion.
> **We conducted synthetic experiments that strengthen our theoretical claim.**
> The experimental setup and results are summarized in the figure at the following URL.
> We will add these experimental results in the revision.
>
> https://drive.google.com/file/d/1bceZ5cZn268glA6KJG197eLzJZNbV6X4/view?usp=sharing
>
> In the first experiment, we construct policies using NN-based and linear reward estimators by optimizing the $\chi^2$-regularized objective, and compare their regret. The results are displayed on the left side of the figure. We observe that the policies induced by NN-based estimators achieve substantially smaller regret than those induced by linear estimators, consistent with our theoretical result on the advantage of feature learning.
>
> In the second experiment, we compare the single-step and multi-step algorithms. The multi-step version alternates between reward estimation using samples from the current policy and policy improvement via the $\chi^2$-regularized objective. The results are shown on the right side of the figure. We observe that multi-step updates yield smaller regret, matching our theoretical claim on the benefit of iterative policy updates.
>
> > The paper is really dense in notations. ... the implementation of this method from a third party would be arduous.
>
> Thank you for this comment. We agree that the presentation is technically dense. **This partly comes from the level of generality we consider**: Besov spaces are a broad function class that allows spatially non-uniform smoothness, and obtaining rigorous guarantees in this setting requires additional technical tools. While we view this generality as one of the main contributions of the paper, we agree that it also creates practical limitations, and we will clarify this point in the revision.
>
> That said, **the core structure of Algorithm 1 is simple**: each iteration alternates between fitting a neural-network reward estimator from samples of the current policy and updating the policy through the $\chi^2$-regularized objective. The intuition behind this is explained in the first two paragraphs of Section 5.1 and Figure 1. Although our formal guarantees are stated for Besov rewards, we believe the insights on the advantage of multi-step updates extend beyond the Besov setting. Extending this perspective to other structural assumptions and more practical algorithms is an important direction for future work.
>
> > 2. Do you expect the Besov space assumption to hold as the model you are trying to align grows?
>
> In our formulation, the Besov assumption is imposed on the true reward function, not on the policy or model being aligned, so its validity is largely orthogonal to model size. Moreover, Besov spaces form a broad class containing Hölder- and Sobolev-type spaces, so we do not view the assumption as overly restrictive.
>
> > 3. Are there any stronger assumption(s) that you will have to make in order to bridge the gap with a realistic setting?
>
> A natural next step is to extend the analysis to autoregressive token generation and Transformer-based reward estimators. To obtain results analogous to those in the current paper, one would need sequence-level counterparts of the assumptions used here, such as sufficient reference-policy mass on near-optimal responses and structural assumptions around the maximizer, e.g., local regularity or concentration of super-level sets. One possible formalization is to assume that token sequences admit a continuous embedding into a representation space where analogous regularity conditions hold. We view this as an important direction for future work.
>
> ---
> We would be happy to clarify any concerns or answer any questions that may come up during the discussion period. We would greatly appreciate it if you could consider increasing the score once all concerns have been resolved.

---

> > ### Author Rebuttal · Reviewer_aqPY · 2026-04-03
> >
> > My main concern was about the lack of experimental settings, which other reviewers have pointed out. The authors have proposed to add experimental results as part of a revised version. This satisfies the questions I had, and I will keep my score as is.

---

### Official Review · Reviewer_u3tw · 2026-03-13

**Soundness:** 3
**Presentation:** 2
**Significance:** 3
**Originality:** 3
**Overall Recommendation:** 4
**Confidence:** 4

**Summary:**

The paper considers the problem of inference time alignment via optimizing a learned reward function from noisy observations. The paper adopts the framework of optimizing a reward function under $\chi^2$ divergence regularization, which has been shown in previous work (Huang et al., 2025) to be effective for preventing over-optimization of the reward function. The theoretical contribution are two-fold:
- The paper first compare the performance of neural network model with linear models for learning the reward function in a Besov space with Gaussian noise. The paper shows that the greedy-optimal policy from the neural network reward learner achieves the optimal rate in nonparametric regression via feature learning, while the greedy-optimal policy from the linear reward learner suffers from a suboptimal rate. This is consistent with existing work on feature learning via Besov spaces (e.g., Suzuki, 2018).
- The paper derives regret guarantees for the greedy-optimal policy from the neural network reward learner.
- To further improve the performance of the inference-time alignment strategy, the paper considers a multi-step policy update strategy, which interactively updates the policy and the reward estimator. By sampling points closer to the optimality and focusing on the local region for the reward function learner, the regret can be improved based on structural parameters in the Besov space.

**Compliance With Llm Reviewing Policy:**

Affirmed.

**Key Questions For Authors:**

In addition to aforementioned weakness.
- Theorem 4, condition (c), what does the notation $\ll$ mean here? And how is this condition verified?
- To what extent the analysis depends on the Gaussian noise assumption? Can it be relaxed?

**Limitations:**

The paper adequately discussed the limitations and potential negative societal impact.

**Strengths And Weaknesses:**

Strengths: Inference-time alignment via reward optimization is an increasingly important direction for post-training of deep generative models. The paper makes a solid step towards understanding its theoretical properties through nonparametric regression perspective. The theoretical results are solid, and the multi-step policy update strategy is a novel statistical idea. Overall, this is a nice paper in theoretical statistics with well-motivated background from post training methods.

Weaknesses: Though theoretically solid, the paper has some limitations.
- The separation between neural networks and linear models are somewhat expected, given the existing literature on nonparametric regression with Besov spaces.
- The multi-step policy update strategy is very specialized to Besov spaces, involving smoothing kernel schedule and sampling strategy that are designed based on structures of Besov spaces. The Besov space is more used as a prototypical function space to illustrate the advantage of feature learning, instead of a practical choice for reward function learning. It is therefore unclear how the multi-step policy update strategy can be applied to more general function spaces, and whether it can be applied to practical reward function learning problems.

---

> ### Author Rebuttal · Authors · 2026-03-31
>
> We thank the reviewer for the helpful feedback.
> We also thank the reviewer for recognizing the contributions and the novelty of our paper.
> We address the specific concerns and questions below.
>
> > The separation between neural networks and linear models are somewhat expected,
>
> Thank you for raising this concern.
> We agree that such a separation is broadly consistent with the nonparametric regression literature on Besov spaces.
> Our contribution here is not to provide a novel technique for proving separation between NNs and linear estimators, but rather to show that **such a gap translates into the inference-time alignment objective**.
> Concretely, Theorems 3 and 4, which compare the value of the regularized objective rather than the regression error, show the separation specialized to inference-time alignment.
> The rates are obtained by relating the objective gap to nonparametric regression error and then invoking known estimation rates.
> We believe this connection is important because it shows that feature-learning theory for regression can be used to rigorously analyze inference-time alignment, thereby bridging two lines of work that have mostly been studied separately.
>
> > The multi-step policy update strategy is very specialized to Besov spaces, involving smoothing kernel schedule and sampling strategy that are designed based on structures of Besov spaces. ... It is therefore unclear how the multi-step policy update strategy can be applied to more general function spaces, and whether it can be applied to practical reward function learning problems.
>
> Thank you for this important comment.
> We agree that our algorithm is designed based on the structure of Besov spaces, and that its practicality in the present form is open to discussion.
> Because of the generality of Besov spaces, providing rigorous theoretical guarantees required technically involved operations such as kernel smoothing and carefully chosen hyperparameters.
>
> That said, we believe the core idea is more general than this instantiation.
> The central mechanism is to progressively concentrate the sampling distribution around the maximizer through repeated policy updates.
> In our analysis, the gain from multi-step updates comes from the geometric assumptions on the super-level sets in Assumption 9, and it appears through improved estimation error and improved coverage, as formalized in Lemmas 11 and 12; Figure 1 provides the corresponding intuition.
> We therefore expect similar effects to arise beyond the Besov setting as well.
>
> While we do not prove such extensions in the current version, establishing analogous results for other function classes and connecting them to practical reward-learning problems is an important direction for future work.
> Moreover, **to strengthen our paper, we also conducted synthetic experiments** supporting our two main messages: (i) the importance of feature learning, and (ii) the advantage of multi-step policy updates.
> Please also see our response to Reviewer aqPY.
>
> > Theorem 4, condition (c), what does the notation $\ll$ mean here? And how is this condition verified?
>
> Thank you for pointing out this ambiguity. In condition (c), $\ll$ means that the left-hand side is $o_n(\varepsilon^2)$, where $n$ is the number of oracle queries; we will replace $\ll$ with $= o_n(\varepsilon^2)$ for clarity.
>
> This condition implies that the prompt-averaged $L^2$-error between $\hat{r}_{\mathrm{lin}}$ and $r^\circ$ is asymptotically negligible compared with the full $L^2$-error.
> The prompt-averaged error is generally smaller than the full $L^2$-error.
> Since the rate of such an error is generally sensitive to the dimension of the space, it is expected that the former is negligibly smaller than the latter.
>
> > To what extent the analysis depends on the Gaussian noise assumption? Can it be relaxed?
>
> The Gaussian assumption is mainly for analytical convenience, and we expect the main results to extend to broader noise classes.
>
> For the lower bound for linear estimators (Theorem 4), the proof essentially relies only on the assumptions of zero mean and finite variance. Therefore, this part can be generalized to a broader class of noise satisfying these conditions.
>
> For the upper bounds for neural networks (Theorems 3, 6, and 10), we rely on estimation-error results from Schmidt-Hieber (2020) and Hayakawa & Suzuki (2020), which are stated under Gaussian noise. We expect that these results can be extended to the sub-Gaussian case. However, since the role of these theorems is to derive guarantees on the regularized objective or the regret for inference-time alignment, rather than to generalize the underlying nonparametric regression theory itself, we chose to work under this standard and well-understood setting.
>
> ---
> We would be happy to clarify any concerns or answer any questions that may come up during the discussion period. We would greatly appreciate it if you could consider increasing the score once all concerns have been resolved.

---

> > ### Author Rebuttal · Reviewer_u3tw · 2026-04-03
> >
> > I thank the authors for the response, which helped me understand the technical contents better. Considering the technical contribution as well as practical limitation, I think this is a borderline paper, and I decide to keep my score.

---

### Decision · Program_Chairs · 2026-04-30

**Decision:**

Accept (regular)

**Comment:**

The paper establishes theoretical foundations for LLM inference-time alignment. Modeling the token advantage function in an anisotropic Besov space, the authors prove that neural networks bypass the limitations of traditional linear estimators to achieve minimax optimal rates. Additionally, they introduce a multi-step iterative algorithm that alternates between policy sampling and model refitting, mathematically demonstrating that this approach progressively improves regret bounds.

Interesting and timely paper showing that feature learning capabilities of NN improves inference-time alignment beyond linear estimators. The paper is mathematically heavy and relies on known approximation-theory results. The final outcome is, in broad strokes, not very surprising, but it’s nice to see it spelled out in such a rigorous manner. All reviewers eventually agreed to support this paper.